# Generalization for Multiclass Classification with Overparameterized Linear Models

**Vignesh Subramanian**
Department of Electrical Engineering and Computer Sciences
University of California Berkeley
Berkeley, CA-94720, USA
vignesh.subramanian@eecs.berkeley.edu

**Rahul Arya**
Department of Electrical Engineering and Computer Sciences
University of California Berkeley
Berkeley, CA-94720, USA
rahularya@berkeley.edu

**Anant Sahai**
Department of Electrical Engineering and Computer Sciences
University of California Berkeley
Berkeley, CA-94720, USA
sahai@eecs.berkeley.edu

## Abstract

Via an overparameterized linear model with Gaussian features, we provide conditions for good generalization for multiclass classification of minimum-norm interpolating solutions in an asymptotic setting where both the number of underlying features and the number of classes scale with the number of training points. The survival/contamination analysis framework for understanding the behavior of overparameterized learning problems is adapted to this setting, revealing that multiclass classification qualitatively behaves like binary classification in that, as long as there are not too many classes (made precise in the paper), it is possible to generalize well even in some settings where the corresponding regression tasks would not generalize. Besides various technical challenges, it turns out that the key difference from the binary classification setting is that there are relatively fewer positive training examples of each class in the multiclass setting as the number of classes increases, making the multiclass problem "harder" than the binary one.

## 1 Introduction

Multiclass classification on standardized datasets is where the current deep-learning revolution really made the community take notice with previously unattainable levels of performance. Contemporary systems have demonstrated tremendous success at these tasks, typically using gigantic models with parameters that vastly exceed the (also large) number of data points used to train these models. In defiance of traditional statistical wisdom regarding overfitting, these big models can be trained to achieve zero training error even with noisy labels, but still generalize well in practice [84, 28].

To better understand this empirical phenomenon, one line of work uses appropriate high-dimensional linear models for regression problems to show how benign fitting of noise in training data is possible [31, 52, 4, 9, 55]. Essentially, the model must have enough "non-preferred" degrees of freedom to be able to absorb the training noise without contaminating predictions by too much. Simultaneously, there has to be enough of a preference for degrees of freedom that can capture the true pattern to enable it to survive the learning procedure and be well represented in the final learned model.

36th Conference on Neural Information Processing Systems (NeurIPS 2022).

A subsequent line of work studies binary classification [56, 13, 76] and shows that binary classification can generalize well beyond what can be proved by classical margin-based bounds [3] and there exist regimes where binary classification can even succeed in generalizing where regression fails — less preference is required for the degrees of freedom that capture the true pattern [56]. Very recently, the generalization of multiclass classification in similar models was studied in Wang et al. [77] but the analysis was limited to a fixed finite number of classes. In practice, we see that larger datasets often come with more classes and are tackled with even bigger models and so it is important to see what happens to generalization when everything scales together. To have a crisply understandable approach that allows everything to scale, this paper also adopts the bi-level covariance model with Gaussian features that is used in Muthukumar et al. [55, 56], Wang et al. [75], Wang et al. [77].

To understand classification, we must understand the role of training loss functions in determining what is learned. Empirical evidence shows that least-squares can yield classification performance competitive to cross-entropy minimization [64, 35, 11]. Muthukumar et al. [56], Hsu et al. [33] show that indeed with sufficient overparameterization, the support vector machine (SVM) solution, which also arises from minimizing the logistic loss using gradient descent [68, 36], is identical to that obtained by the minimum-norm interpolation (MNI) of binary labels — what would be obtained by gradient descent while minimizing the squared loss. A similar equivalence[1] holds for different variations of multiclass SVMs and the MNI of one-hot-encoded labels [77]. Consequently, this paper focuses on the MNI approach to overparameterized learning for multiclass classification.

## 2    Our contributions

Our study provides an asymptotic analysis of the error of the minimum-norm interpolating classifier for the multiclass classification problem with weighted Gaussian features. We consider an overparameterized setting using a bi-level feature weighting model where the number of features, classes, favored features, and the feature weights themselves all scale with the number of training points. Under this model, Theorem 5.1 provides sufficient conditions for good generalization in the form of a region in which as the number of training points increase, the number of classes grows slowly enough, the total number of features (i.e. level of overparameterization) grows fast enough, the number of favored features grows slowly enough, and the amount of favoring of those favored features is sufficient to allow for asymptotic generalization. We assume that our labels are generated noiselessly based on which of the first $k$ features is the largest.[2]

To prove our main result, Theorem 5.1, we present a novel typicality-style argument featuring the feature margin (gap between the largest and second-largest feature) for computing sufficient conditions for correct classification utilizing the signal-processing inspired concepts of survival and contamination from Muthukumar et al. [55, 56] and leveraging the random-matrix analysis tools sharpened in Bartlett et al. [4]. The survival concept relates to the shrinkage induced by the regularizing effect of having lots of features in the context of min-norm interpolation — survival captures what is left of the true pattern after shrinkage. Contamination reflects the consequence of overparameterization when training via optimization: in addition to the true pattern, there is an infinite family of other[3] false patterns (aliases) that also happen to explain the limited training data, and the optimizer ends up hedging its bet across the true pattern and these other competing false explanations. The learned false patterns contaminate the predictions on test points, and this can be quantified by the relevant standard deviation.

The key is analyzing what happens with multiclass training data where there are relatively fewer positive examples of each class, and where the training data for a particular class is not independent of the features corresponding to other classes. The analysis shows that as a result of having fewer positive exemplars for a class relative to the total size of the training data, the survival drops by a factor of $k$ (the number of classes), while the contamination only drops by a factor of $\sqrt{k}$. As in binary classification, the ratio of the relevant survival to contamination terms plays the role of the effective signal-to-noise ratio and shows up as a key quantity in our error analysis (Equation (22) from

---

[1] For an interesting alternative perspective on this equivalence as an indication of a potential bug instead of as a promising feature, see Shamir [67].

[2] This assumption is without loss of generality for the bi-level model as long as the classes are defined by orthogonal directions as in Wang et al. [77].

[3] This is related to what is called the challenge of "underspecification" in ML [19], and this in turn is also one aspect of the challenge of covariate shifts [73].

Section 5.1). When this ratio grows asymptotically to $\infty$, multiclass classification generalizes well. To the best of our knowledge, this is the first work that quantifies this effect of fewer informative samples per class and in what sense that makes multiclass classification harder than binary classification. The closest related work ([77]) only considers multiclass classification in the fixed finite class setting and consequently, doesn't compute exact dependencies on the number of classes $k$. We provide a more detailed comparison of our work with Wang et al. [77] and Muthukumar et al. [56] in Appendix H of the Supplemental material.

## 3 Related Work

The present work is situated within a larger stream of theoretical research trying to understand why overparameterized learning works and its limits. The limited page budget here forces brevity, but we recommend the recent surveys Bartlett et al. [5], Belkin [6], Dar et al. [20] for further context.

Classically, by either operating in the underparameterized regime or by performing explicit regularization, we can force the training procedure to average out the harmful effects of training noise and thereby hope to obtain good generalization. The present cycle of seeking a deeper understanding began after it was observed that modern deep networks were overparameterized, capable of memorizing noise, and yet still generalized well, even when they were trained without explicit regularization [59, 84]. Experiments in Geiger et al. [28], Belkin et al. [8] observed a double-descent behavior of the generalization error where in addition to the traditional U-shaped curve in the underparameterized regime, the error decreases in the overparameterized regime as we increase the number of model parameters. This double descent phenomenon is not unique to deep learning models and was replicated for kernel learning [7]. Further, the good generalization performance in the overparameterized regime cannot be explained by traditional worst-case generalization bounds based on Rademacher complexity or VC-dimension since the models have the capacity to fit purely random labels. Overparameterized models must therefore have some fortuitous combination of the model architecture with the training algorithm that leads us to a particular solution that generalizes well.

To understand the phenomenon better, several works study the simpler setting of overparameterized linear regression. The minimum-$\ell_2$ norm[4] interpolator is of particular interest since gradient descent on the squared loss has an implicit[5] bias towards this solution in the overparameterized regime [24] and has been studied extensively. (An incomplete list is Hastie et al. [31], Mei and Montanari [52], Bartlett et al. [4], Belkin et al. [9], Muthukumar et al. [55], Bibas et al. [10], Kobak et al. [41], Wu and Xu [81], Richards et al. [63].) To generalize well, the underlying feature family must satisfy a balance between having a few important directions that sufficiently favor the true pattern, and a large number of unimportant directions that can absorb the noise in a harmless manner.

### 3.1 High dimensional binary classification

Both concurrently with and subsequent to the wave of analyses on overparameterized regression, researchers turned their attention to binary classification. A line of work poses the overparameterized binary classification problem as an optimization problem and analyzes it directly to obtain precise asymptotic behaviours of the generalization error [22, 66, 37, 69, 54, 38, 70]. The key technical tool employed in these works is the Convex Gaussian Min-max Theorem and the resultant error formulas involve solutions to a system of non-linear equations that typically do not admit closed-form expressions. The generalization error of the max-margin SVM has also been analyzed directly by studying the iterates of gradient descent in [13] and leveraging the implicit regularization perspective of optimization algorithms.

However, although the above works did significantly enhance our understanding of binary classification in the overparameterized regime, a fundamental question was not answered: "Is classification easier than regression?" While the classification task is easier than the regression task at test time (regression requires us to correctly predict a real value while binary classification requires us to only predict its sign correctly), the training data for classification is less informative than that for regression

---

[4]The minimum-$\ell_1$ norm interpolator has also been studied in Muthukumar et al. [55], Mitra [53], Li and Wei [50], Wang et al. [75] and while sparsity-seeking behavior helps preserve the true signal (if the true pattern indeed depends only on a few features), it poses a challenge for the harmless absorption of noise since the desired averaging behaviour is not achieved fully [55].

[5]In fact, there is an important complementary literature that brings out the implicit regularization performed by training methods, especially variants of gradient descent and stochastic gradient descent, and how the underlying architecture of the model shapes this implicit regularization [30, 68, 36, 80, 57, 2, 82].

since the labels are also binary. As described earlier, this question was answered in Muthukumar et al. [56], by exhibiting an asymptotic regime where binary classification error goes to zero, but the regression error does not. This was shown using Gaussian features with a bi-level covariance model. It turns out that the level of anisotropy (favoring of true features) required to perform regression correctly is significantly higher than that required for binary classification.

The key to the result in Muthukumar et al. [56] was the signal-processing inspired survival/contamination framework introduced in Muthukumar et al. [55] as a reconceptualization of the "effective ranks" perspective of Bartlett et al. [4]. For binary classification to succeed, what matters is that the survival exceed the contamination so that the sign of the prediction remains correct. Meanwhile, regression is harder since for regression to succeed, the survival must also tend to $1$.

### 3.2 Multiclass classification and the role of training loss function

There is a large classical body of work on multiclass classification algorithms [79, 12, 23, 18, 46], with further works giving computationally efficient algorithms for extreme multiclass problems with a huge number of classes [15, 83, 62]. Numerous theoretical works investigate the consistency of classifiers [85, 60, 61, 71, 14]. Finite-sample analysis of the generalization error in multiclass classification problems in the underparameterized regime has been studied in Koltchinskii and Panchenko [42], Guermeur [29], Allwein et al. [1], Li et al. [49], Cortes et al. [16], Lei et al. [47], Maurer [51], Lei et al. [48], Kuznetsov et al. [44, 45] and includes both data dependent bounds using Rademacher complexity, Gaussian complexity and covering numbers as well as data-independent bounds using the VC dimension. Recent work [72] leverages the Convex Gaussian Min-max Theorem to precisely characterize the asymptotic behaviour of the least-squares classifier in underparameterized multiclass classification.

So, how different is multiclass classification from binary classification? The test time task is more difficult and for the same total number of training points, we have fewer positive training examples from each class. Several empirical studies comparing the performances of multiclass classification via learning multiple binary classifiers have been undertaken [64, 25, 1]. The effects of the loss function while using deep nets to perform classification has also been investigated [32, 26, 43, 11, 21, 40, 35, 39]. Empirical evidence of least-squares minimization yielding competitive test classification performance to cross-entropy minimization has been presented in Rifkin and Klautau [64], Hui and Belkin [35], Bosman et al. [11].

More recently, Wang et al. [77] makes progress towards bridging the gap between empirical observations and theoretical understanding by proving that in certain overparameterized regimes the solution to a multiclass SVM problem is identical to the one obtained by minimum-norm interpolation of one-hot encoded labels (equivalently, that gradient descent on squared loss leads to the same solution as gradient descent on cross-entropy loss as a result of implicit bias of these algorithms [24, 36, 68]). In addition, Wang et al. [77] extends the analysis presented in Muthukumar et al. [56] for the binary classification problem to the multiclass problem with finitely many classes via an interesting reduction to analyzing a finite set of pairwise competitions, all of which must be won for multiclass classification to succeed. (We give further comments on the relationship of the present paper with Wang et al. [77] in Appendix H of the Supplemental material.)

## 4 Problem setup

We consider the multiclass classification problem with $k$ classes. The training data consists of $n$ pairs $\{\mathbf{x}_i, \ell_i\}_{i=1}^n$ where $x_i \in \mathbb{R}^d$ are i.i.d Gaussian vectors drawn from distribution,

$$\mathbf{x}_i \sim \mathcal{N}(0, I_d). \tag{1}$$

We make the following assumption on how the labels $\ell_i \in [k]$ are generated.

**Assumption 4.1.** *Orthogonal classes noiseless model*[6] *The class labels $\ell_i$ are generated based on which of the first $k$ dimensions of a point $\mathbf{x}_i$ has the largest value,*

$$\ell_i = \operatorname*{argmax}_{m \in [k]} \mathbf{x}_i[m]. \tag{2}$$

---

[6]A more generic model is $\ell_i = \operatorname{argmax}_{m \in [k]} \boldsymbol{\mu}_m^\top \mathbf{x}_i$ where the $\boldsymbol{\mu}_m$ are unit norm orthogonal vectors. If we further assume the bi-level model(Definition 4.2) and that the vectors $\boldsymbol{\mu}_m$ have no support outside of the favored features then it suffices to consider the simplified setting where $\boldsymbol{\mu}_m$ are 1-sparse unit vectors like we do here, due to the indifference of minimum norm interpolation to orthogonal transformations.

We use the notation $x_i[m]$ to refer to the $m^{th}$ element of vector $\mathbf{x}_i$. For clarity of exposition, we make explicit a feature weighting that transforms the training points as follows:

$$x_i^w[j] = \sqrt{\lambda_j} x_i[j] \quad \forall j \in [d]. \tag{3}$$

Here $\boldsymbol{\lambda} \in \mathbb{R}^d$ contains the squared feature weights. The feature weighting serves the role of favoring the true pattern, something that is essential for good generalization.[7]

The weighted feature matrix $\mathbf{X}^w \in \mathbb{R}^{n \times d}$ is given by,

$$\mathbf{X}^w = \begin{bmatrix} \mathbf{x}_1^w & \dots & \mathbf{x}_j^w & \dots & \mathbf{x}_n^w \end{bmatrix}^\top = \begin{bmatrix} \sqrt{\lambda_1}\mathbf{z}_1 & \dots & \sqrt{\lambda_j}\mathbf{z}_j & \dots & \sqrt{\lambda_d}\mathbf{z}_d \end{bmatrix}, \tag{4}$$

where $\mathbf{z}_j \in \mathbb{R}^n$ contains the $j^{th}$ features from the $n$ training points. Note that $\mathbf{z}_j \sim \mathcal{N}(0, I_n)$ are i.i.d Gaussians. We use a one-hot encoding for representing the labels as the matrix $\mathbf{Y}^{oh} \in \mathbb{R}^{n \times k}$,

$$\mathbf{Y}^{oh} = \begin{bmatrix} \mathbf{y}_1^{oh} & \dots & \mathbf{y}_m^{oh} & \dots & \mathbf{y}_k^{oh} \end{bmatrix}, \tag{5}$$

where,

$$y_m^{oh}[i] = \begin{cases} 1, & \text{if } \ell_i = m \\ 0, & \text{otherwise} \end{cases}. \tag{6}$$

A zero-mean variant of the encoding where we subtract the mean $\frac{1}{k}$ from each entry is denoted:

$$\mathbf{y}_m = \mathbf{y}_m^{oh} - \frac{1}{k}\mathbf{1}. \tag{7}$$

Our classifier consists of $k$ coefficient vectors $\hat{\mathbf{f}}_m$ for $m \in [k]$ that are learned by minimum-norm interpolation of the zero-mean one-hot variants using the weighted features.[8]

$$\hat{\mathbf{f}}_m = \arg\min_{\mathbf{f}} \|\mathbf{f}\|_2 \tag{8}$$

$$\text{s.t. } \mathbf{X}^w\mathbf{f} = \mathbf{y}_m^{oh} - \frac{1}{k}\mathbf{1}. \tag{9}$$

We can express these coefficients in closed form as,

$$\hat{\mathbf{f}}_m = (\mathbf{X}^w)^\top \left( \mathbf{X}^w (\mathbf{X}^w)^\top \right)^{-1} \mathbf{y}_m. \tag{10}$$

On a test point $\mathbf{x}_{test} \sim \mathcal{N}(0, I_d)$ we predict a label as follows: First, we transform the test point into the weighted feature space to obtain $\mathbf{x}_{test}^w$ where $x_{test}^w[j] = \sqrt{\lambda_j} x_{test}[j]$ for $j \in [d]$. Then we compute $k$ scalar "scores" and assign the class based on the largest score as follows:

$$\hat{\ell} = \underset{1 \le m \le k}{\arg\max} \hat{\mathbf{f}}_m^\top \mathbf{x}_{test}^w. \tag{11}$$

The true label of the test point is $\ell_{test} = \arg\max_{1 \le m \le k} x_{test}[m]$. A misclassification event $\mathcal{E}_{err}$ occurs iff

$$\underset{1 \le m \le k}{\arg\max} \, x_{test}[m] \ne \underset{1 \le m \le k}{\arg\max} \hat{\mathbf{f}}_m^\top \mathbf{x}_{test}^w. \tag{12}$$

In our work we determine sufficient conditions under which the probability of misclassification (computed over the randomness in both the training data and test point) goes to zero in an asymptotic regime where the number of training points, number of features, number of classes and feature weights scale according to the bi-level ensemble model.

---

[7]Our weighted feature model is equivalent to the one used in other works (e.g. [56]) that assume that the covariates come from an anisotropic Gaussian with a covariance matrix that favors the truly important directions.

[8]The classifier learned via this method is equivalent to those obtained by other natural training methods under sufficient overparameterization [77].

**Definition 4.2.** *(**Bi-level ensemble**): The bi-level ensemble is parameterized by $p, q, r$ and $t$ where $p > 1$, $0 \le r < 1$, $0 < q < (p - r)$ and $0 \le t < r$. Here, parameter $p$ controls the extent of overparameterization, $r$ determines the number of favored features, $q$ controls the weights on favored features and $t$ controls the number of classes. The number of features $(d)$, number of favored features $(s)$, number of classes $(k)$ and feature weights $(\sqrt{\lambda_j})$ all scale with the number of training points $(n)$ as follows:*

$$d = \lfloor n^p \rfloor, s = \lfloor n^r \rfloor, a = n^{-q}, k = c_k \lfloor n^t \rfloor, \tag{13}$$

*where $c_k$ is a positive integer. The feature weights are given by,*

$$\sqrt{\lambda_j} = \begin{cases} \sqrt{\frac{ad}{s}}, & 1 \le j \le s \\ \sqrt{\frac{(1-a)d}{d-s}}, & \text{otherwise} \end{cases}. \tag{14}$$

We provide a visualization of the bi-level model in Figure 1.

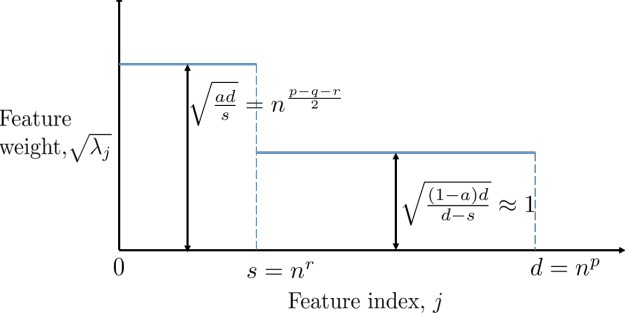

Figure 1: Bi-level feature weighting model. The first $s$ features have a higher weight and are favored during minimum-norm interpolation. These can be thought of as the square-roots of the eigenvalues of the feature covariance matrix in a Gaussian model for the covariates as in Bartlett et al. [4].

## 5 Main result

**Theorem 5.1.** *(**Asymptotic classification region in the bi-level model**): Under the bi-level ensemble model 4.2, when the true data generating process is 1-sparse (Assumption 4.1), the probability of misclassification $P(\mathcal{E}_{err}) \to 0$ as $n \to \infty$ if the following conditions hold:*

$$t < \min\left(r, 1 - r, p + 1 - 2(q + r), p - 2, 2q + r - 2\right) \tag{15}$$
$$q + r > 1. \tag{16}$$

Note that from Muthukumar et al. [56], the condition $q + r > 1$ corresponds to the regime where the corresponding regression does not generalize well and thus our result shows that multiclass classification can generalize in regimes where the corresponding regression problem does not. In this challenging regime, the empirical eigenstructure does not reveal the true nature of underlying features as illustrated in Appendix J.

Figure 2 visualizes the regimes by considering slices of the four dimensional scaling parameter space of $p, q, r$ and $t$. (1a) and (2a) fix the value of $q$ to $0.75$ and $0.95$ respectively and contrast the multiclass problem with a fixed finite number of classes ($t = 0$) to the binary classification and regression problems. From these plots we observe that if we fix $p, q, t$ and increase $r$, i.e. increasing how many features are favored (and thereby favoring each of them less), we transition from the regime where both regression and binary classification work, into the regime where binary classification works but regression does not, then the regime where this paper can prove multiclass classification works and finally to the regime where neither regression nor binary classification works.

In Figure 2, subplots (1b),(1c),(2b) and (2c) each visualize a slice along the $r$ and $t$ (class scaling) dimensions with fixed $p$ and $q$. The x axis itself in these plots corresponds to a fixed finite classes setting. From (1b) we observe that the right-hand boundary of the region where multiclass classification generalizes well contains two slopes. These slopes arise from the two conditions $t < 1 - r$ and

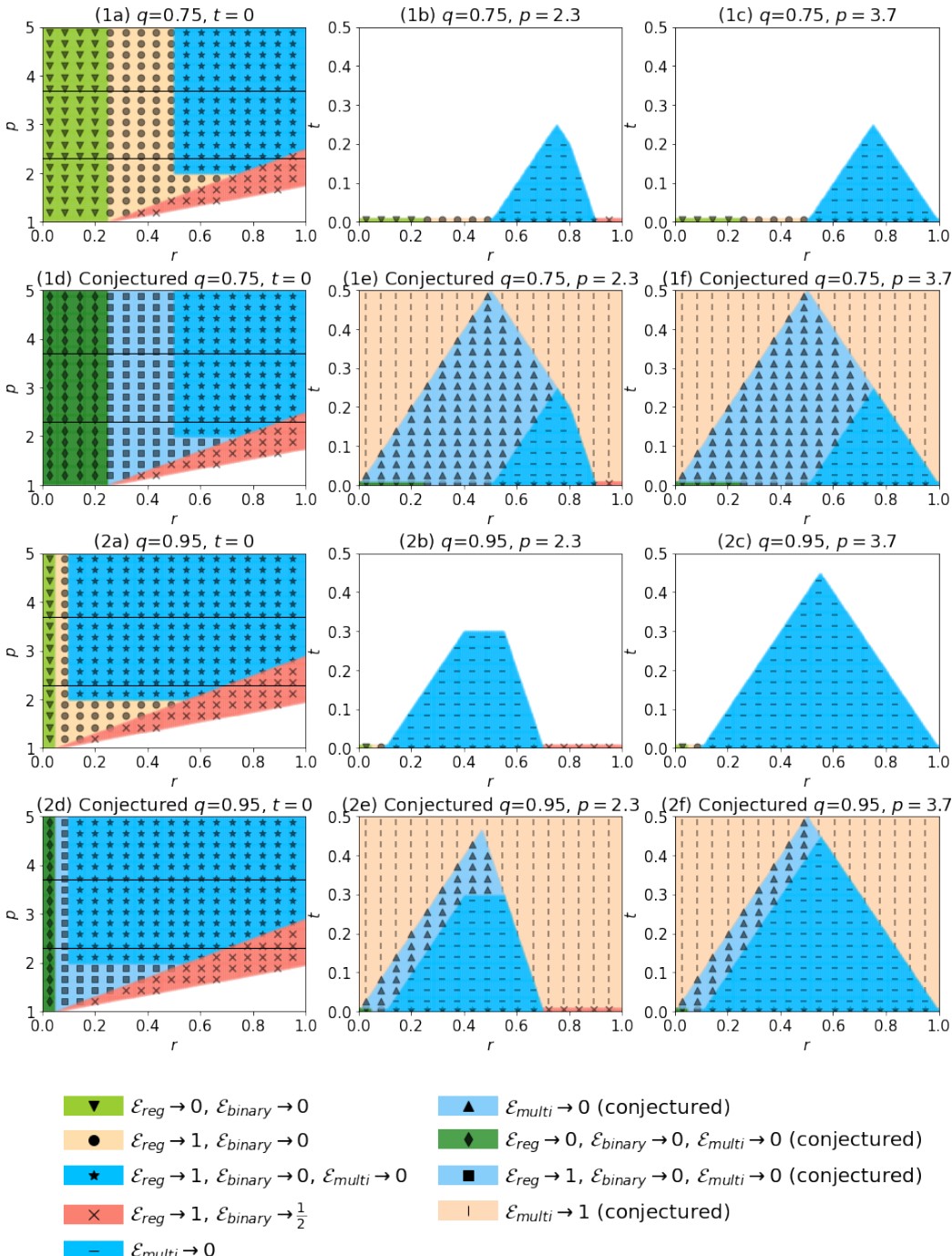

Figure 2: Visualization of the bi-level regimes in four dimensions $p, q, r, t$. (1a) and (2a) contrast multiclass classification with finite classes to binary classification and regression. The horizontal lines $p = 2.3$ and $p = 3.7$ correspond to the slices visualized in (1b), (1c), (2b) and (2c). The conjectured regimes are visualized in (1d), (1e), (1f), (2d), (2e) and (2f).

$t < p + 1 - 2(q + r)$ in Theorem 5.1 and are a result of either contamination from favored (but not true) features dominating or contamination from the unfavored features dominating. In (1c) we are

in the regime where binary classification works for all values of $r < 1$. However, as we increase $t$, eventually multiclass classification stops working.[9]

When we go from the binary problem to a multiclass problem with $k$ classes, the survival drops by a factor of $k$ as a consequence of having only $\frac{1}{k}$ fraction of positive training examples per class. This is because the one-hot labels we interpolate while training have fewer large values close to 1 that are able to positively correlate with the true feature vector. Having fewer positive exemplars also reduces the total energy in the training vector by a factor of $k$, and because of the square-root relationship of the standard deviation to the energy, the contamination only shrinks by a factor of $\sqrt{k}$. The overall survival/contamination ratio decreases by a factor of $\sqrt{k}$ making the multiclass classification task more difficult.[10] An interesting observation here is the amount of favoring required for good generalization is linked to the number of positive training examples per class. Indeed, if we consider a setting where the binary classification problem generalizes well, and we switch to the $k$ class multiclass problem, then by increasing the number of training samples $k$ fold (and thus matching the number of positive training examples per class in the multiclass case to the binary case) and keeping the number of features and feature weights constant we can generalize well for multiclass classification. (Appendix G of the Supplemental material elaborates on this phenomenon, as well as why it is somewhat surprising.)

Next, we present a brief overview of our proof that utilizes the survival/contamination analysis framework from Muthukumar et al. [56] along with a typicality-inspired argument where the feature margin (difference between largest and second largest feature) on the test point plays a key role. The complete proof is provided in Appendices B, C, D, and E of the Supplemental material.

## 5.1 Proof sketch

Assume without loss of generality that for the test point $\mathbf{x}_{test} \sim \mathcal{N}(0, I_d)$, the true class is $\alpha$ for some $\alpha \in [k]$. Let $\mathbf{x}_{test}^w$ be the weighted version of this test point. A necessary and sufficient condition for classification error is that for some $\beta \neq \alpha, \beta \in [k]$,

$$\widehat{f}_\alpha[\alpha]x_{test}^w[\alpha] + \widehat{f}_\alpha[\beta]x_{test}^w[\beta] + \sum_{j \notin \{\alpha,\beta\}} \widehat{f}_\alpha[j]x_{test}^w[j] < \widehat{f}_\beta[\alpha]x_{test}^w[\alpha]$$
$$+ \widehat{f}_\beta[\beta]x_{test}^w[\beta] + \sum_{j \notin \{\alpha,\beta\}} \widehat{f}_\beta[j]x_{test}^w[j]. \tag{17}$$

By converting into the unweighted feature space we obtain

$$\lambda_\alpha \widehat{h}_{\alpha,\beta}[\alpha]x_{test}[\alpha] - \lambda_\beta \widehat{h}_{\beta,\alpha}[\beta]x_{test}[\beta] < \sum_{j \notin \{\alpha,\beta\}} \lambda_j \widehat{h}_{\beta,\alpha}[j]x_{test}[j], \tag{18}$$

where

$$\widehat{h}_{\alpha,\beta}[j] = \lambda_j^{-1/2}(\hat{f}_\alpha[j] - \hat{f}_\beta[j]). \tag{19}$$

Performing some algebraic manipulations and because $\lambda_\alpha = \lambda_\beta = \lambda$ since both $\alpha$ and $\beta$ are favored features, we can rewrite this as

$$\frac{\lambda \widehat{h}_{\alpha,\beta}[\alpha]}{\mathsf{CN}_{\alpha,\beta}} \left( (x_{test}[\alpha] - x_{test}[\beta]) + x_{test}[\beta]\frac{\widehat{h}_{\alpha,\beta}[\alpha] - \widehat{h}_{\beta,\alpha}[\beta]}{\widehat{h}_{\alpha,\beta}[\alpha]} \right)$$
$$< \frac{1}{\mathsf{CN}_{\alpha,\beta}} \sum_{j \notin \{\alpha,\beta\}} \lambda_j \widehat{h}_{\beta,\alpha}[j]x_{test}[j], \quad (20)$$

---

[9]To be precise, what the region actually illustrates is that our proof approach stops being able to show that multiclass classification works. In the Conclusion section, we conjecture where we believe that multiclass classification actually stops working. The conjectured regions are illustrated in (1e),(1f),(2e) and (2f).

[10]This is also responsible for contamination due to favored features being able to cause errors. For binary classification, because the true feature survival is constant (depending only on the level of label noise), the survival can always asymptotically overcome any contamination from other favored features [56].

where

$$\mathsf{CN}_{\alpha,\beta} = \sqrt{\left(\sum_{j\notin\{\alpha,\beta\}} \lambda_j^2(\widehat{h}_{\beta,\alpha}[j])^2\right)}. \tag{21}$$

We divide by $\mathsf{CN}_{\alpha,\beta}$ to normalize the RHS above to have a standard normal distribution. Next, by removing the dependency on $\beta$, we obtain a sufficient condition for correct classification:

$$\underbrace{\frac{\min_\beta \lambda\widehat{h}_{\alpha,\beta}[\alpha]}{\max_\beta \mathsf{CN}_{\alpha,\beta}}}_{\text{SU/CN ratio}}\left(\underbrace{\min_\beta\left(x_{test}[\alpha]-x_{test}[\beta]\right)}_{\text{closest feature margin}} - \underbrace{\max_\beta |x_{test}[\beta]|}_{\text{largest competing feature}} \cdot \underbrace{\max_\beta\left|\frac{\widehat{h}_{\alpha,\beta}[\alpha]-\widehat{h}_{\beta,\alpha}[\beta]}{\widehat{h}_{\alpha,\beta}[\alpha]}\right|}_{\text{survival variation}}\right)$$

$$> \max_\beta \frac{1}{\mathsf{CN}_{\alpha,\beta}}\underbrace{\left(\sum_{j\notin\{\alpha,\beta\}} \lambda_j\widehat{h}_{\beta,\alpha}[j]x_{test}[j]\right)}_{\text{normalized contamination}}. \tag{22}$$

Here the min and max are over all competing features: $1 \leq \beta \leq k, \beta \neq \alpha$ and the sum is over all $d$ feature indices except $\alpha$ and $\beta$, but we simplify the notation for convenience. We show via intermediate lemmas introduced in Appendix B of the Supplemental material that under the conditions specified in Theorem 5.1, with sufficiently high probability[11], the relevant survival to contamination SU/CN ratio grows at a polynomial rate $n^v$ for some $v > 0$, the closest feature margin shrinks at a less-than-polynomial rate $1/\sqrt{\ln nk}$, the survival variation decays at a polynomial rate $n^{-u}$ for some $u > 0$. Further, the magnitudes of the largest competing feature and the normalized contamination are no more than $\sqrt{\ln(nk)}$.

This implies that the left-hand side of Equation (22) grows at a polynomial rate $n^v$ (ignoring logarithmic terms) and dominates the right-hand side which grows at the much slower rate $\sqrt{\ln nk}$. A survival/contamination ratio also plays a key role in the analysis of the binary classification problem in Muthukumar et al. [56] but in the multiclass setting, we additionally have the survival variation term and feature margin playing important roles since we are comparing different scores while predicting the class label. For correct classification, the survival/contamination ratio must be sufficiently large, the survival variation must be small enough and the feature margin must be sufficiently large.

## 6 Conclusion

In this work we compute sufficient conditions for good generalization of multiclass classification in a bi-level overparameterized linear model with Gaussian features. We observed that multiclass classification can generalize even when the regression problem does not generalize (for $q + r > 1$). Further, the multiclass problem is "harder" than the binary problem because we have fewer positive training examples per class. The nature of the training data complicates our analysis in the multiclass setting since the true class labels are generated by comparing $k$ features and thus we no longer have independence of the encoded class label $y$ with any of these features. This becomes relevant when we compute bounds on the survival and contamination quantities since the Hanson-Wright inequality [65] is no longer applicable directly on the quantities of interest as was the case for the binary classification problem in prior work [56]. As a consequence of working around this non-independence we believe that our sufficient conditions for good generalization in the regime $q + r > 1$ are loose.

Even though in our work we focus on the regime where regression does not work, $q + r > 1$, we can extend the analysis to the regime where $q + r < 1$ by grinding through the expressions for survival and contamination in this regime. Even in this regime, for multiclass training data, survival is of the order $\frac{1}{k}$ while contamination scales similarly to the regime $q + r > 1$. Thus, while it is true that for

---

[11]This is where we leverage the idea of typicality-style proofs in information theory [17] to avoid unnecessarily loose union bounds that end up being dominated by the atypical behavior of quantities. In our case, by pulling the feature margin out explicitly, we can just deal with its typical behavior. Similarly, the typical behavior of the largest competing feature and the true feature is all that matters.

binary classification or a fixed number of classes, the regime where regression works is a regime where classification also works, this need not be true if there are too many classes.

We conjecture that the following is a set of necessary and sufficient conditions for asymptotically good generalization (We elaborate on this in Appendix F in the Supplemental material):

**Conjecture 6.1.** (***Conjectured bi-level regions***): *Under the bi-level ensemble model 4.2, when the true data generating process is 1-sparse (Assumption 4.1), as $n \to \infty$, the probability of misclassification event $P(\mathcal{E}_{err})$ behaves as follows:*

$$P(\mathcal{E}_{err}) \to \begin{cases} 0, & \text{if } t < \min\left(r, 1-r, p+1-2 \cdot \max(1, q+r)\right) \\ 1, & \text{if } t > \min\left(r, 1-r, p+1-2 \cdot \max(1, q+r)\right) \end{cases}. \tag{23}$$

The conjectured regions are visualized in (1d),(1e),(1f),(2d),(2e) and (2f) in Figure 2. Subfigures (1d) and (2d) illustrate that we believe multiclass classification with finitely many classes works if binary classification works. Further, comparing (1e) to (2e) when we increase $q$, the conjectured parameter region where multiclass classification works shrinks since we decrease the amount of favoring of true features. Interestingly, the nature of the looseness in our approach is such that our proof technique is able to recover a larger fraction of the conjectured region for larger $q$ which intuitively is a result of less favoring leading to stronger concentration of certain random quantities. Tightening the potential looseness in our analysis and proving the converse result by computing sufficient conditions for poor generalization of multiclass classification are interesting avenues of future work.

Further, although the present analysis focuses on solutions that exactly interpolate the training data, we can extend our results to account for additional ridge regularization by viewing ridge regularization as minimum-norm interpolation using augmented contamination-free features as in the Appendix of Muthukumar et al. [55] and computing bounds leveraging tools from Tsigler and Bartlett [74]. Our assumption of the strict bi-level weighting model is largely to simplify the calculations and by substituting terms appropriately in our lemmas from Appendix B in the Supplemental material, it should be possible to compute results for other weighting models. Finally, exploring the new phenomena that can be encountered as we go beyond the 1-sparse noiseless model is an exciting direction for future work.

## Acknowledgments and Disclosure of Funding

We are grateful to our earlier collaborators Vidya Muthukumar, Misha Belkin, Daniel Hsu, and Adhyyan Narang. In addition, we want to thank the students and course staff for the Fall 2020 iteration of Berkeley's CS189/289A machine learning courses, where we had adapted ideas from Muthukumar et al. [55, 56] in teaching the foundations of modern machine learning — the need for the present paper became more clear during this process.

We gratefully acknowledge the support from ML4Wireless center member companies and NSF grants AST-2132700 and AST-2037852 for making this research possible.

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
