## A  Notation

We summarize the notation used in the problem setup (as well as some terms defined later) as follows:

## B  Proof of Theorem 5.1

We restate Theorem 5.1, our main result, here for convenience:

**Theorem 5.1.** *(**Asymptotic classification region in the bi-level model**):  Under the bi-level ensemble model 4.2, when the true data generating process is 1-sparse (Assumption 4.1), the probability of misclassification $P(\mathcal{E}_{err}) \to 0$ as $n \to \infty$ if the following conditions hold:*

$$t < \min\left(r, 1-r, p+1-2(q+r), p-2, 2q+r-2\right) \tag{15}$$
$$q + r > 1. \tag{16}$$

Our proof that utilizes the survival/contamination analysis framework from Muthukumar et al. [56] along with a typicality-inspired argument where the feature margin (difference between largest and second largest feature) on the test point plays a key role.

Assume without loss of generality that for the test point $\mathbf{x}_{test} \sim \mathcal{N}(0, I_d)$, the true class is $\alpha$ for some $\alpha \in [k]$. Let $\mathbf{x}_{test}^w$ be the weighted version of this test point. A necessary and sufficient condition for classification error is that for some $\beta \neq \alpha, \beta \in [k]$, the score associated with class $\beta$ is higher than the score associated with class $\alpha$. Pulling out the key terms associated with the $\alpha$ and $\beta$ weighted features, we get:

$$\widehat{f}_\alpha[\alpha]x_{test}^w[\alpha] + \widehat{f}_\alpha[\beta]x_{test}^w[\beta] + \sum_{j \notin \{\alpha,\beta\}} \widehat{f}_\alpha[j]x_{test}^w[j] < \widehat{f}_\beta[\alpha]x_{test}^w[\alpha]$$
$$+ \widehat{f}_\beta[\beta]x_{test}^w[\beta] + \sum_{j \notin \{\alpha,\beta\}} \widehat{f}_\beta[j]x_{test}^w[j] \tag{24}$$

$$\implies (\widehat{f}_\alpha[\alpha] - \widehat{f}_\beta[\alpha])x_{test}^w[\alpha] - (\widehat{f}_\beta[\beta] - \widehat{f}_\alpha[\beta])x_{test}^w[\beta] < \sum_{j \notin \{\alpha,\beta\}} (\widehat{f}_\beta[j] - \widehat{f}_\alpha[j])x_{test}^w[j]. \tag{25}$$

$$\tag{26}$$

Note that $\sum_{j \notin \alpha, \beta}$ refers to the sum over all feature indices 1 to $d$ excluding $\alpha$ and $\beta$.

Table 1: Notation

| Symbol | Definition | Dimension | Source |
|---|---|---|---|
| $k$ | Number of classes | Scalar | Sec. 4 |
| $n$ | Number of training points | Scalar | Sec. 4 |
| $d$ | Dimension of each point — the total number of features | Scalar | Sec. 4 |
| $s$ | The number of favored features | Scalar | Def. 4.2 |
| $p$ | Parameter controlling overparameterization ($d = n^p$) | Scalar | Def. 4.2 |
| $r$ | Parameter controlling the number of favored features ($s = n^r$) | Scalar | Def. 4.2 |
| $a$ | Parameter controlling the favored weights ($a = n^{-q}$) | Scalar | Def. 4.2 |
| $t$ | Parameter controlling the number of classes ($k = c_k n^t$) | Scalar | Def. 4.2 |
| $c_k$ | The number of classes when $t = 0$ ($k = c_k n^t$) | Scalar | Def. 4.2 |
| $\lambda_j$ | Squared weight of the $j$th feature | Scalar | Def. 4.2 |
| $\mathbf{x}_i$ | $i$th training point (unweighted) | Length-$n$ vector | Eqn. 1 |
| $\ell_i$ | Class label of $i$th training point | Scalar | Eqn. 2 |
| $\mathbf{x}_i^w$ | $i$th training point (weighted) | Length-$n$ vector | Eqn. 3 |
| $\mathbf{X}^w$ | Weighted feature matrix | $(n \times d)$-matrix | Eqn. 4 |
| $\mathbf{z}_j$ | The collected $j$th features of all training points | Length-$n$ vector | Eqn. 4 |
| $\mathbf{y}_m^{oh}$ | One-hot encoding of all the training points for label $m$ | Length-$n$ vector | Eqn. 6 |
| $\mathbf{Y}^{oh}$ | One-hot label matrix | $(n \times k)$-matrix | Eqn. 5 |
| $\mathbf{y}_m$ | Zero-mean encoding of the training points for label $m$ | Length-$n$ vector | Eqn. 7 |
| $\hat{\mathbf{f}}_m$ | Learned coefficients for label $m$ using min-norm interpolation | Length-$d$ vector | Eqn. 10 |
| $\mathbf{x}_{test}$ | A single test point | Length-$d$ vector | Sec. 4 |
| $\mathbf{x}_{test}^w$ | A single weighted test point | Length-$d$ vector | Sec. 4 |
| $\mathbf{A}$ | $\mathbf{A} = \mathbf{X}^w (\mathbf{X}^w)^\top$ | $(n \times n)$-matrix | Eqn. 38 |
| $\mu_i(\mathbf{A})$ | The $i$th eigenvalue of matrix $\mathbf{A}$, sorted in descending order | Scalar | App. B |
| $\mathbf{\Lambda}$ | Matrix of squared feature weights: $\mathrm{diag}(\lambda_1, \lambda_2, \ldots, \lambda_d)$ | $(d \times d)$-matrix | App. B |
| $\widehat{h}_{\alpha,\beta}$ | Relative survival $\widehat{h}_{\alpha,\beta}[j] = \lambda_j^{-1/2}(\hat{f}_\alpha[j] - \hat{f}_\beta[j])$ | Length-$d$ vector | Eqn. 28 |
| $\mathsf{CN}_{\alpha,\beta}$ | Normalizing factor $\mathsf{CN}_{\alpha,\beta} = \sqrt{\left(\sum_{j \notin \{\alpha,\beta\}} \lambda_j^2 (\widehat{h}_{\beta,\alpha}[j])^2\right)}$ | Scalar | Eqn. 34 |
| $\|\cdot\|_{\psi_2}$ | The sub-Gaussian norm of a scalar random variable | Scalar | Eqn. 74 |
| $\bar{\mu}$ | Center of the eigenvalue bounds for $\mathbf{A}^{-1}$, $\bar{\mu} = \frac{1}{\sum_j \lambda_j}$ | Scalar | Eqn. 41 |
| $\Diamond$ | Deviation term in eigenvalue bounds for $\mathbf{A}$ | Scalar | Eqn. 77 |
| $\Delta_\mu$ | Deviation term in eigenvalue bounds for $\mathbf{A}^{-1}$ | Scalar | Eqn. 43 |

By converting into the unweighted feature space we obtain,

$$\lambda_\alpha \widehat{h}_{\alpha,\beta}[\alpha] x_{test}[\alpha] - \lambda_\beta \widehat{h}_{\beta,\alpha}[\beta] x_{test}[\beta] < \sum_{j \notin \{\alpha,\beta\}} \lambda_j \widehat{h}_{\beta,\alpha}[j] x_{test}[j], \tag{27}$$

where we introduce the short-hand notation,

$$\widehat{h}_{\alpha,\beta}[j] = \lambda_j^{-1/2}(\hat{f}_\alpha[j] - \hat{f}_\beta[j]) \tag{28}$$

$$\widehat{h}_{\beta,\alpha}[j] = \lambda_j^{-1/2}(\hat{f}_\beta[j] - \hat{f}_\alpha[j]). \tag{29}$$

Since both $\alpha$ and $\beta$ are favored feature indices, by leveraging the definition of the bi-level model and denoting $\lambda_\alpha = \lambda_\beta = \lambda$, we get

$$\lambda \left( \widehat{h}_{\alpha,\beta}[\alpha] x_{test}[\alpha] - \widehat{h}_{\beta,\alpha}[\beta] x_{test}[\beta] \right) < \sum_{j \notin \{\alpha,\beta\}} \lambda_j \widehat{h}_{\beta,\alpha}[j] x_{test}[j]. \tag{30}$$

Next, we perform some algebraic manipulations,

$$\lambda\left(\widehat{h}_{\alpha,\beta}[\alpha]x_{test}[\alpha] - \widehat{h}_{\beta,\alpha}[\beta]x_{test}[\beta]\right) < \sum_{j\notin\{\alpha,\beta\}} \lambda_j\widehat{h}_{\beta,\alpha}[j]x_{test}[j] \tag{31}$$

$$\implies \lambda\widehat{h}_{\alpha,\beta}[\alpha](x_{test}[\alpha] - x_{test}[\beta]) + \lambda x_{test}[\beta](\widehat{h}_{\alpha,\beta}[\alpha] - \widehat{h}_{\beta,\alpha}[\beta]) < \sum_{j\notin\{\alpha,\beta\}} \lambda_j\widehat{h}_{\beta,\alpha}[j]x_{test}[j] \tag{32}$$

$$\implies \lambda\widehat{h}_{\alpha,\beta}[\alpha]\left((x_{test}[\alpha] - x_{test}[\beta]) + x_{test}[\beta]\frac{\widehat{h}_{\alpha,\beta}[\alpha] - \widehat{h}_{\beta,\alpha}[\beta]}{\widehat{h}_{\alpha,\beta}[\alpha]}\right) < \sum_{j\notin\{\alpha,\beta\}} \lambda_j\widehat{h}_{\beta,\alpha}[j]x_{test}[j]. \tag{33}$$

We divide both sides by the quantity $\mathsf{CN}_{\alpha,\beta}$ defined as,

$$\mathsf{CN}_{\alpha,\beta} = \sqrt{\left(\sum_{j\notin\{\alpha,\beta\}} \lambda_j^2(\widehat{h}_{\beta,\alpha}[j])^2\right)}. \tag{34}$$

This normalizes the RHS of (33) to have a standard normal distribution. Thus, the necessary and sufficient condition for a misclassification error is for some $\beta \neq \alpha, \beta \in [k]$,

$$\frac{\lambda\widehat{h}_{\alpha,\beta}[\alpha]}{\mathsf{CN}_{\alpha,\beta}}\left((x_{test}[\alpha] - x_{test}[\beta]) + x_{test}[\beta]\frac{\widehat{h}_{\alpha,\beta}[\alpha] - \widehat{h}_{\beta,\alpha}[\beta]}{\widehat{h}_{\alpha,\beta}[\alpha]}\right) < \frac{1}{\mathsf{CN}_{\alpha,\beta}}\sum_{j\notin\{\alpha,\beta\}} \lambda_j\widehat{h}_{\beta,\alpha}[j]x_{test}[j]. \tag{35}$$

A sufficient condition for correct classification can then be obtained by ensuring that the smallest potential value of the LHS is still greater than the value of the RHS for all values of $\beta$. Thus, we obtain a sufficient condition for correct classification by appropriately minimizing or maximizing quantities over competing feature indices $\beta \neq \alpha, \beta \in [k]$ (for notational convenience we simply denote this as $\min_\beta$ or $\max_\beta$).

$$\underbrace{\frac{\min_\beta \lambda\widehat{h}_{\alpha,\beta}[\alpha]}{\max_\beta \mathsf{CN}_{\alpha,\beta}}}_{\text{SU/CN ratio}}\left(\underbrace{\min_\beta (x_{test}[\alpha] - x_{test}[\beta])}_{\text{closest feature margin}} - \underbrace{\max_\beta |x_{test}[\beta]|}_{\text{largest competing feature}} \cdot \underbrace{\max_\beta \left|\frac{\widehat{h}_{\alpha,\beta}[\alpha] - \widehat{h}_{\beta,\alpha}[\beta]}{\widehat{h}_{\alpha,\beta}[\alpha]}\right|}_{\text{survival variation}}\right)$$

$$> \underbrace{\max_\beta \frac{1}{\mathsf{CN}_{\alpha,\beta}}\left(\sum_{j\notin\{\alpha,\beta\}} \lambda_j\widehat{h}_{\beta,\alpha}[j]x_{test}[j]\right)}_{\text{normalized contamination}}. \tag{36}$$

We will show that under the conditions specified in Theorem 5.1, with sufficiently high probability, the relevant survival to contamination SU/CN ratio grows at a polynomial rate $n^v$ for some $v > 0$, the closest feature margin shrinks at a less-than-polynomial rate $1/\sqrt{\ln nk}$, and the survival variation decays at a polynomial rate $n^{-u}$ for some $u > 0$. Further, the magnitudes of the largest competing feature and the normalized contamination are no more than $2\sqrt{\ln(nk)}$. Here, we leverage the idea of typicality-style proofs in information theory [17] to avoid unnecessarily loose union bounds that end up being dominated by the atypical behavior of quantities. In our case, by pulling the feature margin out explicitly, we can just deal with its typical behavior. Similarly, the typical behavior of the largest competing feature and the true feature is all that matters. Before we proceed with the rest of our proof we remind the reader of a few important definitions.

Recall from (10) that our learned feature coefficients are

$$\hat{\mathbf{f}}_m = (\mathbf{X}^w)^\top \left(\mathbf{X}^w(\mathbf{X}^w)^\top\right)^{-1} \mathbf{y}_m. \tag{37}$$

Let

$$\mathbf{A} = \mathbf{X}^w (\mathbf{X}^w)^\top. \tag{38}$$

Then we can express our learned coefficients as

$$\hat{f}_m[j] = \sqrt{\lambda_j} \mathbf{z}_j^\top \mathbf{A}^{-1} \mathbf{y}_m, \tag{39}$$

where $\mathbf{z}_j \in \mathbb{R}^n$ contains the $j^{th}$ features of all $n$ training points. The rows of $\mathbf{X}^w$ are i.i.d. Gaussians with covariance matrix $\mathbf{\Lambda} = \mathrm{diag}(\lambda_1, \lambda_2, \ldots, \lambda_d)$. Let $\mu_1(\mathbf{A})$ denote the largest eigenvalue and $\mu_n(\mathbf{A})$ denote the smallest eigenvalue of $\mathbf{A}$ respectively, with $\mu_i(\mathbf{A})$ being the $i$-th largest eigenvalue of $\mathbf{A}$.

Next, we state a useful lemma adapted from Bartlett et al. [4] [12] that bounds the eigenvalues of $\mathbf{A}^{-1}$. Subsequent lemmas will utilize these eigenvalue bounds.

**Lemma B.1.** *(Eigenvalue bounds on $\mathbf{A}^{-1}$ adapted from Bartlett et al. [4]):*
*If $\mathbf{\Lambda}$ is such that $\Diamond \ll \sum_j \lambda_j$, then with probability at least $(1 - 2e^{-n})$,*

$$\bar{\mu} - \Delta_\mu \leq \mu_n(\mathbf{A}^{-1}) \leq \mu_1(\mathbf{A}^{-1}) \leq \bar{\mu} + \Delta_\mu, \tag{40}$$

*where,*

$$\bar{\mu} = \frac{1}{\sum_j \lambda_j} \tag{41}$$

$$\Diamond = \frac{32}{9} \left( \lambda_1 (1 + \ln 9) n + \sqrt{(1 + \ln 9) n \sum_j \lambda_j^2} \right) \tag{42}$$

$$\Delta_\mu = \bar{\mu} \left( \frac{\Diamond}{\sum_j \lambda_j} + \Theta \left( \frac{\Diamond}{\sum_j \lambda_j} \right)^2 \right). \tag{43}$$

*Further this implies that with probability at least $(1 - 2e^{-n})$,*

$$\left| \mu_i(\mathbf{A}^{-1} - \bar{\mu} \mathbf{I}_n) \right| \leq \Delta_\mu \tag{44}$$

*for all $i \in [n]$.*

The subsequent lemmas bound the feature margin, survival, contamination and survival variation terms, utilizing tools from [4] and building on results from [56].

**Lemma B.2.** *(Lower bound on the closest feature margin as $k \to \infty$): For any constant $\varepsilon > 0$, there exists a constant $\theta$ such that, for sufficiently large $k$ with probability at least $(1 - \varepsilon)$,*

$$\min_{\beta: 1 \leq \beta \neq \alpha \leq k} (x_{test}[\alpha] - x_{test}[\beta]) \geq \frac{\theta}{\sqrt{2 \ln(k)}}. \tag{45}$$

*Here, $\alpha$ is fixed and corresponds to the index of the true class — i.e. $\alpha$ corresponds to the index of the maximum feature among the first $k$ features.*

**Lemma B.3.** *(Lower bound on the closest feature margin when $k$ is constant): If $k = c_k$ for some fixed constant $c_k$, for any constant $\varepsilon > 0$, there exists a constant $\varepsilon' > 0$ such that*

$$\Pr \left( \min_{\beta, \gamma: 1 \leq \beta \neq \gamma \leq c_k} |x_{test}[\beta] - x_{test}[\gamma]| \geq \varepsilon' \right) \geq 1 - \varepsilon. \tag{46}$$

*Thus, with probability at least $(1 - \varepsilon)$,*

$$\min_{\beta: 1 \leq \beta \neq \alpha \leq k} (x_{test}[\alpha] - x_{test}[\beta]) \geq \varepsilon'. \tag{47}$$

*Here, $\alpha$ is fixed and corresponds to the index of the true class — i.e. $\alpha$ corresponds to the index of the maximum feature among the first $k$ features.*

---

[12]More precisely this lemma appeared in the first version of this work at `https://arxiv.org/pdf/1906.11300v1.pdf`. In subsequent versions the authors use a slightly weaker version of this result since it is sufficient for their purposes.

**Lemma B.4.** *(**Lower bound on relative survival of true feature**): For any fixed $\beta \in [k]$, $\beta \neq \alpha$, with $\lambda_\alpha = \lambda_\beta = \lambda$ we have with probability at least $(1 - 5/(nk))$,*

$$\lambda \widehat{h}_{\alpha,\beta}[\alpha] \geq \lambda \left( c_{10}\bar{\mu}\frac{n}{k}\sqrt{\ln(k)} - c_9(\bar{\mu}\sqrt{n}\sqrt{\ln(nk)} + \Delta_\mu \cdot n/\sqrt{k}) \right), \tag{48}$$

*for universal positive constants $c_9$ and $c_{10}$.*

By substituting the asymptotic behavior of parameters from our bi-level ensemble model we get the following corollary:

**Corollary B.4.1.** *Under the bi-level ensemble model 4.2, for any fixed $\beta \in [k]$, $\beta \neq \alpha$, $\lambda_\alpha = \lambda_\beta = \lambda$ if $t < 1/2$, $t < 2(q + r - 1)$ and $1 < q + r < (p+1)/2$, with probability at least $(1 - 5/(nk))$,*

$$\lambda \widehat{h}_{\alpha,\beta}[\alpha] \geq c_{12}n^{1-q-r-t}\sqrt{\ln(k)}, \tag{49}$$

*for universal positive constant $c_{12}$.*

**Lemma B.5.** *(**Upper bound on contamination**): For any fixed $\beta \in [k]$, $\beta \neq \alpha$, with probability at least $(1 - 7/(nk))$,*

$$\mathsf{CN}_{\alpha,\beta} \leq c_7(\bar{\mu}\sqrt{\frac{n}{k}} \cdot \sqrt{\ln(ndk)} + \Delta_\mu \cdot n/\sqrt{k})\sqrt{\sum \lambda_j^2}, \tag{50}$$

*for universal positive constant $c_7$.*

As before, for our bi-level ensemble model we have the corollary:

**Corollary B.5.1.** *Under the bi-level model 4.2, in the regime $1 < q + r < (p+1)/2$, with probability at least $(1 - 7/(nk))$,*

$$\mathsf{CN}_{\alpha,\beta} \leq c_{13}n^{(1-t-p)/2+\max(0,3/2-q-r)+\max(0,p/2-q-r/2)}\sqrt{\ln(ndk)}, \tag{51}$$

*for universal positive constant $c_{13}$.*

**Lemma B.6.** *(**Upper bound on survival variance**): For any fixed competing feature $\beta \in [k]$, $\beta \neq \alpha$ with $\lambda_\alpha = \lambda_\beta$, we have with probability at least $(1 - 15/(nk))$,*

$$\frac{\widehat{h}_{\alpha,\beta}[\alpha] - \widehat{h}_{\beta,\alpha}[\beta]}{\widehat{h}_{\alpha,\beta}[\alpha]} \leq \frac{2c_9(\bar{\mu}\sqrt{n}\sqrt{\ln(nk)} + \Delta_\mu \cdot n/\sqrt{k})}{c_{10}\bar{\mu}\frac{n}{k}\sqrt{\ln(k)} - c_9(\bar{\mu}\sqrt{n}\sqrt{\ln(nk)} + \Delta_\mu \cdot n/\sqrt{k})}, \tag{52}$$

*for universal positive constants $c_9$ and $c_{10}$.*

As before, we can also obtain the asymptotic bound:

**Corollary B.6.1.** *Under the bi-level ensemble model 4.2, for any fixed $\beta \in [k]$, $\beta \neq \alpha$, if $t < 1/2$, $t < 2(q + r - 1)$, and $1 < q + r < (p+1)/2$, with probability at least $(1 - 15/(nk))$,*

$$\frac{\widehat{h}_{\alpha,\beta}[\alpha] - \widehat{h}_{\beta,\alpha}[\beta]}{\widehat{h}_{\alpha,\beta}[\alpha]} < n^{-u}, \tag{53}$$

*for large enough $n$ for some fixed $u > 0$.*

Next, we assume that the lemmas and corollaries stated above are true and complete the proof for Theorem 5.1. We provide proofs for these lemmas in Appendices C, D and E.

Assume we are in the regime where $t < 1/2$, $t < 2(q + r - 1)$, and $1 < q + r < (p+1)/2$, so all our corollaries above hold. Denote the misclassification event as $\mathcal{E}_{err}$ and let $\varepsilon > 0$ be an arbitrarily chosen constant.

Substitute Corollaries B.4.1, B.5.1, and B.6.1 into (22), applying them on all $1 \leq \beta \neq \alpha \leq k$. They hold with probability at least $1 - 5/(nk)$, $1 - 7/(nk)$, and $1 - 15/(nk)$ respectively for a given test point and choice of $\beta$. So by the union bound across the three bounds and all $k - 1$ choices of $\beta$, with probability at most $27/n$, one of these corollaries will not hold for our test point for some $\beta$. Let this failure event be denoted $E_1$.

In the case when $E_1$ does not occur, misclassification occurs only if

$$\frac{c_{12}\sqrt{\ln(k)}}{c_7\sqrt{\ln(ndk)}}n^v \left( \min_\beta (x_{test}[\alpha] - x_{test}[\beta]) - \max_\beta |x_{test}[\beta]| \cdot n^{-u} \right) < \max_\beta Z^{(\beta)}, \tag{54}$$

where we define the exponent

$$v = 1 - q - r - t - (1 - t - p)/2 - \max\left(0, \frac{3}{2} - q - r\right) - \max\left(0, \frac{p}{2} - q - \frac{q}{2}\right) \tag{55}$$

$$= \frac{p+1}{2} - q - r - \frac{t}{2} - \max\left(0, \frac{3}{2} - q - r, \frac{p}{2} - q - \frac{r}{2}, \frac{3}{2} - 2q - \frac{3r}{2}\right), \tag{56}$$

and

$$Z^{(\beta)} = \frac{1}{\mathsf{CN}_{\alpha,\beta}} \left( \sum_{j \notin \{\alpha,\beta\}} \lambda_j \widehat{h}_{\beta,\alpha}[j] x_{test}[j] \right). \tag{57}$$

For each class $\beta$, observe that we have $Z^{(\beta)} \sim \mathcal{N}(0,1)$.[13] Thus, by the Gaussian tail bound, for each $\beta$ with probability at least $(1 - 1/(nk))$,

$$Z^{(\beta)} < \sqrt{2\ln(nk)}. \tag{58}$$

So by the union bound over all $k$ classes $\beta$, with probability at least $(1 - 1/n)$,

$$\max_{\beta} Z^{(\beta)} < \sqrt{2\ln(nk)}. \tag{59}$$

Let the failure event where this is not the case be $E_2$.

An identical argument shows that with probability at least $(1 - 2/n)$, $\max_{\beta} |x_{test}[\beta]| \leq \sqrt{2\ln(nk)}$. Let $E_3$ be the failure event where this is not the case.

From Lemma B.2, we know with probability $1 - \varepsilon$ that, if $t > 0$, then for sufficiently large $n$ (and so sufficiently large $k$)

$$\min_{\beta} (x_{test}[\alpha] - x_{test}[\beta]) > \frac{\theta}{\sqrt{2\ln(k)}}. \tag{60}$$

If $t = 0$ and $k = c_k$, then Lemma B.3 states that, with probability $1 - \varepsilon$,

$$\Pr\left( \min_{1 \leq \beta \neq \gamma \leq c_k} |x_{test}[\beta] - x_{test}[\gamma]| \geq \varepsilon' \right) \geq 1 - \varepsilon, \tag{61}$$

for some constant $\varepsilon'$. Let the $\varepsilon$-probability event of the appropriate margin bound (depending on whether $t = 0$ or $t > 0$) being violated be the error event $E_4$.

Assuming $E_1$, $E_2$, $E_3$, and $E_4$ all do not take place, misclassification can only occur if

$$\frac{c_{12}\sqrt{\ln(k)}}{c_7\sqrt{\ln(ndk)}} n^v \left( \min\left(1 - \varepsilon, \frac{\theta}{\sqrt{2\ln(k)}}\right) - \sqrt{2\ln(nk)}n^{-u} \right) < \sqrt{2\ln(nk)}. \tag{62}$$

Clearly, if $v > 0$, then (for sufficiently large $n$) misclassification becomes asymptotically impossible (except via the specified error events), since the LHS of the above grows asymptotically faster than the RHS.

The union bound shows that the probability of any of $E_1, E_2, E_3, E_4$ occurring tends to $\varepsilon$ as $n \to \infty$ (since the probability of the first three tend to zero). So in the regime where

$$t < \frac{1}{2} \tag{63}$$

$$t < 2(q + r - 1) \tag{64}$$

$$q + r > 1 \tag{65}$$

$$\frac{p+1}{2} > q + r + \frac{t}{2} + \max\left(0, \frac{3}{2} - q - r\right) + \max\left(0, \frac{p}{2} - q - \frac{r}{2}\right), \tag{66}$$

---

[13]To be precise, here we can think of fixing the training data and looking purely at the randomness arising from the features in the test point. The resulting $Z^{(\beta)}$ is a standard normal. Since we are using the union bound in our proof finally, this is sufficient for our purposes.

the probability of misclassification tends to $\varepsilon$ for sufficiently large $n$, for any $\varepsilon > 0$.

Consolidation of the above bounds produces the conditions [14]

$$t < \min\left(1 - r, p + 1 - 2(q+r), p - 2, 2q + r - 2\right) \tag{71}$$
$$q + r > 1. \tag{72}$$

Finally, note that the condition $t < r$ comes from the definition of the bi-level model (4.2). This condition simply states that for good generalization we must favor all the features used to determine classes. Since the analysis above holds for any $\varepsilon$, we see that within this regime the probability of misclassification must approach zero in the limit. This completes the proof. Note that while we show that probability of misclassification goes to zero, we do not show it to do so at any particular rate, because the result from Lemma B.2 does not specify the rate of convergence.

## C   Useful results from elsewhere that we need

This section collects results that are used in our proof, but which come from elsewhere or are lightly adapted to our purposes.

**Hanson-Wright inequality [65]**: Let $\mathbf{z}$ be a random vector composed of i.i.d. random variables that are zero mean and with sub-Gaussian norm at most $K$. The sub-Gaussian norm $\|\xi\|_{\psi_2}$ of a random variable $\xi$ is defined as in Rudelson and Vershynin [65],

$$\|\xi\|_{\psi_2} = \inf_{K>0} K \tag{73}$$

$$\text{s.t. } \mathbb{E}\exp\left(\xi^2/K^2\right) \le 2. \tag{74}$$

Then, there exists universal constant $c > 0$ such that for any positive semi-definite matrix $M$ and for every $t \ge 0$, we have

$$\Pr\left[|\mathbf{z}^T M \mathbf{z} - \mathbb{E}[\mathbf{z}^T M \mathbf{z}]| > t\right] \le 2\exp\left\{-c\min\left\{\frac{t^2}{K^4\|M\|_{\mathsf{F}}^2}, \frac{t}{K^2\|M\|_{\mathsf{op}}}\right\}\right\} \tag{75}$$

The next result bounds the eigenvalues of the $n \times n$ matrix $\mathbf{A} = \mathbf{X}^w(\mathbf{X}^w)^\top$, where recall that the rows of $\mathbf{X}^w$ are i.i.d. Gaussians with covariance matrix $\mathbf{\Lambda} = \mathrm{diag}(\lambda_1, \lambda_2, \ldots, \lambda_d)$. Let $\mu_1(\mathbf{A})$ denote the largest eigenvalue and $\mu_n(\mathbf{A})$ denote the smallest eigenvalue of $\mathbf{A}$ respectively.

From Bartlett et al. [4] , we have the following result

**Lemma C.1.** *With probability at least $(1 - 2e^{-n})$, the eigenvalues of $\mathbf{A}$ satisfy:*

$$\sum_j \lambda_j - \Diamond \le \mu_n(\mathbf{A}) \le \mu_1(\mathbf{A}) \le \sum_j \lambda_j + \Diamond, \tag{76}$$

*where,*

$$\Diamond = \frac{32}{9}\left(\lambda_1(1 + \ln 9)n + \sqrt{(1 + \ln 9)n \sum_j \lambda_j^2}\right). \tag{77}$$

---

[14]We can simplify (66) as follows:

$$\frac{p+1}{2} > q + r + \frac{t}{2} \implies t < p + 1 - 2(q+r) \tag{67}$$

$$\frac{p+1}{2} > q + r + \frac{t}{2} + \frac{3}{2} - q - r \implies t < p - 2 \tag{68}$$

$$\frac{p+1}{2} > q + r + \frac{t}{2} + \frac{p}{2} - q - \frac{r}{2} \implies t < 1 - r \tag{69}$$

$$\frac{p+1}{2} > q + r + \frac{t}{2} + \frac{3}{2} - q - r + \frac{p}{2} - q - \frac{r}{2} \implies t < 2q + r - 2. \tag{70}$$

Then we note that $t < \min(r, 1 - r) \implies t < 1/2$.

Next, as stated previously in Lemma B.1 we will use this result to obtain bounds on the eigenvalues of $\mathbf{A}^{-1}$ assuming that $\mathbf{\Lambda}$ is such that $\Diamond \ll \sum_j \lambda_j$.[15]

**Lemma B.1.** *(Eigenvalue bounds on $\mathbf{A}^{-1}$ adapted from Bartlett et al. [4]):*
*If $\mathbf{\Lambda}$ is such that $\Diamond \ll \sum_j \lambda_j$, then with probability at least $(1 - 2e^{-n})$,*

$$\bar{\mu} - \Delta_\mu \leq \mu_n(\mathbf{A}^{-1}) \leq \mu_1(\mathbf{A}^{-1}) \leq \bar{\mu} + \Delta_\mu, \tag{40}$$

*where,*

$$\bar{\mu} = \frac{1}{\sum_j \lambda_j} \tag{41}$$

$$\Diamond = \frac{32}{9} \left( \lambda_1 (1 + \ln 9)n + \sqrt{(1 + \ln 9)n \sum_j \lambda_j^2} \right) \tag{42}$$

$$\Delta_\mu = \bar{\mu} \left( \frac{\Diamond}{\sum_j \lambda_j} + \Theta \left( \frac{\Diamond}{\sum_j \lambda_j} \right)^2 \right). \tag{43}$$

*Further this implies that with probability at least $(1 - 2e^{-n})$,*

$$\left| \mu_i(\mathbf{A}^{-1} - \bar{\mu}\mathbf{I}_n) \right| \leq \Delta_\mu \tag{44}$$

*for all $i \in [n]$.*

*Proof.* Let $S = \sum_j \lambda_j$.

$$\frac{1}{S + \Diamond} = \frac{1}{S} \left( 1 + \frac{\Diamond}{S} \right)^{-1} \tag{78}$$

$$= \frac{1}{S} \left( 1 - \frac{\Diamond}{S} + \Theta \left( \frac{\Diamond}{S} \right)^2 \right) \tag{79}$$

$$= \bar{\mu} - \Delta_\mu, \tag{80}$$

and analogously $(S - \Diamond)^{-1} = \bar{\mu} + \Delta_\mu$. Taking reciprocals of everything in the inequality 76, and since the eigenvalues of $\mathbf{A}$ and $\mathbf{A}^{-1}$ are reciprocals of each other, the desired result follows.

$\square$

As a Corollary of Lemma B.1:

**Corollary C.1.1.** *(Asymptotic eigenvalue bounds on $\mathbf{A}^{-1}$) Considering the asymptotic scaling of the model parameters from the bi-level model (Definition 4.2), in the regime $1 < q + r < (1 + p)/2$,*

$$\bar{\mu} = n^{-p} \tag{81}$$

$$\Delta_\mu \leq c_4 n^{1-p-q-r} \ll \bar{\mu}, \tag{82}$$

*where $\bar{\mu}$ and $\Delta_\mu$ are defined as in Lemma B.1, and $c_4$ is a universal constant.*

*Proof.* From the asymptotic scaling of the $\lambda_j$ from (13) and (14), we see that (from the definition provided in Lemma B.1)

$$\bar{\mu} = \frac{1}{\sum_j \lambda_j} \tag{83}$$

$$= \frac{1}{n^r n^{p-q-r} + (n^p - n^r)(1 - n^q) \cdot n^p/(n^p - n^r)} \tag{84}$$

$$= \frac{1}{n^{p-q} + n^p - n^{p-q}} \tag{85}$$

$$= n^{-p}. \tag{86}$$

---

[15]Note that in the regime $q + r < 1$ (where regression works [56]), we do not have $\Diamond \ll \lambda_j$ and in such scenarios we cannot simply rely on eigenvalue bounds and need to use other techniques in the proof.

Next, we have that

$$\Diamond = \frac{32}{9}\left(\lambda_1(1+\ln 9)n + \sqrt{(1+\ln 9)n\sum_j \lambda_j^2}\right) \tag{87}$$

$$\leq c_1 n^{1+p-q-r} + c_2\sqrt{n(n^r n^{2p-2q-2r} + (n^p - n^r))} \tag{88}$$

$$\leq c_1 n^{1+p-q-r} + c_2\sqrt{n^{1+2p-2q-r} + n^{1+p}} \tag{89}$$

for constants $c_1$ and $c_2$,

The second term is of the order $n^{\max((1-r)/2+p-q,(1+p)/2)}$. Thus, in the regime $q + r < (1 + p)/2$, and since $r < 1$ we have $1 + p - q - r > (1-r)/2 + p - q$ and $1 + p - q - r > (1+p)/2$ and the first term dominates.

Thus, $\Diamond \leq c_3 n^{1+p-q-r}$ for some constant $c_3$ and sufficiently large $n$.

Observe that since $q + r > 1$, $\Diamond \ll \sum_j \lambda_j = n^p$. Thus, we can substitute into our relation for $\Delta_\mu$ from Lemma B.1, to see that

$$\Delta_\mu = \bar{\mu}\left(\frac{\Diamond}{\sum_j \lambda j} + \Theta\left(\frac{\Diamond}{\sum_j \lambda j}\right)^2\right) \tag{90}$$

$$\leq n^{-p}\left((c_3 n^{1+p-q-r})(n^{-p}) + \Theta((c_3 n^{1+p-q-r})^2(n^{-p})^2)\right) \tag{91}$$

$$= n^{-p}(c_3 n^{1-q-r} + \Theta(c_3 n^{2(1-q-r)})). \tag{92}$$

In the regime where $q + r > 1$, the first term in the sum dominates the second, giving us,

$$\Delta_\mu \leq c_4 n^{1-p-q-r} \tag{93}$$

for some constant $c_4$ and sufficiently large $n$. This completes the proof. $\qquad\square$

Finally, in this section, we restate well-known bounds concerning Gaussian random variables.

**Lemma C.2.** *Chi-squared tail bound:*
*Let $\mathbf{z} \sim \mathcal{N}(0, I_n)$. For any $\delta \in (0, 1)$, with probability at least $(1 - 2e^{-n\delta^2})$ we have:*

$$n(1 - \delta) \leq \|\mathbf{z}\|^2 \leq n(1 + \delta). \tag{94}$$

From bounds on the expectation of the maximum of $k$ Gaussians:

**Lemma C.3.** *Let $\mathbf{z}_\alpha = \max_{1 \leq j \leq k} \mathbf{z}_j$ where $\mathbf{z}_j \sim \mathcal{N}(0, 1)$. Then,*

$$\frac{1}{\sqrt{\pi \ln 2}} \cdot \sqrt{\ln k} \leq \mathbb{E}[\mathbf{z}_\alpha] \leq \sqrt{2} \cdot \sqrt{\ln k}. \tag{95}$$

## D   Utility Bounds

The big technical challenge in moving from binary classification (as studied in Muthukumar et al. [56]) to multiclass classification has to do with the nature of the training data. Whereas for binary classification one could change coordinates so that the binary labels only depended on a single Gaussian random variable and were independent of all other directions of Gaussian variation in the covariates, no such change of coordinates exists for multiclass labels. The one-hot-style encoding of the labels fundamentally depends on the realizations of all $k$ of the Gaussian random variables representing each of the $k$ classes. This means that we can no longer simply leverage independence to simplify the analysis and certain clever approaches used to invoke Hanson-Wright are no longer available to us. However, the need remains to appropriately bound quadratic forms of the form $\left|\mathbf{z}_j^\top \mathbf{A}^{-1}\Delta y\right|$ both for the cases when $j$ represents a feature that is not dominant in the computation of $\Delta y$ as well as in cases where $j$ represents a feature that is dominant in $\Delta y$. To be able to control such quantities in the absence of the independence we could leverage in the binary case, this section of the Appendix derives two lemmas which can be viewed as helper bounds. These bounds will later be used to bound the various quantities from (22). Because our focus is on the asymptotic scaling, we will use $c_i$ to denote the appropriate global constants.

In the subsequent lemmas, $\bar{\mu}$ and $\Delta_\mu$ are defined as in the bounds on the eigenvalues of $\mathbf{A}^{-1}$ from Lemma B.1.

The following lemma is used to upper-bound the contamination term $\text{CN}_{\alpha,\beta}$ in Lemma B.5:

**Lemma D.1.** *Let $\Delta y = \mathbf{y}_\alpha - \mathbf{y}_\beta$. Let $\alpha$, $\beta$, and $j$ be distinct. Then, with probability at least $(1 - 7/(ndk))$, we have,*

$$\left| \mathbf{z}_j^\top \mathbf{A}^{-1} \Delta y \right| \le c_7 (\bar{\mu} \sqrt{\frac{n}{k}} \cdot \sqrt{\ln(ndk)} + \Delta_\mu \cdot n/\sqrt{k}), \tag{96}$$

*for some constant $c_7$.*

This next lemma is used to bound the numerator of the survival variation term from (22):

**Lemma D.2.** *Let $\Delta y = \mathbf{y}_\alpha - \mathbf{y}_\beta$. With probability at least $(1 - 5/(nk))$, we have each of*

$$\mathbf{z}_\alpha^\top \mathbf{A}^{-1} \Delta y \le \bar{\mu} (\mathbb{E}[\mathbf{z}_\alpha^\top \mathbf{y}_\alpha] - \mathbb{E}[\mathbf{z}_\alpha^\top \mathbf{y}_\beta]) + c_9 (\bar{\mu} \sqrt{n} \sqrt{\ln(nk)} + \Delta_\mu \cdot n/\sqrt{k}) \tag{97}$$

$$\mathbf{z}_\alpha^\top \mathbf{A}^{-1} \Delta y \ge \bar{\mu} (\mathbb{E}[\mathbf{z}_\alpha^\top \mathbf{y}_\alpha] - \mathbb{E}[\mathbf{z}_\alpha^\top \mathbf{y}_\beta]) - c_9 (\bar{\mu} \sqrt{n} \sqrt{\ln(nk)} + \Delta_\mu \cdot n/\sqrt{k}), \tag{98}$$

*for some constant $c_9$.*

The following corollary of the above is used to lower-bound the relative survival $\widehat{h}_{\alpha,\beta}[\alpha]$, which in turn bounds the SU/CN ratio and the denominator of the survival variation term:

**Corollary D.2.1.** *Let $\Delta y = \mathbf{y}_\alpha - \mathbf{y}_\beta$. With probability at least $(1 - 5/(nk))$, we have,*

$$\mathbf{z}_\alpha^\top \mathbf{A}^{-1} \Delta y \ge c_{10} \bar{\mu} \frac{n}{k} \sqrt{\ln(k)} - c_9 (\bar{\mu} \sqrt{n} \sqrt{\ln(nk)} + \Delta_\mu \cdot n/\sqrt{k}), \tag{99}$$

*for some constant $c_{10}$.*

### D.1 Proof of Lemma D.1

We will write $\mathbf{A}^{-1} = \bar{\mu} \mathbf{I}_n + \Delta A_{inv}$, and split up the expression $\mathbf{z}_j^\top \mathbf{A}^{-1} \Delta y$ into components involving $\bar{\mu} \mathbf{I}_n$, and components involving $\Delta A_{inv}$. To bound the first term, we will use Hanson-Wright, and to bound the second we will use Cauchy-Schwartz. Throughout the proof, we rely on the concentration of the eigenvalues of $\mathbf{A}^{-1}$.

Next, we bound the first term (we set aside the constant $\bar{\mu}$ for now and deal with it later).

#### D.1.1 Bounds on $\mathbf{z}_j^T (\mathbf{y}_\alpha - \mathbf{y}_\beta)$

Throughout this section, let $j$ be a feature index distinct from $\alpha$ and $\beta$. Define the diagonal matrix $\mathbf{M} \in \mathbb{R}^{n \times n}$ with diagonal entries given by:

$$M_{ii} = \begin{cases} 1, & \text{if } \Delta y[i] \ne 0 \\ 0, & \text{otherwise} \end{cases}. \tag{100}$$

In other words, $M_{ii}$ is 1 only if training point $i$ belongs to class $\alpha$ or $\beta$ and is 0 otherwise. Thus for each $i \in [n]$, $M_{ii} \sim Bernoulli(2/k)$ and are independent of each other. We introduce this matrix $\mathbf{M}$ to ensure that our bound reflects the fact that most of the entries of $\Delta y$ are 0. In particular $\Delta y[i] \ne 0$ only if point $i$ belongs to class $\alpha$ or $\beta$ and only contains roughly $2n/k$ non-zero entries.[16] Note that we have by definition,

$$\mathbf{z}_j^T \Delta y = \mathbf{z}_j^T \mathbf{M} \Delta y. \tag{101}$$

Our strategy is to bound $\mathbf{z}_j^\top \mathbf{M} \Delta y$ for every typical realization $\mathcal{M}$ of the random variable $\mathbf{M}$ using the Hanson-Wright inequality. Subsequently, we will apply these bounds with high probability over typical realizations of $\mathbf{M}$ that satisfy the Proposition below, which merely asserts that with high probability, the number of 1s in $\Delta y$ is close to its expected value.

**Proposition D.1.** *For $\delta \in (0,1)$, with probability at least $(1 - 2e^{-\frac{2n\delta^2}{3k}})$, the trace of $\mathbf{M}$ is bounded as:*

$$(1 - \delta) \frac{2n}{k} \le \|\Delta y\|_2^2 = \text{Tr}(\mathbf{M}) \le (1 + \delta) \frac{2n}{k}. \tag{102}$$

---

[16]An alternative bounding technique that first converted $\mathbf{z}_j^\top \Delta y$ to a quadratic form and applied Hanson-Wright would be looser by a factor of $\sqrt{k}$ if we did not introduce $\mathbf{M}$.

*Proof.* Note that $\mathrm{Tr}(\mathbf{M})$ is the sum of $n$ i.i.d Bernoulli random variables with mean $2/k$. The result follows by application of the Chernoff bound. $\qquad\square$

Note that once we fix the realization $\mathcal{M}$, the distributions of $\mathbf{z}_j$ and $\mathbf{\Delta}y$ will now have to be conditioned on this realization and we need to deal with the modified distributions while applying the Hanson-Wright inequality. In particular, once we know that a feature was not the winning feature, it is no longer zero-mean.

Now,

$$\mathbf{z}_j^T \mathcal{M} \mathbf{\Delta}y = \sum_i z_j[i]\mathcal{M}_{ii}\Delta y[i] \tag{103}$$

$$= \sum_{i:\mathcal{M}_{ii}=1} z_j[i]\Delta y[i] \tag{104}$$

$$= \sum_{i:\mathcal{M}_{ii}=1} \left(z_j[i] - \mathbb{E}[z_j[i] \mid M_{ii}=1]\right)\Delta y[i] + \sum_{i:\mathcal{M}_{ii}=1} \mathbb{E}[z_j[i] \mid M_{ii}=1]\Delta y[i] \tag{105}$$

$$= \sum_{i:\mathcal{M}_{ii}=1} \widetilde{z}_{j,\mathcal{M}}[i]\Delta y[i] + \sum_{i:\mathcal{M}_{ii}=1} \mathbb{E}[z_j[i] \mid M_{ii}=1]\Delta y[i], \tag{106}$$

where $\widetilde{z}_{j,\mathcal{M}}[i]$ is now a zero-mean random variable conditioned on the realization $\mathcal{M}$.

First, we bound the term $\sum_{i:\mathcal{M}_{ii}=1} \widetilde{z}_{j,\mathcal{M}}[i]\Delta y[i]$. We collect the elements corresponding to indices where $\mathcal{M}_{ii} = 1$ into the vectors $\mathbf{z}'_{j,\mathcal{M}}$ and $\mathbf{\Delta}y'_\mathcal{M}$, which are both length $\mathrm{Tr}(\mathcal{M})$ (Figure 3 shows an example of collecting elements).

$$\underbrace{\begin{bmatrix}1\\2\\3\\4\\5\end{bmatrix}}_{\widetilde{\mathbf{z}}_{j,\mathcal{M}}}, \underbrace{\begin{bmatrix}1\\0\\-1\\1\\0\end{bmatrix}}_{\mathbf{\Delta}y} \rightarrow \underbrace{\begin{bmatrix}1\\3\\4\end{bmatrix}}_{\mathbf{z}'_{j,\mathcal{M}}}, \underbrace{\begin{bmatrix}1\\-1\\1\end{bmatrix}}_{\mathbf{\Delta}y'_\mathcal{M}}$$

Figure 3: An example of collecting elements at indices where $\mathcal{M}_{ii} = 1$ into smaller vectors of length $\mathrm{Tr}(\mathcal{M})$. Recall that $\mathbf{\Delta}y[i] \neq 0$ iff $\mathcal{M}_{ii} = 0$.

We can then express

$$\sum_{i:\mathcal{M}_{ii}=1} \widetilde{z}_{j,\mathcal{M}}[i]\Delta y[i] \tag{107}$$

$$= (\mathbf{z}'_{j,\mathcal{M}})^T \mathbf{\Delta}y'_\mathcal{M} \tag{108}$$

$$= \frac{1}{4}\left((\mathbf{z}'_{j,\mathcal{M}} + \mathbf{\Delta}y'_\mathcal{M})^T \mathbf{I}_{\mathrm{Tr}(\mathcal{M})}(\mathbf{z}'_{j,\mathcal{M}} + \mathbf{\Delta}y'_\mathcal{M}) - (\mathbf{z}'_{j,\mathcal{M}} - \mathbf{\Delta}y'_\mathcal{M})^T \mathbf{I}_{\mathrm{Tr}(\mathcal{M})}(\mathbf{z}'_{j,\mathcal{M}} - \mathbf{\Delta}y'_\mathcal{M})\right), \tag{109}$$

where we added and subtracted terms in the last equality.

We prove via the subsequent propositions that conditioned on the realization $\mathcal{M}$, the entries of $\mathbf{z}'_{j,\mathcal{M}} \pm \mathbf{\Delta}y'_{j,\mathcal{M}}$ are i.i.d. and sub-Gaussian with bounded norm. Thus, they satisfy the requirements to apply the Hanson-Wright inequality from Rudelson and Vershynin [65] to bound the two quadratic forms in the above expression (109).

**Proposition D.2.** *Conditioned on the realization $\mathcal{M}$, $z'_{j,\mathcal{M}}[i']$ has sub-Gaussian norm at most 6.*

*Proof.* Let $i$ be the original index from which $z'_{j,\mathcal{M}}[i']$ was sampled.

If $j > k$, then $z'_{j,\mathcal{M}}[i'] = \widetilde{z}_{j,\mathcal{M}}[i] = z_j[i]$ irrespective of the realization $\mathcal{M}$ because feature $j$ is not used in the comparison to determine the class label and is independent to $y_\alpha$ and $y_\beta$ (and thus independent to $\mathbf{M}$). Further, $z_j[i]$ is simply a Gaussian (and therefore sub-Gaussian with sub-Gaussian

norm $\|z_j[i]\|_{\psi_2} \leq 2$. Here we use the definition of sub-Gaussian norm from (74) reproduced here for convenience:

The sub-Gaussian norm of a random variable $\xi$ is given by,

$$\|\xi\|_{\psi_2} = \inf_{K>0} K \tag{110}$$

$$\text{s.t. } \mathbb{E} \exp\left(\xi^2/K^2\right) \leq 2. \tag{111}$$

Otherwise, if $j$ is one of the $k$ features that define classes, since

$$z'_{j,\mathcal{M}}[i'] = \widetilde{z}_{j,\mathcal{M}}[i] \tag{112}$$

$$= z_j[i] - \mathbb{E}[z_j[i] \mid M_{ii} = 1], \tag{113}$$

the triangle inequality states that

$$\|\widetilde{z}_{j,\mathcal{M}}[i]\|_{\psi_2} \leq \|z_j[i]\|_{\psi_2} + \|\mathbb{E}[z_j[i] \mid M_{ii} = 1]\|_{\psi_2}. \tag{114}$$

Note that the distribution of $z_j[i]$ conditioned on realization $\mathcal{M}$ is equivalent to the distribution obtained by conditioning on the event $M_{ii} = 1$. So it is sufficient to compute these sub-Gaussian norms conditioned on the event $M_{ii} = 1$.

We will first bound $\|z_j[i]\|_{\psi_2}$. Let $\mathcal{E}_j$ be the event that $z_j[i]$ is the maximum out of the first $k$ features, and let $\mathcal{E}_j^c$ be the complementary event.

First, without conditioning on $\mathcal{E}_j$, we know by well-known results for the standard Gaussian that

$$\mathbb{E} \exp(\mathbf{z}_j[i]^2/5) = \sqrt{\frac{5}{3}} \leq \frac{4}{3}. \tag{115}$$

Using the law of iterated expectation we can relate this to the expectation conditioned on the events $\mathcal{E}_j$ and $\mathcal{E}_j^c$, noting that $P(\mathcal{E}_j) = 1/k$:

$$\frac{4}{3} \geq \mathbb{E} \exp(\mathbf{z}_j[i]^2/5) \tag{116}$$

$$= P(\mathcal{E}_j)\mathbb{E} \exp(\mathbf{z}_j[i]^2/5|\mathcal{E}_j) + P(\mathcal{E}_j^c)\mathbb{E} \exp(\mathbf{z}_j[i]^2/5|\mathcal{E}_j^c) \tag{117}$$

$$= \frac{1}{k}\mathbb{E} \exp(\mathbf{z}_j[i]^2/5|\mathcal{E}_j) + \frac{k-1}{k}\mathbb{E} \exp(\mathbf{z}_j[i]^2/5|\mathcal{E}_j^c). \tag{118}$$

Rearranging terms, we obtain,

$$\frac{k-1}{k}\mathbb{E} \exp(\mathbf{z}_j[i]^2/5|\mathcal{E}_j^c) \leq \frac{4}{3} - \frac{1}{k}\mathbb{E} \exp(\mathbf{z}_j[i]^2/5|\mathcal{E}_j) \tag{119}$$

$$\implies \mathbb{E} \exp(\mathbf{z}_j[i]^2/5|\mathcal{E}_j^c) \leq \frac{k}{k-1}\left(\frac{4}{3} - \frac{1}{k}\mathbb{E} \exp(\mathbf{z}_j[i]^2/5|\mathcal{E}_j)\right) \tag{120}$$

$$\leq \frac{k}{k-1} \cdot \frac{4}{3} \tag{121}$$

$$\leq 2, \tag{122}$$

where in the second to last inequality we used the non-negativity of $\mathbb{E} \exp(\mathbf{z}_j[i]^2/5|\mathcal{E}_j)$ and in the last equality we assumed $k \geq 3$. We then have

$$\mathbb{E} \exp(\mathbf{z}_j[i]^2/5|\mathcal{E}_j^c) = \sum_{m \neq j} \mathbb{E} \exp(\mathbf{z}_j[i]^2/5|\mathcal{E}_j^c \cap \mathcal{E}_m)P(\mathcal{E}_m \mid \mathcal{E}_j^c) \tag{123}$$

$$= \frac{1}{k-1} \sum_{m \neq j} \mathbb{E} \exp(\mathbf{z}_j[i]^2/5|\mathcal{E}_j^c \cap \mathcal{E}_m) \tag{124}$$

where the last equality follows by symmetry. Further by symmetry, all the terms in the above summation that we are averaging are equal, so we can express it as an average of just the terms corresponding to $m = \alpha$ and $m = \beta$, as follows:

$$(124) = \frac{1}{2}\mathbb{E} \exp(\mathbf{z}_j[i]^2/5|\mathcal{E}_j^c \cap \mathcal{E}_\alpha) + \frac{1}{2}\mathbb{E} \exp(\mathbf{z}_j[i]^2/5|\mathcal{E}_j^c \cap \mathcal{E}_\beta) \tag{125}$$

$$= P(\mathcal{E}_\alpha \mid \mathcal{E}_j^c \cap (\mathcal{E}_\alpha \cup \mathcal{E}_\beta))\mathbb{E} \exp(\mathbf{z}_j[i]^2/5|\mathcal{E}_j^c \cap \mathcal{E}_\alpha)$$
$$+ P(\mathcal{E}_\beta \mid \mathcal{E}_j^c \cap (\mathcal{E}_\alpha \cup \mathcal{E}_\beta))\mathbb{E} \exp(\mathbf{z}_j[i]^2/5|\mathcal{E}_j^c \cap \mathcal{E}_\beta), \tag{126}$$

again by symmetry. Since exactly one of $\mathcal{E}_\alpha$ and $\mathcal{E}_\beta$ are true when conditioned on $\mathcal{E}_j^c \cap (\mathcal{E}_\alpha \cup \mathcal{E}_\beta)$, we can rewrite the above as our desired expectation

$$(126) = \mathbb{E}\exp(\mathbf{z}_j[i]^2/5|\mathcal{E}_j^c \cap (\mathcal{E}_\alpha \cup \mathcal{E}_\beta)) \tag{127}$$

$$= \mathbb{E}\exp(\mathbf{z}_j[i]^2/5|\mathcal{E}_\alpha \cup \mathcal{E}_\beta) \tag{128}$$

$$= \mathbb{E}\exp(\mathbf{z}_j[i]^2/5|M_{ii} = 1), \tag{129}$$

since $M_{ii} = 1$ is equivalent to the event $\mathcal{E}_\alpha \cup \mathcal{E}_\beta$. Thus, conditioned on the event $M_{ii} = 1$, $\|z_j[i]\|_{\psi_2} \leq \sqrt{5}$.

Next we consider $\|\mathbb{E}[z_j[i] \mid M_{ii} = 1]\|_{\psi_2}$. By a similar argument to above, we have that $\mathbb{E}[z_j[i] \mid M_{ii} = 1] = \mathbb{E}[z_j[i] \mid \mathcal{E}_j^c]$, so we will focus on the second quantity instead. Bounds on the max of Gaussians (Lemma C.3) state that:

$$0 < \mathbb{E}[\mathbf{z}_j[i] \mid \mathcal{E}_j] \leq \sqrt{2\log(k)} \tag{130}$$

$$\implies \quad 0 > \mathbb{E}[\mathbf{z}_j[i] \mid \mathcal{E}_j^c] \geq -\frac{1}{k-1}\sqrt{2\log(k)} \geq -2 \tag{131}$$

$$\implies \quad \exp\left(\frac{\mathbb{E}[\mathbf{z}_j[i] \mid \mathcal{E}_j^c]^2}{3^2}\right) < 2. \tag{132}$$

In the second last inequality we use the fact that the function $f(k) = \left|\sqrt{2\log k}/(k-1)\right|$ is monotonically decreasing in $k$ and assumed $k \geq 3$.

Thus, the (constant) random variable $\mathbb{E}[\mathbf{z}_j[i] \mid M_{ii} = 1]$ is sub-Gaussian with parameter 3. So, by the triangle inequality, conditioned on $M_{ii} = 1$

$$\|\widetilde{z}_{j,m}[i]\|_{\psi_2} \leq \|z_j[i]\|_{\psi_2} + \|\mathbb{E}[\widetilde{z}_{j,m}]\|_{\psi_2} \tag{133}$$

$$\leq \sqrt{5} + 3 \tag{134}$$

$$\leq 6. \tag{135}$$

This completes the proof that conditioned on the realization $\mathcal{M}$, $z'_{j,\mathcal{M}}[i']$ is sub-Gaussian with norm at most 6. $\qquad\square$

We can now prove our target result:

**Proposition D.3.** *With probability at least* $(1 - 6/(ndk))$,

$$\left|\mathbf{z}_j^\top \mathbf{\Delta} y\right| \leq c_6 \sqrt{\frac{n}{k}} \cdot \sqrt{\log(ndk)}. \tag{136}$$

*for universal constant $c_6$.*

*Proof.* Our strategy will be to bound $\mathbf{z}_j^\top \mathbf{\Delta} y = \mathbf{z}_j^\top \mathbf{M}\mathbf{\Delta} y$ for every typical realization $\mathcal{M}$ of $\mathbf{M}$ that satisfies Proposition D.1. Recall that for a given realization $\mathcal{M}$ we have,

$$\mathbf{z}_j^T \mathcal{M}\mathbf{\Delta} y = \sum_{i:\mathcal{M}_{ii}=1} \widetilde{z}_{j,\mathcal{M}}[i]\Delta y[i] + \sum_{i:\mathcal{M}_{ii}=1} \mathbb{E}[z_j[i] \mid M_{ii} = 1]\Delta y[i]. \tag{137}$$

We will use Hanson-Wright to bound the first term, which we previously expressed in (109) as:

$$\sum_{i:\mathcal{M}_{ii}=1} \widetilde{z}_{j,\mathcal{M}}[i]\Delta y[i] \tag{138}$$

$$= \frac{1}{4}\left((\mathbf{z}'_{j,\mathcal{M}} + \mathbf{\Delta} y'_{\mathcal{M}})^T \mathbf{I}_{\text{Tr}(\mathcal{M})}(\mathbf{z}'_{j,\mathcal{M}} + \mathbf{\Delta} y'_{\mathcal{M}}) - (\mathbf{z}'_{j,\mathcal{M}} - \mathbf{\Delta} y'_{\mathcal{M}})^T \mathbf{I}_{\text{Tr}(\mathcal{M})}(\mathbf{z}'_{j,\mathcal{M}} - \mathbf{\Delta} y'_{\mathcal{M}})\right). \tag{139}$$

By Proposition D.2, the sub-Gaussian conditions for the entries of $\mathbf{z}'_{j,m}$ are satisfied. Further, $\mathbf{\Delta} y'_{\mathcal{M}}$ is bounded in $[-1, 1]$, so $\|\mathbf{\Delta} y'_{\mathcal{M}}\|_{\psi_2} \leq 2$. Thus, by the triangle inequality, the sub-Gaussian norm of the entries of $\mathbf{z}'_{j,\mathcal{M}} \pm \mathbf{\Delta} y'_{\mathcal{M}}$ is bounded by $K \leq 6 + 2 = 8$. Also note that conditioned on the

realization $\mathcal{M}$, $\mathbf{z}'_{j,\mathcal{M}}$ is zero-mean by construction and $\boldsymbol{\Delta} y'_{\mathcal{M}}$ is zero-mean by symmetry between $\alpha$ and $\beta$, so we can now apply the Hanson-Wright inequality to both terms.

We choose parameter

$$t = \frac{K^2}{\sqrt{c}} \sqrt{\mathrm{Tr}(\mathcal{M})} \sqrt{\log(ndk)}. \tag{140}$$

where $c$ is the constant from the Hanson-Wright result.

So

$$\frac{t^2}{K^4 \|\mathbf{I}_{\mathrm{Tr}(\mathcal{M})}\|_{\mathsf{F}}^2} = \frac{1}{c} \log(ndk) \tag{141}$$

$$\frac{t}{K^2 \|\mathbf{I}_{\mathrm{Tr}(\mathcal{M})}\|_{\mathsf{op}}} = \frac{1}{\sqrt{c}} \sqrt{\mathrm{Tr}(\mathcal{M})} \sqrt{\log(ndk)} > \frac{1}{c} \log(ndk). \tag{142}$$

The last inequality follows since with high probability $\mathrm{Tr}(\mathcal{M}) = \Theta(\sqrt{n/k})$, by Proposition D.1, $\sqrt{\mathrm{Tr}(\mathcal{M})} \sqrt{\log(ndk)} = \Theta(\sqrt{n \log(ndk)/k})$ grows faster than $\log(ndk)$.

Finally, note that:

$$\mathbb{E}[(\mathbf{z}'_{j,\mathcal{M}})^T \boldsymbol{\Delta} y'_{\mathcal{M}} \mid \mathbf{M} = \mathcal{M}] = \sum_{i:\mathcal{M}_{ii}=1} \mathbb{E}[\widetilde{z}_{j,\mathcal{M}}[i] \Delta y[i] \mid \mathbf{M} = \mathcal{M}] \tag{143}$$

$$= \sum_{i:\mathcal{M}_{ii}=1} \mathbb{E}[\widetilde{z}_{j,\mathcal{M}}[i] \Delta y[i] \mid M_{ii} = 1] \tag{144}$$

$$= \sum_{i:\mathcal{M}_{ii}=1} \frac{1}{2} \mathbb{E}[\widetilde{z}_{j,\mathcal{M}}[i] \mid \Delta y[i] = 1] - \frac{1}{2} \mathbb{E}[\widetilde{z}_{j,\mathcal{M}}[i] \mid \Delta y[i] = -1] \tag{145}$$

$$= 0, \tag{146}$$

where the last equation follows by symmetry. Knowing which of $\mathbf{z}_\alpha[i]$ or $\mathbf{z}_\beta[i]$ was the maximum does not change the conditional expectation of $\widetilde{z}_{j,\mathcal{M}}[i]$.

So, applying Hanson-Wright, with probability at least $(1 - 4/(ndk))$ we have

$$-\frac{K^2}{2} c_5 \sqrt{\mathrm{Tr}(\mathcal{M})} \sqrt{\log(ndk)} = -\frac{t}{2} \leq \widetilde{\mathbf{z}}_{j,m}^T \Delta \mathbf{y} \leq \frac{t}{2} = \frac{K^2}{2} c_5 \sqrt{\mathrm{Tr}(\mathcal{M})} \sqrt{\log(ndk)}, \tag{147}$$

where $c_5 = \frac{1}{\sqrt{c}}$.

We next consider the second term $\sum_{i:\mathcal{M}_{ii}=1} \mathbb{E}[z_j[i] \mid M_{ii} = 1] \Delta y[i]$ from (106) conditioned on the realization $\mathcal{M}$. By an identical symmetry argument as for the previous term we have, $0 \geq \mathbb{E}[z_j[i] \mid \mathcal{E}_j^c] = \mathbb{E}[z_j[i] \mid M_{ii} = 1]$. Then as a consequence of Lemma C.3 and using the fact that $M_{ii} = 1$ implies $z_j[i]$ is not the maximum of $k$ Gaussians we have, $\mathbb{E}[z_j[i] \mid \mathcal{E}_j^c] \geq -2\sqrt{\log(k)}/(k-1)$. So we can bound

$$\left| \sum_{i:\mathcal{M}_{ii}=1} \mathbb{E}[z_j[i] \mid M_{ii} = 1] \Delta y[i] \right| \leq \frac{2\sqrt{\log(k)}}{k-1} \left| \sum_{i:\mathcal{M}_{ii}=1} \boldsymbol{\Delta} y_i \right| \leq \frac{2\delta'\sqrt{\log(k)}}{k-1}, \tag{148}$$

with probability $1 - 2e^{-\delta'^2/(6 \cdot \mathrm{Tr}(\mathcal{M}))}$, by application of the Chernoff bound and using the fact that conditioned on $M_{ii} = 1$, $\Delta y[i]$ takes value $\pm 1$ with probability half by symmetry among features $\alpha$ and $\beta$.

Next, we apply the high probability bounds above on typical realizations $\mathcal{M}$. In particular, we substitute bounds on $\mathrm{Tr}(\mathbf{M})$ from (102) from Proposition D.1 with $\delta = 1/2$ into (148), and set $\delta' = \sqrt{6(1+\delta)(n/k)\log(ndk)}$. Then $e^{-\delta'^2/(6 \cdot \mathrm{Tr}(\mathbf{M}))} \leq 1/(ndk)$ and $e^{-\frac{2n\delta^2}{3k}} < 1/(ndk)$, so

using the union bound we have with probability at least $(1 - 4/(ndk) - 1/(ndk) - 1/(ndk))$,

$$\left|\mathbf{z}_j^T \Delta \mathbf{y}\right| \leq \left|\sum_{i:\mathcal{M}_{ii}=1} \widetilde{z}_{j,\mathcal{M}}[i]\Delta y[i]\right| + \left|\sum_{i:\mathcal{M}_{ii}=1} \mathbb{E}[z_j[i] \mid M_{ii} = 1]\Delta y[i]\right| \tag{149}$$

$$\leq \frac{K^2}{2}c_5\sqrt{1+\delta} \cdot \sqrt{\frac{2n}{k}} \cdot \sqrt{\log(ndk)} + \frac{2\sqrt{(1+\delta)(n/k)\log(ndk)}\sqrt{\log(k)}}{k-1} \tag{150}$$

$$\leq \frac{K^2}{2}c_5\sqrt{1+\delta} \cdot \sqrt{\frac{2n}{k}} \cdot \sqrt{\log(ndk)} + \frac{2\sqrt{(1+\delta)}\sqrt{\log(k)}}{k-1} \cdot \sqrt{\frac{n}{k}} \cdot \sqrt{\log(ndk)} \tag{151}$$

$$\leq c_6\sqrt{\frac{n}{k}} \cdot \sqrt{\log(ndk)}, \tag{152}$$

for a suitable choice of $c_6$. $\qquad\square$

### D.1.2 Bounds on $\mathbf{z}_j^\top \mathbf{A}^{-1}(\mathbf{y}_\alpha - \mathbf{y}_\beta)$

We can now prove bounds on our target quantity. We restate the lemma that we are trying to prove below for convenience.

**Lemma D.1.** *Let* $\Delta y = \mathbf{y}_\alpha - \mathbf{y}_\beta$. *Let* $\alpha$, $\beta$, *and* $j$ *be distinct. Then, with probability at least* $(1 - 7/(ndk))$, *we have,*

$$\left|\mathbf{z}_j^\top \mathbf{A}^{-1}\Delta y\right| \leq c_7(\bar{\mu}\sqrt{\frac{n}{k}} \cdot \sqrt{\ln(ndk)} + \Delta_\mu \cdot n/\sqrt{k}), \tag{96}$$

*for some constant* $c_7$.

*Proof.* We can rewrite

$$\mathbf{z}_j^\top \mathbf{A}^{-1}\Delta y = \mathbf{z}_j^\top \left(\bar{\mu}\mathbf{I}_n + \Delta A_{inv}\right)\Delta y \tag{153}$$

$$= \bar{\mu}\mathbf{z}_j^\top \Delta y + \mathbf{z}_j^\top \Delta A_{inv}\Delta y. \tag{154}$$

Next we can bound $\left|\mathbf{z}_j^\top \Delta A_{inv}\Delta y\right|$ simply as

$$\left|\mathbf{z}_j^\top \Delta A_{inv}\Delta y\right| \leq \|\mathbf{z}_j\|_2\|\Delta A_{inv}\Delta y\|_2 \tag{155}$$

$$\leq \|\Delta A_{inv}\|_{op}\|\mathbf{z}_j\|_2\|\Delta y\|_2 \tag{156}$$

$$\leq \Delta_\mu\|\mathbf{z}_j\|_2\|\Delta y\|_2, \tag{157}$$

where we use the fact that $\Delta A_{inv}$ is a symmetric matrix and its 2-norm is its maximum absolute eigenvalue. We obtain the eigenvalue bounds for $\Delta A_{inv}$ from Lemma B.1, holding with probability at least $1 - 2e^{-n}$.

So, by the triangle inequality, we have with probability at least $(1 - 6/(ndk) - 2e^{-n} - 2e^{-\frac{2n\delta^2}{3k}} - 2e^{-n\delta^2})$

$$\left|\mathbf{z}_j^\top \mathbf{A}^{-1}\Delta y\right| \leq c_6\bar{\mu}\sqrt{\frac{n}{k}} \cdot \sqrt{\ln(ndk)} + \Delta_\mu \cdot \sqrt{(1+\delta)n} \cdot \sqrt{(1+\delta)\frac{2n}{k}}. \tag{158}$$

The first term follows from Proposition D.3, and the second from our bound on $\text{Tr}(\mathbf{M}) = \|\Delta y\|_2^2$ from Proposition D.1, as well as an analogous application of the chi-squared bound (Lemma C.2) on $\|\mathbf{z}_j\|_2$.

The proof follows by setting $\delta$ to any value in $(0, 1)$, choosing an appropriate constant $c_7$, and noting that for large enough $n$, $1/(ndk) \gg d_1 e^{-d_2 n/k}$ for any positive constants $d_1, d_2$. $\qquad\square$

## D.2 Proof of Lemma D.2

Next we use a similar technique as in Appendix D.1 to bound $\mathbf{z}_\alpha^\top \mathbf{A}^{-1}\Delta y$. We will write $\mathbf{A}^{-1} = \bar{\mu}\mathbf{I}_n + \Delta A_{inv}$, and split up the expression $\mathbf{z}_\alpha^\top \mathbf{A}^{-1}\Delta y$ into components involving $\bar{\mu}\mathbf{I}_n$, and components involving $\Delta A_{inv}$.

**Proposition D.4.** *Consider two arbitrary length-$n$ zero-mean vectors $\mathbf{y}$ and $\mathbf{z}$ whose components each has sub-Gaussian norm at most $K$. With probability at least $1 - 4/(nk)$ we have each of*

$$\mathbf{z}^\top\mathbf{y} \le \mathbb{E}[\mathbf{z}^\top\mathbf{y}] + 2c_8\sqrt{n}\cdot\sqrt{\ln(nk)} \tag{159}$$

$$\mathbf{z}^\top\mathbf{y} \ge \mathbb{E}[\mathbf{z}^\top\mathbf{y}] - 2c_8\sqrt{n}\cdot\sqrt{\ln(nk)}, \tag{160}$$

*for some universal constant $c_8$.*

*Proof.* The upper-bound follows as

$$\mathbf{z}^\top\mathbf{y} = \frac{1}{4}\left((\mathbf{z}+\mathbf{y})^\top(\mathbf{z}+\mathbf{y}) - (\mathbf{z}-\mathbf{y})^\top(\mathbf{z}-\mathbf{y})\right) \tag{161}$$

$$\le \mathbb{E}[\mathbf{z}^\top\mathbf{y}] + \frac{K^2}{2\sqrt{c}}\sqrt{n}\cdot\sqrt{\ln nk}, \tag{162}$$

with probability at least $(1 - 4/(nk))$, where we apply the Hanson-Wright inequality to each of the quadratic terms with $t = \frac{K^2}{\sqrt{c}}\sqrt{n}\sqrt{\ln(nk)}$ and use the fact that, letting $\mathbf{M} = \mathbf{I}_n$, $\|\mathbf{M}\|_F^2 = n$, $\|\mathbf{M}\|_{op} = 1$. The lower-bound can be obtained analogously, and an appropriate choice of $c_8$ completes the proof. $\square$

From this, we can now prove Lemma D.2, restated below for convenience:

**Lemma D.2.** *Let $\mathbf{\Delta}y = \mathbf{y}_\alpha - \mathbf{y}_\beta$. With probability at least $(1 - 5/(nk))$, we have each of*

$$\mathbf{z}_\alpha^\top\mathbf{A}^{-1}\mathbf{\Delta}y \le \bar{\mu}(\mathbb{E}[\mathbf{z}_\alpha^\top\mathbf{y}_\alpha] - \mathbb{E}[\mathbf{z}_\alpha^\top\mathbf{y}_\beta]) + c_9(\bar{\mu}\sqrt{n}\sqrt{\ln(nk)} + \Delta_\mu \cdot n/\sqrt{k}) \tag{97}$$

$$\mathbf{z}_\alpha^\top\mathbf{A}^{-1}\mathbf{\Delta}y \ge \bar{\mu}(\mathbb{E}[\mathbf{z}_\alpha^\top\mathbf{y}_\alpha] - \mathbb{E}[\mathbf{z}_\alpha^\top\mathbf{y}_\beta]) - c_9(\bar{\mu}\sqrt{n}\sqrt{\ln(nk)} + \Delta_\mu \cdot n/\sqrt{k}), \tag{98}$$

*for some constant $c_9$.*

*Proof.* We have

$$\mathbf{z}_\alpha^\top\mathbf{A}^{-1}(\mathbf{y}_\alpha - \mathbf{y}_\beta) = \mathbf{z}_\alpha^\top\left(\bar{\mu}\mathbf{I}_n + \mathbf{\Delta}A_{inv}\right)(\mathbf{y}_\alpha - \mathbf{y}_\beta) \tag{163}$$

$$= \bar{\mu}\mathbf{z}_\alpha^\top(\mathbf{y}_\alpha - \mathbf{y}_\beta) + \mathbf{z}_\alpha^\top\mathbf{\Delta}A_{inv}(\mathbf{y}_\alpha - \mathbf{y}_\beta) \tag{164}$$

$$= \bar{\mu}\mathbf{z}_\alpha^\top(\mathbf{y}_\alpha - \mathbf{y}_\beta) + \mathbf{z}_\alpha^\top\mathbf{\Delta}A_{inv}(\mathbf{y}_\alpha^{oh} - \mathbf{y}_\beta^{oh}). \tag{165}$$

We again simply bound

$$|\mathbf{z}_\alpha^\top\mathbf{\Delta}A_{inv}\mathbf{\Delta}y| \le \|\mathbf{z}_j\|_2\|\mathbf{\Delta}A_{inv}\mathbf{\Delta}y\|_2 \tag{166}$$

$$\le \|\mathbf{\Delta}A_{inv}\|_{op}\|\mathbf{z}_\alpha\|_2\|\mathbf{\Delta}y\|_2 \tag{167}$$

$$\le \Delta_\mu\|\mathbf{z}_\alpha\|_2\|\mathbf{\Delta}y\|_2 \tag{168}$$

$$\le \Delta_\mu \cdot \sqrt{(1+\delta)n}\cdot\sqrt{(1+\delta)\frac{2n}{k}} \tag{169}$$

$$= \Delta_\mu(1+\delta)\sqrt{2}\frac{n}{\sqrt{k}}, \tag{170}$$

with probability $(1-2e^{-n\delta^2} - 2e^{-\frac{2n\delta^2}{3k}})$, using chi-squared bounds for $\mathbf{z}_\alpha$ (Lemma C.2) and Chernoff bounds for $\mathbf{\Delta}y$ (Proposition D.1).

With probability $(1 - 2e^{-n\delta^2} - 2e^{-\frac{2n\delta^2}{3k}})$, we get each of

$$\mathbf{z}_\alpha^T\mathbf{A}^{-1}(\mathbf{y}_\alpha - \mathbf{y}_\beta) \le \bar{\mu}\mathbf{z}_\alpha^T(\mathbf{y}_\alpha - \mathbf{y}_\beta) + \Delta_\mu(1+\delta)\sqrt{2}\frac{n}{\sqrt{k}} \tag{171}$$

$$\mathbf{z}_\alpha^T\mathbf{A}^{-1}(\mathbf{y}_\alpha - \mathbf{y}_\beta) \ge \bar{\mu}\mathbf{z}_\alpha^T(\mathbf{y}_\alpha - \mathbf{y}_\beta) - \Delta_\mu(1+\delta)\sqrt{2}\frac{n}{\sqrt{k}}. \tag{172}$$

By applying Proposition D.4 on the relevant terms, setting $\delta$ to be an arbitrary value in $(0,1)$, and choosing an appropriate constant $c_9$, we obtain with probability $(1 - 5/(nk))$ each of

$$\mathbf{z}_\alpha^T\mathbf{A}^{-1}(\mathbf{y}_\alpha - \mathbf{y}_\beta) \le \bar{\mu}(\mathbb{E}[\mathbf{z}_\alpha^T\mathbf{y}_\alpha] - \mathbb{E}[\mathbf{z}_\alpha^T\mathbf{y}_\beta]) + c_9(\bar{\mu}\sqrt{n}\sqrt{\ln(nk)} + \Delta_\mu n/\sqrt{k}) \tag{173}$$

$$\mathbf{z}_\alpha^T\mathbf{A}^{-1}(\mathbf{y}_\alpha - \mathbf{y}_\beta) \ge \bar{\mu}(\mathbb{E}[\mathbf{z}_\alpha^T\mathbf{y}_\alpha] - \mathbb{E}[\mathbf{z}_\alpha^T\mathbf{y}_\beta]) - c_9(\bar{\mu}\sqrt{n}\sqrt{\ln(nk)} + \Delta_\mu n/\sqrt{k}). \tag{174}$$

The probability comes from the union bound $(1 - 2e^{-n\delta^2} - 2e^{-\frac{2n\delta^2}{3k}} - 4/(nk)) \ge 1 - 5/(nk)$ (for sufficiently large $n$). $\square$

### D.3 Proof of Corollary D.2.1

We claim the following bound:

**Proposition D.5.** *Bounds on* $\mathbb{E}[\mathbf{z}_\alpha^\top \mathbf{y}_\alpha]$.

$$\frac{1}{\sqrt{\pi \ln 2}} \cdot \frac{n}{k} \cdot \sqrt{\ln k} \leq \mathbb{E}[\mathbf{z}_\alpha^\top \mathbf{y}_\alpha] \leq \sqrt{2} \cdot \frac{n}{k} \cdot \sqrt{\ln k} \tag{175}$$

*Proof.*

$$\mathbb{E}[\mathbf{z}_\alpha^\top \mathbf{y}_\alpha] = \mathbb{E}[\mathbf{z}_\alpha^\top \mathbf{y}_\alpha^{oh}] - \mathbb{E}[\mathbf{z}_\alpha^\top \frac{1}{c} \mathbf{1}] \tag{176}$$

$$= \mathbb{E}[\mathbf{z}_\alpha^\top \mathbf{y}_\alpha^{oh}] \tag{177}$$

$$= n \left( \mathbb{E}[z_{\alpha,i} y_{\alpha,i}^{oh} | y_{\alpha,i}^{oh} = 1] P(y_{\alpha,i}^{oh} = 1) + \mathbb{E}[z_{\alpha,i} y_{\alpha,i}^{oh} | y_{\alpha,i}^{oh} = 0] P(y_{\alpha,i}^{oh} = 0) \right) \tag{178}$$

$$= \frac{n}{k} \mathbb{E}[z_{\alpha,i} | y_{\alpha,i}^{oh} = 1]. \tag{179}$$

So the desired bound follows from the bounds in Lemma C.3. $\square$

We can obtain a similar bound for when $\beta \neq \alpha$:

**Proposition D.6.** *Bounds on* $\mathbb{E}[\mathbf{z}_\alpha^\top \mathbf{y}_\beta]$.

$$-\sqrt{2} \cdot \frac{n}{k} \cdot \frac{1}{k-1} \cdot \sqrt{\ln k} \leq \mathbb{E}[\mathbf{z}_\alpha^\top \mathbf{y}_\beta] \leq -\frac{1}{\sqrt{\pi \ln 2}} \cdot \frac{n}{k} \cdot \frac{1}{k-1} \cdot \sqrt{\ln k} \tag{180}$$

*Proof.* Observe that,

$$\mathbb{E}[\mathbf{z}_\alpha^\top \mathbf{y}_\beta] = \mathbb{E}[\mathbf{z}_\alpha^\top \mathbf{y}_\beta^{oh}] - \mathbb{E}[\mathbf{z}_\alpha^\top \frac{1}{k} \mathbf{1}] \tag{181}$$

$$= \mathbb{E}[\mathbf{z}_\alpha^\top \mathbf{y}_\beta^{oh}] - \frac{1}{k} \mathbb{E}[\mathbf{z}_\alpha]^\top \mathbf{1} \tag{182}$$

$$= \mathbb{E}[\mathbf{z}_\alpha^\top \mathbf{y}_\beta^{oh}] \tag{183}$$

$$= \sum_i \mathbb{E}[z_{\alpha,i} y_{\beta,i}^{oh}] \tag{184}$$

$$= n \left( \mathbb{E}[z_{\alpha,i} y_{\beta,i}^{oh} | y_{\beta,i}^{oh} = 1] P(y_{\beta,i}^{oh} = 1) + \mathbb{E}[z_{\alpha,i} y_{\beta,i}^{oh} | y_{\beta,i}^{oh} = 0] P(y_{\beta,i}^{oh} = 1) \right) \tag{185}$$

$$= \frac{n}{k} \mathbb{E}[z_{\alpha,i} | y_{\beta,i}^{oh} = 1] \tag{186}$$

Now, observe that

$$\mathbb{E}[z_{\alpha,i} | y_{\alpha,i}^{oh} = 0] = \sum_{\beta \neq \alpha} \mathbb{E}[z_{\alpha,i} | y_{\beta,i}^{oh} = 1] \Pr(y_{\beta,i} = 1 \mid y_{\alpha,i}^{oh} = 0) \tag{187}$$

$$= \frac{1}{k-1} \sum_{\beta \neq \alpha} \mathbb{E}[z_{\alpha,i} | y_{\beta,i}^{oh} = 1] \tag{188}$$

$$= \mathbb{E}[z_{\alpha,i} | y_{\beta,i}^{oh} = 1] \tag{189}$$

for a particular $\beta$, by symmetry over the possible $\beta$.

Next we bound $\mathbb{E}[z_{\alpha,i} | y_{\alpha,i}^{oh} = 0]$ as follows:

$$\mathbb{E}[z_{\alpha,i} | y_{\alpha,i}^{oh} = 1] P(y_{\alpha,i}^{oh} = 1)$$

$$+ \quad \mathbb{E}[z_{\alpha,i} | y_{\alpha,i}^{oh} = 0] P(y_{\alpha,i}^{oh} = 0) = \mathbb{E}[z_{\alpha,i}] = 0 \tag{190}$$

$$\implies \quad \mathbb{E}[z_{\alpha,i} | y_{\alpha,i}^{oh} = 0] \frac{k-1}{k} = -\mathbb{E}[z_{\alpha,i} | y_{\alpha,i}^{oh} = 1] \frac{1}{k} \tag{191}$$

$$\implies \quad \mathbb{E}[z_{\alpha,i} | y_{\alpha,i}^{oh} = 0] = -\mathbb{E}[z_{\alpha,i} | y_{\alpha,i}^{oh} = 1] \frac{1}{k-1} \tag{192}$$

Thus, substituting in the results from Lemma C.3, and plugging back into (186), we obtain

$$-\sqrt{2} \cdot \frac{n}{k} \cdot \frac{1}{k-1} \cdot \sqrt{\ln k} \leq \mathbb{E}[\mathbf{z}_\alpha^\top \mathbf{y}_\beta] \leq -\frac{1}{\sqrt{\pi \ln 2}} \cdot \frac{n}{k} \cdot \frac{1}{k-1} \cdot \sqrt{\ln k}, \qquad (193)$$

the desired result. □

We can now prove Corollary D.2.1, which we restate below for convenience:

**Corollary D.2.1.** *Let* $\mathbf{\Delta} y = \mathbf{y}_\alpha - \mathbf{y}_\beta$. *With probability at least* $(1 - 5/(nk))$, *we have,*

$$\mathbf{z}_\alpha^\top \mathbf{A}^{-1} \mathbf{\Delta} y \geq c_{10} \bar{\mu} \frac{n}{k} \sqrt{\ln(k)} - c_9 (\bar{\mu}\sqrt{n}\sqrt{\ln(nk)} + \Delta_\mu \cdot n/\sqrt{k}), \qquad (99)$$

*for some constant* $c_{10}$.

*Proof.* This follows by substituting the lower bound from (175) in Proposition D.5 and the upper bound from (180) in Proposition D.6 into (98) from Lemma D.2, making an appropriate choice for $c_{10}$. □

# E   Misclassification Events: Proof of Lemmas used in Theorem 5.1

With the previous section's utility bounds that allow us to deal with multiclass training data in hand, we are in a position to establish all the lemmas that we need to analyze misclassification.

## E.1   Proof of Lemma B.2: Lower bound on $\min_\beta (X_\alpha - X_\beta)$

With these bounds in hand, we can look at each misclassification event in turn. The first event to consider is if the best competing feature is unusually close to the true (maximum) feature.

**Lemma B.2.** *(**Lower bound on the closest feature margin as** $k \to \infty$): For any constant $\varepsilon > 0$, there exists a constant $\theta$ such that, for sufficiently large $k$ with probability at least $(1 - \varepsilon)$,*

$$\min_{\beta:1\leq\beta\neq\alpha\leq k} (x_{test}[\alpha] - x_{test}[\beta]) \geq \frac{\theta}{\sqrt{2\ln(k)}}. \qquad (45)$$

*Here, $\alpha$ is fixed and corresponds to the index of the true class — i.e. $\alpha$ corresponds to the index of the maximum feature among the first $k$ features.*

*Proof.* The following result from [34] whose proof we reproduce here[17], enables us to bound the closest feature margin as:

$$\Pr(\min_\beta (x_{test}[\alpha] - x_{test}[\beta]) > \theta/\sqrt{2\ln(k)}) \geq c_{11} e^{-\theta}, \qquad (194)$$

for some universal positive constant $c_{11}$, for sufficiently large $k$. Thus, by selecting a constant $\theta$ such that $c_{11} e^{-\theta} = 1 - \varepsilon$ and choosing a sufficiently large $k$, we have that with probability $(1 - \varepsilon)$:

$$\min_\beta(x_{test}[\alpha] - x_{test}[\beta]) \geq \theta/\sqrt{2\ln k}. \qquad (195)$$

The proof [34] is reproduced below, with slight adaptations to match our use-case: Let $\beta$ be the index of the largest competing feature to $x_{test}[\alpha]$. Then, their joint PDF becomes

$$f(x_{test}[\beta], x_{test}[\alpha]) = k(k-1)F(x_{test}[\beta])^{k-2}f(x_{test}[\beta])f(x_{test}[\alpha])\mathbf{1}(x_{test}[\beta] < x_{test}[\alpha]). \qquad (196)$$

where $F$ and $f$ are the CDF and PDF of the standard Gaussian. Let

$$x = \frac{\theta}{\sqrt{2\ln(k)}}. \qquad (197)$$

Thus,

$$\Pr(x_{test}[\alpha] - x_{test}[\beta] > x) = k(k-1)J, \qquad (198)$$

---

[17]We do this for the convenience of the reviewers since the source we are citing is a URL online. We believe that this is in the spirit of fair use.

where $J$ is defined as

$$J = \int_{-\infty}^{\infty} \int_{x}^{\infty} F(w)^{k-2} f(w) f(v+w) \, dv \, dw \tag{199}$$

$$= \int_{-\infty}^{\infty} F(w)^{k-2} f(w) \left( \int_{x}^{\infty} f(v+w) \, dv \right) dw \tag{200}$$

$$= \int_{-\infty}^{\infty} F(w)^{k-2} f(w)(1 - F(x+w)) \, dw. \tag{201}$$

Substituting $u = F(w)$, we have that

$$J = \int_{0}^{1} u^{k-2}(1 - F(x + F^{-1}(u))) \, du \tag{202}$$

$$= \int_{0}^{1-\ln(k)^2/k} u^{k-2}(1 - F(x + F^{-1}(u))) \, du + \int_{1-\ln(k)^2/k}^{1-1/(k\ln(k))} u^{k-2}(1 - F(x + F^{-1}(u))) \, du$$

$$+ \int_{1-1/(k\ln(k))}^{1} u^{k-2}(1 - F(x + F^{-1}(u))) \, du, \tag{203}$$

splitting $[0, 1]$ into three intervals, and integrating separately over each one. Let the three integrals be $J_1$, $J_2$, and $J_3$.

We have that

$$J_1 = \int_{0}^{1-\ln(k)^2/k} u^{k-2}(1 - F(x + F^{-1}(u))) \, du \tag{204}$$

$$\leq \int_{0}^{1-\ln(k)^2/k} u^{k-2} \, du \tag{205}$$

$$\leq (1 - \ln(k)^2/k)^{k-2} \tag{206}$$

$$\leq \exp\left( -\frac{k-2}{k} \ln(k)^2 \right) \tag{207}$$

$$= o\left( \frac{1}{k^2} \right). \tag{208}$$

Similarly,

$$J_3 = \int_{1-1/(k\ln(k))}^{1} u^{k-2}(1 - F(x + F^{-1}(u))) \, du \tag{209}$$

$$= \int_{1-1/(k\ln(k))}^{1} u^{k-2}(1 - F(F^{-1}(u))) \, du \tag{210}$$

$$\leq \int_{1-1/(k\ln(k))}^{1} u^{k-2}(1 - u) \, du \tag{211}$$

$$\leq \frac{1}{k\ln(k)} \int_{1-1/(k\ln(k))}^{1} u^{k-2} \, du \tag{212}$$

$$= o\left( \frac{1}{k^2} \right). \tag{213}$$

Finally, in the intermediate interval $u \in [1 - \ln(k)^2/k, 1 - 1/(k\ln(k))]$, as $k \to \infty$, we see that $u \geq (1 - \ln(k)^2/k) \to 1$, $x \to 0$, and $F^{-1}(u) \to \infty$, so for sufficiently large $k$,

$$1 - F(x + F^{-1}(u)) \simeq \frac{f(x + F^{-1}(u))}{x + F^{-1}(u)} \tag{214}$$

$$\simeq \frac{f(x + F^{-1}(u))}{F^{-1}(u)} \tag{215}$$

$$\simeq \frac{f(F^{-1}(u))}{F^{-1}(u)} e^{-xF^{-1}(u)} \tag{216}$$

$$\simeq (1 - F(F^{-1}(u)))e^{-xF^{-1}(u)} \tag{217}$$

$$\simeq (1 - u)e^{-xF^{-1}(u)}, \tag{218}$$

applying the well-known approximation for the Gaussian CCDF $1 - F(w) \approx f(w)/w$ for large $w$ (for example, see Eqn. 8.2.38 from Gallager [27]), and substituting in the Gaussian PDF.

Further, since $F^{-1}(u) \to \infty$, we have that

$$1 - F(F^{-1}(u)) \simeq \frac{f(F^{-1}(u))}{F^{-1}(u)} \tag{219}$$

$$\implies \qquad 1 - u \simeq \frac{e^{-(F^{-1}(u))^2/2}}{F^{-1}(u)\sqrt{2\pi}} \tag{220}$$

$$\simeq e^{-(F^{-1}(u))^2/(2+o(1))} \tag{221}$$

$$\implies \qquad F^{-1}(u) \simeq \sqrt{-2\ln(1-u)} \tag{222}$$

$$\simeq \sqrt{2\ln(k)}, \tag{223}$$

where the last step follows from the bounds on $u$ in the intermediate interval.

Substituting the bounds from (218) and (223) into the expression for $J_2$, and applying the definition of $x$ from (197), we have,

$$J_2 \simeq \int_{1-\ln(k)^2/k}^{1-1/(k\ln(k))} u^{k-2}(1 - F(x + F^{-1}(u)))\,\mathrm{d}u \tag{224}$$

$$\simeq \int_{1-\ln(k)^2/k}^{1-1/(k\ln(k))} u^{k-2}(1 - u)e^{-xF^{-1}(u)}\,\mathrm{d}u \tag{225}$$

$$\simeq \int_{1-\ln(k)^2/k}^{1-1/(k\ln(k))} u^{k-2}(1 - u)e^{-(\theta/\sqrt{2\ln(k)})\sqrt{2\ln(k)}}\,\mathrm{d}u \tag{226}$$

$$\simeq \int_{1-\ln(k)^2/k}^{1-1/(k\ln(k))} u^{k-2}(1 - u)e^{-\theta}\,\mathrm{d}u \tag{227}$$

$$\simeq e^{-\theta} \left[ \frac{u^{k-1}}{k-1} - \frac{u^{k-2}}{k-2} \right]_{1-\ln(k)^2/k}^{1-1/(k\ln(k))} \tag{228}$$

$$\simeq \frac{e^{-\theta}}{(k-1)(k-2)}. \tag{229}$$

Combining the terms from (208), (213), and (229), and substituting back into (198), we see that

$$\Pr\left(x_{test}[\alpha] - x_{test}[\beta] > x\right) = k(k-1)J \tag{230}$$

$$= k(k-1)(J_1 + J_2 + J_3) \tag{231}$$

$$\simeq k(k-1)\left( \frac{e^{-\theta}}{(k-1)(k-2)} + o\left(\frac{1}{k^2}\right) \right) \tag{232}$$

$$\simeq e^{-\theta}. \tag{233}$$

Expressing this as a non-asymptotic lower-bound on the probability, holding for sufficiently large $k$, yields the cited result in (194).

$\square$

**Lemma B.3.** (***Lower bound on the closest feature margin when $k$ is constant***): *If $k = c_k$ for some fixed constant $c_k$, for any constant $\varepsilon > 0$, there exists a constant $\varepsilon' > 0$ such that*

$$\Pr\left(\min_{\beta,\gamma:1\leq\beta\neq\gamma\leq c_k} |x_{test}[\beta] - x_{test}[\gamma]| \geq \varepsilon'\right) \geq 1 - \varepsilon. \tag{46}$$

*Thus, with probability at least $(1 - \varepsilon)$,*

$$\min_{\beta:1\leq\beta\neq\alpha\leq k} (x_{test}[\alpha] - x_{test}[\beta]) \geq \varepsilon'. \tag{47}$$

*Here, $\alpha$ is fixed and corresponds to the index of the true class — i.e. $\alpha$ corresponds to the index of the maximum feature among the first $k$ features.*

*Proof.* Observe that,

$$\min_{1\leq\beta\neq\alpha\leq c_k} (x_{test}[\alpha] - x_{test}[\beta]) \geq \min_{1\leq\beta\neq\gamma\leq c_k} |x_{test}[\beta] - x_{test}[\gamma]|. \tag{234}$$

In other words, rather than bounding the margin between the largest and second-largest features, we will lower-bound the absolute difference between any pair of features.

Consider a particular $(\beta, \gamma)$ tuple. Observe that $x_{test}[\beta] - x_{test}[\gamma] \sim N(0, 2)$, since each feature is drawn independently from a standard Gaussian. For any $\epsilon' > 0$, we can upper-bound

$$\Pr\left(|x_{test}[\beta] - x_{test}[\gamma]| \leq \varepsilon'\right) \leq \frac{\varepsilon'}{\sqrt{\pi}} \tag{235}$$

by taking the product of the maximum value of the Gaussian pdf and the width, $2\epsilon'$, of the region we are interested in. Taking the union bound across all $(\beta, \gamma)$ tuples, we find that

$$\Pr\left(\min_{1\leq\beta\neq\gamma\leq c_k} |x_{test}[\beta] - x_{test}[\gamma]| \leq \varepsilon'\right) \leq \frac{c_k^2\varepsilon'}{\sqrt{\pi}}. \tag{236}$$

So for any given $\varepsilon > 0$, we can choose $\varepsilon' = \varepsilon\sqrt{\pi}/c_k^2$, and have that

$$\Pr\left(\min_{1\leq\beta\neq\gamma\leq c_k} |x_{test}[\beta] - x_{test}[\gamma]| \geq \varepsilon'\right) \geq 1 - \varepsilon. \tag{237}$$

$\square$

## E.2   Lower bound on $\frac{\lambda\widehat{h}_{\alpha,\beta}[\alpha]}{\max_\beta \mathsf{CN}_{\alpha,\beta}}$

Next, we will find a lower bound for survival-contamination ratio within the regime with low survival variance.

**Lemma B.4.** (***Lower bound on relative survival of true feature***): *For any fixed $\beta \in [k]$, $\beta \neq \alpha$, with $\lambda_\alpha = \lambda_\beta = \lambda$ we have with probability at least $(1 - 5/(nk))$,*

$$\lambda\widehat{h}_{\alpha,\beta}[\alpha] \geq \lambda\left(c_{10}\bar{\mu}\frac{n}{k}\sqrt{\ln(k)} - c_9(\bar{\mu}\sqrt{n}\sqrt{\ln(nk)} + \Delta_\mu \cdot n/\sqrt{k})\right), \tag{48}$$

*for universal positive constants $c_9$ and $c_{10}$.*

*Proof.* Using Corollary D.2.1, we lower bound $\widehat{h}_{\alpha,\beta}[\alpha]$ with probability at least $(1 - 5/(nk))$ as

$$\widehat{h}_{\alpha,\beta}[\alpha] = \lambda_\alpha^{-1/2}(\hat{f}_\alpha[\alpha] - \hat{f}_\beta[\alpha]) \tag{238}$$

$$= \mathbf{z}_\alpha^\top \mathbf{A}^{-1}\mathbf{y}_\alpha - \mathbf{z}_\alpha^\top \mathbf{A}^{-1}\mathbf{y}_\beta \tag{239}$$

$$\geq c_{10}\bar{\mu}\frac{n}{k}\sqrt{\ln(k)} - c_9(\bar{\mu}\sqrt{n}\sqrt{\ln(nk)} + \Delta_\mu \cdot n/\sqrt{k}). \tag{240}$$

Multiplying through by $\lambda$ gives the desired result. $\square$

From the above result, under the scalings of our bi-level model we obtain:

**Corollary B.4.1.** *Under the bi-level ensemble model 4.2, for any fixed $\beta \in [k]$, $\beta \neq \alpha$, $\lambda_\alpha = \lambda_\beta = \lambda$ if $t < 1/2$, $t < 2(q + r - 1)$ and $1 < q + r < (p + 1)/2$, with probability at least $(1 - 5/(nk))$,*

$$\lambda \widehat{h}_{\alpha,\beta}[\alpha] \geq c_{12} n^{1-q-r-t} \sqrt{\ln(k)}, \tag{49}$$

*for universal positive constant $c_{12}$.*

*Proof.* Substituting our asymptotic scalings into the results from Lemma B.4 and using the decay rate of $\bar{\mu} \asymp n^{-p}$ from Corollary C.1.1 (which we can do since $1 < q + r < (p + 1)/2$), we find that

$$\lambda \widehat{h}_{\alpha,\beta}[\alpha] \geq n^{p-q-r} \left( c_{10} \bar{\mu} \frac{n}{k} \sqrt{\ln(k)} - c_9 (\bar{\mu}\sqrt{n}\sqrt{\ln(nk)} + \Delta_\mu \cdot n/\sqrt{k}) \right) \tag{241}$$

$$= c_{10} n^{1-q-r-t} \sqrt{\ln(k)} - c_9 n^{1/2-q-r} \sqrt{\ln(nk)} - c_9 n^{2-2q-2r-t/2} \tag{242}$$

$$\geq c_{12} n^{\max(1-q-r-t, 2-2q-2r-t/2)} \sqrt{\ln(k)} \tag{243}$$

$$= c_{12} n^{1-q-r-t+\max(0, 1-q-r+t/2)} \sqrt{\ln(k)} \tag{244}$$

$$\geq c_{12} n^{1-q-r-t} \sqrt{\ln(k)}, \tag{245}$$

for an appropriately chosen universal constant $c_{12}$ and sufficiently large $n$. $\qquad\square$

Next we upper bound $\max_\beta \mathsf{CN}_{\alpha,\beta}$.

**Lemma B.5.** *(**Upper bound on contamination**): For any fixed $\beta \in [k]$, $\beta \neq \alpha$, with probability at least $(1 - 7/(nk))$,*

$$\mathsf{CN}_{\alpha,\beta} \leq c_7 (\bar{\mu}\sqrt{\frac{n}{k}} \cdot \sqrt{\ln(ndk)} + \Delta_\mu \cdot n/\sqrt{k}) \sqrt{\sum \lambda_j^2}, \tag{50}$$

*for universal positive constant $c_7$.*

*Proof.* For each $\beta$ we have,

$$\mathsf{CN}_{\alpha,\beta} = \sqrt{\left( \sum_{j \notin \{\alpha,\beta\}} \lambda_j^2 (\widehat{h}_{\beta,\alpha}[j])^2 \right)} \tag{246}$$

For $j \notin \{\alpha, \beta\}$, by Lemma D.1,

$$\left| \widehat{h}_{\beta,\alpha}[j] \right| = \left| \widehat{h}_{\alpha,\beta}[j] \right| \tag{247}$$

$$= \left| \widehat{f}_j - \widehat{g}_j \right| \tag{248}$$

$$= \left| \mathbf{z}_j^\top \mathbf{A}^{-1} \mathbf{y}_\alpha - \mathbf{z}_j^\top \mathbf{A}^{-1} \mathbf{y}_\beta \right| \tag{249}$$

$$= \left| \mathbf{z}_j^\top \mathbf{A}^{-1} (\mathbf{y}_\alpha - \mathbf{y}_\beta) \right| \tag{250}$$

$$\leq c_7 (\bar{\mu}\sqrt{\frac{n}{k}} \cdot \sqrt{\ln(ndk)} + \Delta_\mu \cdot n/\sqrt{k}), \tag{251}$$

with probability $1 - 7/(ndk)$.

So taking the union bound over all $d - 2$ terms in the expression for the contamination, we can upper-bound it as

$$\mathsf{CN}_{\alpha,\beta} \leq c_7 (\bar{\mu}\sqrt{\frac{n}{k}} \cdot \sqrt{\ln(ndk)} + \Delta_\mu \cdot n/\sqrt{k}) \sqrt{\sum \lambda_j^2}, \tag{252}$$

with probability $(1 - 7/(nk))$, the desired result. $\qquad\square$

**Corollary B.5.1.** *Under the bi-level model 4.2, in the regime $1 < q + r < (p+1)/2$, with probability at least $(1 - 7/(nk))$,*

$$\mathsf{CN}_{\alpha,\beta} \leq c_{13} n^{(1-t-p)/2 + \max(0, 3/2 - q - r) + \max(0, p/2 - q - r/2)} \sqrt{\ln(ndk)}, \tag{51}$$

*for universal positive constant $c_{13}$.*

*Proof.* Since $1 < q + r < (p+1)/2$, we can apply Corollary C.1.1 to the result from Lemma B.5 and substitute in the known scalings of various terms, to obtain

$$\mathsf{CN}_{\alpha,\beta} \leq c_7 (n^{1/2 - t/2 - p} \sqrt{\ln(ndk)} + c_4 n^{2 - p - q - r - t/2})(n^{p - q - r/2} + n^{p/2}) \tag{253}$$

$$\leq c_{13} n^{(1-t-p)/2 + \max(0, 3/2 - q - r) + \max(0, p/2 - q - r/2)} \sqrt{\ln(ndk)}, \tag{254}$$

for an appropriately chosen universal positive constant $c_{13}$.

$\square$

### E.3  Proof of Lemma B.6: Bounds on Survival Variance

Finally, we look at the error event where a competing feature has unusually high survival relative to the true feature, so it is incorrectly selected.

**Lemma B.6.** *(**Upper bound on survival variance**):  For any fixed competing feature $\beta \in [k]$, $\beta \neq \alpha$ with $\lambda_\alpha = \lambda_\beta$, we have with probability at least $(1 - 15/(nk))$,*

$$\frac{\widehat{h}_{\alpha,\beta}[\alpha] - \widehat{h}_{\beta,\alpha}[\beta]}{\widehat{h}_{\alpha,\beta}[\alpha]} \leq \frac{2c_9(\bar{\mu}\sqrt{n}\sqrt{\ln(nk)} + \Delta_\mu \cdot n/\sqrt{k})}{c_{10}\bar{\mu}\frac{n}{k}\sqrt{\ln(k)} - c_9(\bar{\mu}\sqrt{n}\sqrt{\ln(nk)} + \Delta_\mu \cdot n/\sqrt{k})}, \tag{52}$$

*for universal positive constants $c_9$ and $c_{10}$.*

*Proof.* We first consider the numerator of the LHS of (52). By Lemma D.2, with probability at least $(1 - 5/(nk))$,

$$\widehat{h}_{\alpha,\beta}[\alpha] = \lambda_\alpha^{-1/2}(\hat{f}_\alpha[\alpha] - \hat{f}_\beta[\alpha]) \tag{255}$$

$$= \mathbf{z}_\alpha^\top \mathbf{A}^{-1} \mathbf{y}_\alpha - \mathbf{z}_\alpha^\top \mathbf{A}^{-1} \mathbf{y}_\beta \tag{256}$$

$$\leq \bar{\mu}(\mathbb{E}[\mathbf{z}_\alpha^\top \mathbf{y}_\alpha] - \mathbb{E}[\mathbf{z}_\alpha^\top \mathbf{y}_\beta]) + c_9(\bar{\mu}\sqrt{n}\sqrt{\ln(nk)} + \Delta_\mu \cdot n/\sqrt{k}). \tag{257}$$

Similarly, with probability at least $(1 - 5/(nk))$,

$$\widehat{h}_{\beta,\alpha}[\beta] = \lambda_\beta^{-1/2}(\hat{f}_\beta[\beta] - \hat{f}_\alpha[\beta]) \tag{258}$$

$$= \mathbf{z}_\beta^\top \mathbf{A}^{-1} \mathbf{y}_\beta - \mathbf{z}_\beta^\top \mathbf{A}^{-1} \mathbf{y}_\alpha \tag{259}$$

$$\geq \bar{\mu}(\mathbb{E}[\mathbf{z}_\beta^\top \mathbf{y}_\beta] - \mathbb{E}[\mathbf{z}_\beta^\top \mathbf{y}_\alpha]) - c_9(\bar{\mu}\sqrt{n}\sqrt{\ln(nk)} + \Delta_\mu \cdot n/\sqrt{k}). \tag{260}$$

By symmetry,

$$\mathbb{E}[\mathbf{z}_\beta^\top \mathbf{y}_\beta] = \mathbb{E}[\mathbf{z}_\alpha^\top \mathbf{y}_\alpha] \tag{261}$$

$$\mathbb{E}[\mathbf{z}_\beta^\top \mathbf{y}_\alpha] = \mathbb{E}[\mathbf{z}_\alpha^\top \mathbf{y}_\beta]. \tag{262}$$

Thus with probability at least $(1 - 10/(nk))$,

$$\widehat{h}_{\alpha,\beta}[\alpha] - \widehat{h}_{\beta,\alpha}[\beta] \leq 2c_9(\bar{\mu}\sqrt{n}\sqrt{\ln(nk)} + \Delta_\mu \cdot n/\sqrt{k}). \tag{263}$$

Using Corollary D.2.1 to lower-bound the denominator of the LHS of (52), we obtain with probability at least $(1 - 15/(nk))$

$$\frac{\widehat{h}_{\alpha,\beta}[\alpha] - \widehat{h}_{\beta,\alpha}[\beta]}{\widehat{h}_{\alpha,\beta}[\alpha]} \leq \frac{2c_9(\bar{\mu}\sqrt{n}\sqrt{\ln(nk)} + \Delta_\mu \cdot n/\sqrt{k})}{c_{10}\bar{\mu}\frac{n}{k}\sqrt{\ln(k)} - c_9(\bar{\mu}\sqrt{n}\sqrt{\ln(nk)} + \Delta_\mu \cdot n/\sqrt{k})}. \tag{264}$$

$\square$

We can apply Corollary C.1.1 to simplify our results from Lemma B.6 in the asymptotic regime for the bi-level model.

**Corollary B.6.1.** *Under the bi-level ensemble model 4.2, for any fixed $\beta \in [k]$, $\beta \neq \alpha$, if $t < 1/2$, $t < 2(q + r - 1)$, and $1 < q + r < (p + 1)/2$, with probability at least $(1 - 15/(nk))$,*

$$\frac{\widehat{h}_{\alpha,\beta}[\alpha] - \widehat{h}_{\beta,\alpha}[\beta]}{\widehat{h}_{\alpha,\beta}[\alpha]} < n^{-u}, \tag{53}$$

*for large enough $n$ for some fixed $u > 0$.*

*Proof.* Substituting, using Corollary C.1.1, in the regime where $1 < q + r < (p + 1)/2$ and $t < 1/2$, we find that

$$\frac{\widehat{h}_{\alpha,\beta}[\alpha] - \widehat{h}_{\beta,\alpha}[\beta]}{\widehat{h}_{\alpha,\beta}[\alpha]} \leq \frac{2c_9(n^{1/2-p}\sqrt{\ln(nk)} + c_4 n^{2-p-q-r-t/2})}{c_{10}n^{1-p-t}\sqrt{\ln(k)} - c_9(n^{1/2-p}\sqrt{\ln(nk)} + c_4 n^{2-p-q-r-t/2})} \tag{265}$$

$$\leq \frac{2c_9}{c_{10}} \cdot \frac{n^{1/2}\sqrt{\ln(nk)} + c_4 n^{2-q-r-t/2}}{n^{1-t} - (c_9/c_{10})n^{1/2}\sqrt{\ln(n)} + (c_9 \cdot c_4/c_{10})n^{2-q-r-t/2}} \tag{266}$$

$$\leq c_{14} \frac{n^{1/2}\sqrt{\ln(nk)} + n^{2-q-r-t/2}}{n^{1-t}} \tag{267}$$

$$\leq c_{14} n^{\max(t-1/2, t/2+1-q-r)}\sqrt{\ln(nk)}, \tag{268}$$

for sufficiently large $n$ and an appropriate choice of positive constant $c_{14}$. Thus, if $\max(t - 1/2, t/2 + 1 - q - r) < 0$, our quantity of interest tends to zero at a polynomial rate as $n \to \infty$, completing the proof. $\square$

# F  Conjectured Looseness of Bound

In (155) in the proof of Lemma D.1, we upper bound $\mathbf{z}_j^\top \mathbf{\Delta} A_{inv} \mathbf{\Delta} y$ using the Cauchy-Schwarz inequality as

$$|\mathbf{z}_j^\top \mathbf{\Delta} A_{inv} \mathbf{\Delta} y| \leq \|\mathbf{z}_j\|_2 \|\mathbf{\Delta} A_{inv} \mathbf{\Delta} y\|_2 \tag{269}$$

$$\leq \|\mathbf{\Delta} A_{inv}\|_{op} \|\mathbf{z}_j\|_2 \|\mathbf{\Delta} y\|_2 \tag{270}$$

$$\leq \Delta_\mu \|\mathbf{z}_j\|_2 \|\mathbf{\Delta} y\|_2. \tag{271}$$

This results in a high-probability bound of the order $\Delta_\mu n/\sqrt{k}$. Essentially this bound fears that $\mathbf{\Delta} A_{inv}$ can, in worst case, align $\mathbf{z}_j$ and $\mathbf{\Delta} y$ to be in the same direction. However, since there is only a weak dependence between $\mathbf{\Delta} A_{inv}$ and $\mathbf{z}_j$ and $\mathbf{\Delta} y$ this bound is likely overly cautious. We conjecture that this bound is loose by a factor $\sqrt{n}$. Why do we conjecture this? If we ignored the dependency of $\mathbf{\Delta} A_{inv}$ on $\mathbf{z}_j$ and $\mathbf{\Delta} y$ and blindly applied the Hanson-Wright inequality (with the $\mathbf{M}$ matrix introduced as in Appendix D.1.1 to leverage the fact that $\mathbf{\Delta} y$ is mostly zeros) then we would obtain a high-probability upper bound of the form $\Delta_\mu \sqrt{n/k}$ (ignoring the logarithmic factors).

Assuming this tighter conjectured bound holds and similarly assuming an analogously tighter bound for $|\mathbf{z}_\alpha^\top \mathbf{\Delta} A_{inv} \mathbf{\Delta} y|$ in Appendix D.2 and following through with the rest of our analysis, we obtain the conjectured sufficient conditions for good generalization as in Equation (23) from Conjecture 6.1 for the regime $q + r > 1$.

It turns out that whenever the survival/contamination ratio grows at a polynomial rate $n^v$ for $v > 0$ then the survival variation term also shrinks at a polynomial rate $n^{-u}$ for $u > 0$. Thus ensuring the survival/contamination ratio is large enough (i.e. the number of classes is not too large relative to the level of favoring of potentially true features) is key to obtaining good generalization.

Although we focus on the regime $q + r > 1$ in our work, our proof technique is also applicable to the regime $q + r < 1$, i.e where regression works and by grinding through the math for this setting we should be able to get sufficient conditions for good generalization here as well. The survival in the multi-class setting in the regime $q + r < 1$ will scale roughly as $1/k$ due to the fewer positive training examples per class instead of behaving like the constant $\sqrt{2/\pi}$ as was the case for binary classification (Lemma 32, [56]). Moreover, Lemma 34 from Muthukumar et al. [56] shows that for

the binary classification setting the contamination scales as $n^{-\min(p-1,1-r)/2}$ when $q + r \leq 1$. In the multiclass setting the contamination will be lower by a factor of $\sqrt{k}$ and substituting this in our error analysis we obtain Conjecture 6.1 for the regime $q + r < 1$.

Finally, we believe that we can adapt our analysis from the Proof of Theorem 5.1 in Appendix B to write a set of sufficient conditions for poor generalization. The primary condition for this would be for the relevant survival/contamination ratio to go to zero. We conjecture that computing conditions on $p, q, r, t$ under which this occurs results in the converse result in the form of sufficient conditions for poor generalization present in Conjecture 6.1. Intuitively, if the survival/contamination ratio goes to zero, then the contamination can with significant probability flip the sign of a comparison involving the score that should be winning — this parallels the way that the converse is proved in Muthukumar et al. [56] for binary classification.

## G  Scaling parameters with the number of positive training examples per class

From our results in Figure 2 we observed that as the number of classes $k$ increases (i.e. larger values of $t$), the region where multiclass classification generalizes well shrinks. A justification for this is when the number of classes $k$ increases while the number of training points $n$ stays constant, we have fewer positive training examples from each class, and this makes the task harder.

To see if the reduced number of positive training examples is indeed the dominant effect, we can explore what happens if we increase the number of total training points to compensate for this effect? Instead of scaling all parameters with the total number of training points, what happens if we scale them with the number of positive training examples per class?

Let $N = n^b$ be the new number of training points for some $b > 1$, while rest of the parameters in the bi-level model scale as before. We have,

$$N = n^b \tag{272}$$

$$d = n^p = N^{p/b} \tag{273}$$

$$s = n^r = N^{r/b} \tag{274}$$

$$a = n^{-q} = N^{-q/b} \tag{275}$$

$$k = c_k n^{-t} = c_k N^{-t/b}. \tag{276}$$

We can interpret this as our standard setup, albeit parameterized by $N$, rather than $n$. To keep the model well-defined we require the following:

- $b < p$, to ensure we are still overparameterized;

- $r < b$, to ensure the number of favored features does not exceed the total number of training points;

- $q < p - r$ to ensure we are actually favoring the first $s$ features.

For this setup, Theorem 5.1 states that the probability of misclassification tends to zero if

$$\frac{t}{b} < \min\left(\frac{r}{b}, 1 - \frac{r}{b}, \frac{p}{b} + 1 - 2\left(\frac{q}{r} + \frac{r}{b}\right), \frac{p}{b} - 2, \frac{2q}{b} + \frac{r}{b} - 2\right) \tag{277}$$

$$\frac{q}{b} + \frac{r}{b} > 1. \tag{278}$$

Rearranging, we obtain the condition

$$t < \min(r, b - r, p + b - 2(q + r), p - 2b, 2q + r - 2b) \tag{279}$$

$$q + r > b. \tag{280}$$

To hold the number of training samples per class fixed we can set $b = t + 1$, so the ratio $N/k$ becomes constant. Doing so, we obtain the following sufficient conditions for good generalization:

$$t < \min\left(r, \frac{p-2}{3}, \frac{2q+r-2}{3}\right) \tag{281}$$

$$0 < 1 - r \tag{282}$$

$$0 < p + 1 - 2(q + r) \tag{283}$$

$$t < q + r - 1. \tag{284}$$

Additionally for the model to be well defined we require $t < p - 1$. (The other conditions $r < t + 1$ and $q < p - r$ for model to be well defined are automatically satisfied if the above conditions for good generalization are satisfied).

If we assume Conjecture 6.1 then a set of sufficient conditions for good generalization is:

$$0 \leq p + 1 - 2(q + r) \tag{285}$$

$$r < 1 \tag{286}$$

$$t < r \tag{287}$$

$$t < p - 1. \tag{288}$$

The first two conditions must be satisfied for binary classification problem to generalize well and thus for multi-class classification to succeed in this setting we need to ensure binary classification succeeds. The condition $t < r$ arises because if we don't favor the features used in the comparison while assigning class labels then we have no hope of succeeding in overparameterized settings. The condition $t < p - 1$ ensures that the problem is overparameterized. If any of these conditions is not met then the probability of classification error will tend to 1.

Figure 4 visualizes the conjectured regimes for this alternative setup where the number of positive training examples per class is held fixed as we vary the number of classes for fixed values of $p$ and $q$. In the white region, our model is not well defined. Note that in subfigure (a), the limiting factor to the model being well defined is the inequality $r < 1 + t$ (we must have more training examples than favored features) while in subfigure (b), the limiting factor for the model being well defined in the right-hand boundary is the inequality $r < p - q$ (we must put a larger weight on the features we favor as compared to those that we do not favor). In subfigure(b) we see that the top boundary for the model being well defined is the inequality $t < p - 1$ which is necessary for the problem to be overparameterized and support the existence of interpolating solutions. Further, the right-hand bound for good generalization in subfigure (a) corresponds to the inequality $r < 1$ while in subfigure (b) it corresponds to $p + 1 > 2(q + r)$. The left-hand boundary for good generalization in both figures is the inequality $t < r$, which reflects the fact that for MNI-based classification to succeed, all the features defining the classes must be favored.

It is interesting to note that when we add more training points so as to increase the number of positive examples, we are effectively decreasing the level of overparameterization in the problem. We know from Nakkiran [58] that adding training data in a way that reduces overparameterization can sometimes make performance worse instead of better. However, in the deeply overparameterized setting of the bi-level models explored here, this effect is counteracted by the survival benefits of having more positive examples — in effect, reducing the overall level of overparameterization reduces the shrinkage induced by the regularizing effect of overparameterization. This reduction in shrinkage compensates for the $\frac{1}{k}$ hit to survival induced by the larger number of classes.

## H   Additional related work

### H.1   Comparisons to Wang et al. [77]

Recent work [77] provides an analysis of the generalization error of the minimum-norm interpolation of one-hot labels for multiclass classification with Gaussian features. Using the bi-level model, the authors present parameter regimes where multiclass classification error goes to zero asymptotically. While our work has many similarities in terms of model and problem setting, there are some key differences.

The first key difference is in how the training data is generated. In this paper, we assume the true label of a point is generated based on which of the first $k$ dimensions is the largest, while Wang et al.

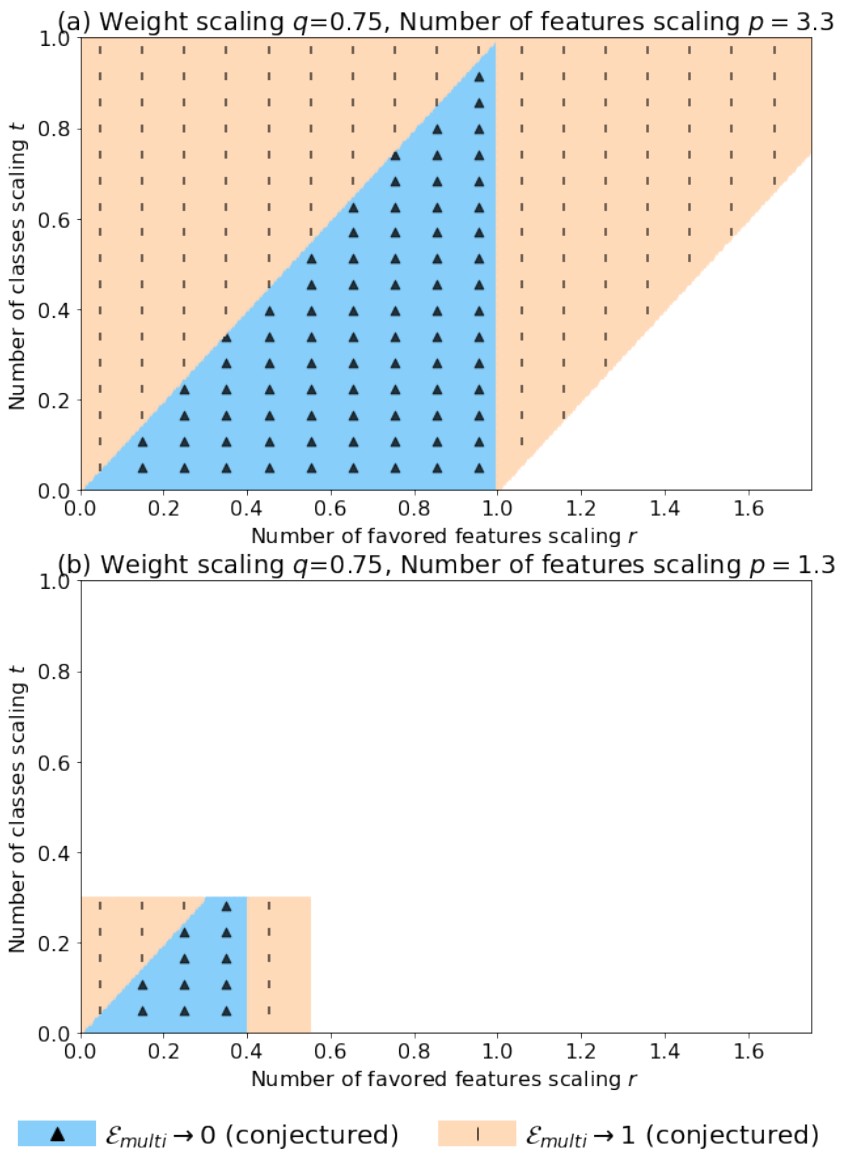

Figure 4: Visualization of the conjectured bi-level classification regimes when we scale everything with the number of positive training examples per class, instead of with the total number of training points.

[77] consider a Gaussian mixture model and a multinomial logistic model where the true labels have some randomness even conditioned on the first $k$ dimensions. Like us, however, they also consider the case of orthogonal classes.

Second, we consider the asymptotic case where the number of classes, $k$ scales with the number of training points as $k = cn^t$ for some positive integer $c$ and non-negative real $t$. The work in Wang et al. [77] considers only the finite classes setting i.e. $t = 0$ in our model. The error analysis technique employed by us here in the form of a typicality-style argument featuring the feature margin (difference between the largest and second largest feature) is much tighter than the method employed in Wang et al. [77] and allows us to compute regimes where multiclass classification succeeds even when $t > 0$. A straight substitution into the analysis from Wang et al. [77] does not work since that analysis is too loose for this setting. Furthermore, in our expressions for survival and contamination

(Lemmas B.4 and B.5) we compute an exact dependence on $k$.[18] The expressions from Wang et al. [77] don't compute this exact dependence because it is not required for their purposes. By using our novel analysis technique we are able to elucidate the challenges posed by fewer positive training examples per class in the multiclass setting and provide sufficient conditions for generalization when number of classes scales with the number of training points.

An equivalence between the solution obtained by minimum-$\ell_2$-norm interpolation on the adjusted zero-mean one-hot encoded labels that we perform in our approach (Equation (9)) and the solution obtained by other training methods has been established in [77]. In particular the minimum-norm interpolating solution is typically identical to the solution obtained via one-vs-all SVM and multi-class SVM (and thus gradient descent on cross-entropy loss due to its implicit bias [36, 68], under sufficient overparameterization. From Wang et al. [77], the sufficient conditions for the equivalence of solutions are,

$$\frac{\sum_{j=1}^{n} \lambda_j}{\lambda_1} > C_1 k^2 n \ln(kn), \tag{289}$$

$$\frac{(\sum_{j=1}^{n} \lambda_j)^2}{\sum_{j=1}^{n} \lambda_j^2} > C_2(\ln(kn) + n), \tag{290}$$

where $C_1, C_2$ are positive constants. Under our bi-level model (Definition 4.2) these conditions translate to:

$$q + r > 2t + 1, \tag{291}$$
$$2p - \max(2p - 2q - r, p) > 1. \tag{292}$$

This can be succinctly expressed as,

$$0 < t < \frac{q + r - 1}{2}. \tag{293}$$

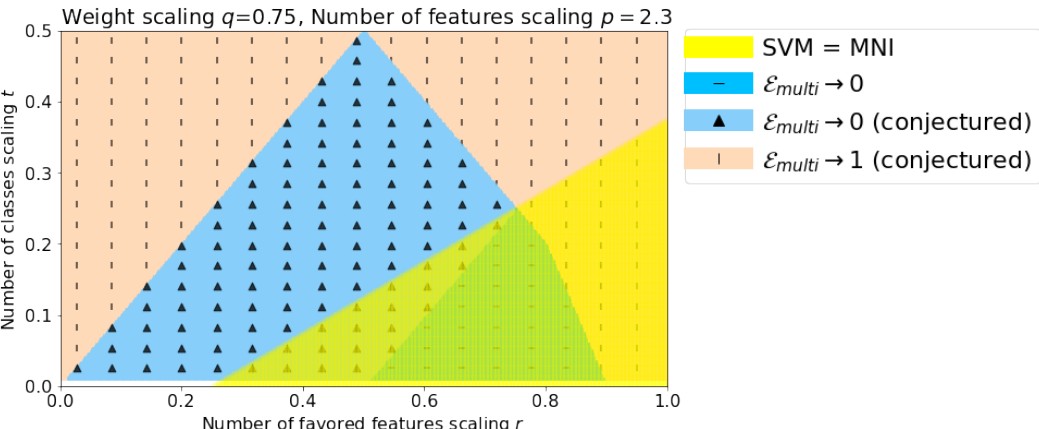

Figure 5: Visualization of regime where SVM solution is identical to MNI solution.

In Figure 5 we plot our provable as well as conjectured regimes alongside the regime where minimum-norm interpolation leads to same solution as other training methods. Notice the overlap. Thus our analysis is not limited only to the minimum-norm interpolator but holds for other training methods when the problem is sufficiently overparameterized. In this sense, the results in Wang et al. [77] and the present paper should be read together to tell a more full story of overparameterized multiclass classification.

---

[18]In particular, our analysis here brings out the fact that multiclass training data becomes less informative per training sample as the number of classes increases. This results in a $\frac{1}{k}$ scaling term in survival and a $\frac{1}{\sqrt{k}}$ scaling in contamination. It is this effect that makes it possible in some regimes for the contamination from other favored features to dominate — whereas in the case of binary classification, it is always the contamination from unfavored features that dominates.

## H.2 Comparisons to Muthukumar et al. [56]

The work in Muthukumar et al. [56] provides an analysis of the binary classification and regression problem with Gaussian features in the overparameterized regime and shows that binary classification is easier than regression by proving the existence of a regime in a bi-level model where binary classification generalizes well but regression does not. In this work we use a similar bi-level model and the signal-processing inspired concepts of survival and contamination in our proofs but the nature of the training data in the multiclass classification problem is the key challenge and complicates our analysis considerably. Since the true class labels are generated by comparing $k$ features, we no longer have independence of the class label $y$ with any of these features. This is relevant when we compute bounds on the the term $z_\alpha^\top \mathbf{A}^{-1}(y_\alpha - y_\beta)$ an integral part of our survival quantity (Equations (97),(98) from Lemma D.2), since the Hanson-Wright inequality is no longer applicable directly as was the case for the binary classification problem in prior work (Appendix D.3.1 of Muthukumar et al. [56]). Working through these challenges, we prove that the multiclass problem is fundamentally different from (and harder than) than the binary problem due to the effect of fewer informative samples (positive training examples) per class. In particular we show via dominant terms from Lemmas B.6 and B.5 that, as we increase the number of classes $k$, survival shrinks as $1/k$ while contamination shrinks only as $1/\sqrt{k}$. Thus the survival/contamination ratio which plays a key role in the expression for classification error decreases as $1/\sqrt{k}$ in the multiclass setting as we increase $k$. Thus, for good generalization we need to ensure number of classes is not too large in addition to having sufficient favoring of true features.

## I Experimental results

Theorem 5.1 is proved rigorously and so we know that the asymptotic result is true. Underlying the result is the analysis of survival (how strongly is the true feature underlying this class represented in the learned score) and contamination (what is the standard deviation of the contamination in predictions that comes from learning nonzero coefficients to features that have nothing to do with this class). Multiclass classification asymptotically succeeds when the survival dominates the contamination.

In Fig. 6, we plot experimental results using the bi-level ensemble model (Definition 4.2) for a setting where regression does not work but multiclass classification is conjectured to work. We plot quantities from Equation (22) in our error analysis. From subfigures (a),(b) and (c) we observe that while both survival and contamination are decreasing as we increase $n$, the survival/contamination ratio increases. The survival/contamination ratio growing with $n$ is important for correct classification. The trend is very clear from the experimental results and indicates that continuing to grow $n$ (together with the number of classes $k$, the number of features $d$, the number of favored features $s$, and the level of favoring as per our bi-level idealized model) would result in ever improving performance. Furthermore, we see that the empirical slope of these quantities on a log-log plot (and thus the power-law scaling of these quantities with respect to $n$) agree with the theoretical slopes calculated based on our conjecture.

Subfigure (d) plots the binary classification error when only trying to distinguish between the true class and one other particular class. (In this experiment, the true class was determined by feature 1. We calculate the binary error as the probability of misclassifiying a point from class 1 as belong to class 2 when we only compare the scores for class 1 and class 2.) We see that this error clearly decreases as we increase $n$. One way of thinking about successful multiclass classification is that the true class must win such pairwise competitions against all competing classes. Finally, subfigure (e) plots the total multiclass misclassification error overlaid with the number of classes. Here too we see a downward trend in classification error as $n$ increases, even though we would have to go to significantly larger $n$ than our compute could handle to see this error probability drop to very low values. Notice the integer effects arising from the number of classes $k$ sometimes not growing with $n$ that result in small upward spikes in the classification error.

## J Comment on empirical eigenstructures of feature matrices

It is well known (for instance Remark 6 from Muthukumar et al. [56] that cites [78]) that for a spiked covariance model when the ratio of the top to the bottom eigenvalues grows as $\Omega(d/n)$, the top $s$ eigenvalues can be estimated reliably from samples, even when the number of training samples $n$ is less than the number of features $d$. The ratio of the top to the bottom eigenvalue in our bilevel model

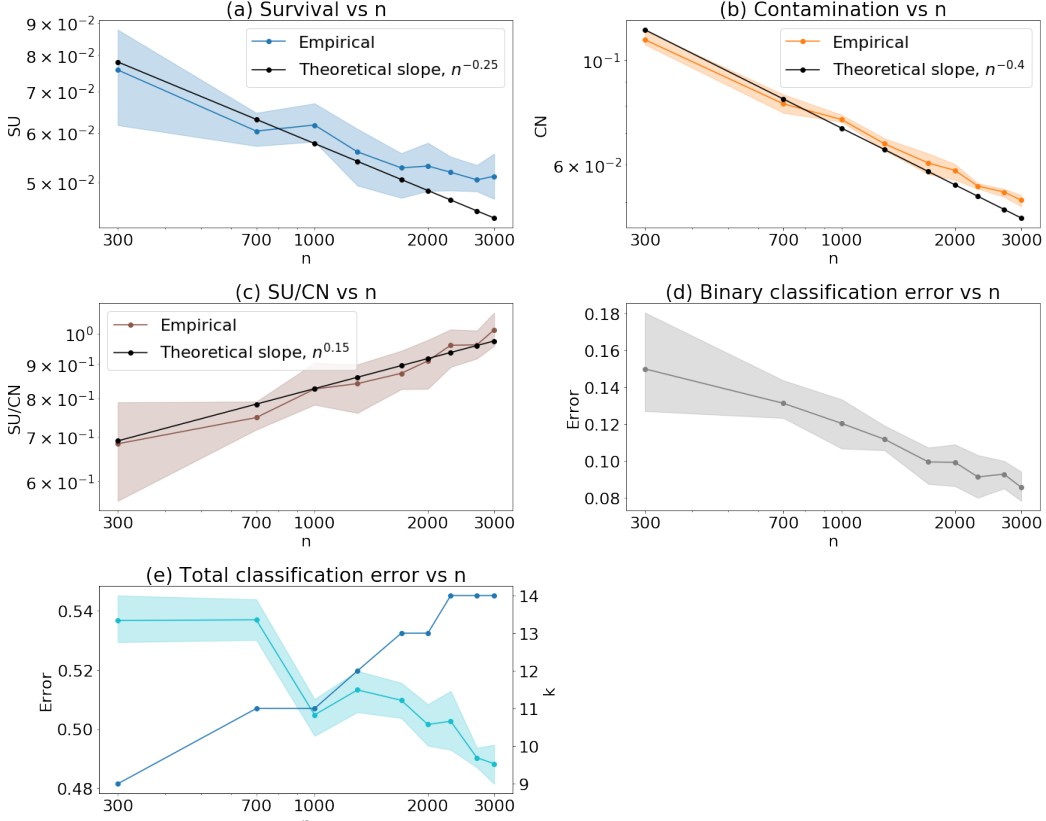

Figure 6: Experimental results using the bi-level ensemble model with $p = 1.5, q = 0.55, r = 0.5, t = 0.2$. Here, the number of training samples $n$ varies from 300 to 3000 and the number of classes is computed as $k = \lfloor 3n^t \rfloor$ and varies from 9 to 14. We calculated the classification errors over a batch size of 10000, and ran 10 trials. The plots show the mean plotted with error bars corresponding to the $10^{th}, 90^{th}$ percentile values. We also plot the theoretical slopes for survival, contamination and the survival/contamination ratio based on our conjecture and notice that it closely matches the empirical slope of the quantities when plotted on a log-log scale. Notice that jaggedness in the plots is often due to integer effects as $k$ grows or does not grow with $n$.

scales as

$$\frac{\frac{ad}{s}}{\frac{(1-a)d}{d-s}} = n^{p-q-r}, \tag{294}$$

and when $q+r < 1$, this ratio is larger than $d/n = n^{p-1}$. Fig. 7 shows empirical results of estimating the eigenvalues via the singular value decomposition of the training feature matrices. The visual distinction is quite striking. In the regime $q + r > 1$, the SVD of the training features matrix (and thus the empirical covariance matrix's eigenvalues) does not reveal that there are actually $s$ favored features in the data. By contrast, in the regime $q + r < 1$, the SVD clearly shows an eigenvalue gap that reveals exactly what $s$ is.

Theorem 5.1 shows that when $q + r > 1$ and there is not enough structure in the feature matrix to reliably estimate the top eigenvalues of the feature covariance matrix features, multiclass classification can still succeed for interpolating solutions as long as the number of classes does not increase too fast. It is because we are in the regime $q + r > 1$ that new techniques for analysis had to be developed in this paper, and the gap between the regime where we can prove the results and our conjectured results points interestingly to where there is a need for even better technique.

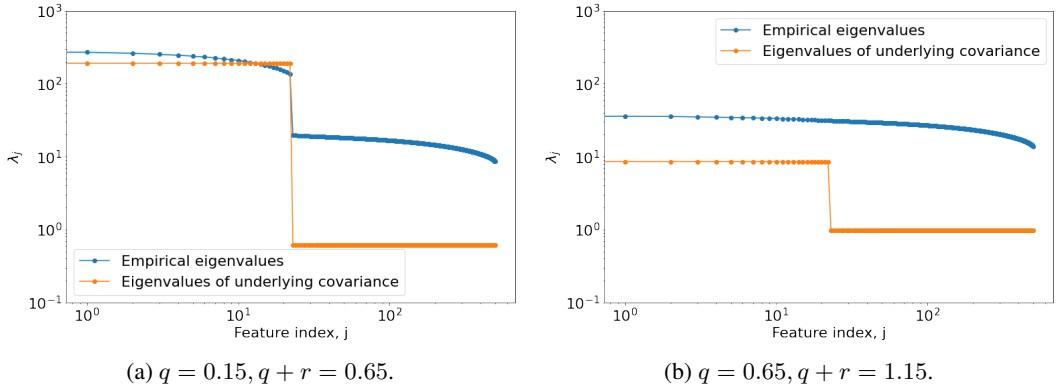

(a) $q = 0.15, q + r = 0.65$.  (b) $q = 0.65, q + r = 1.15$.

Figure 7: Estimating the eigenvalues of the covariance matrix of features empirically. Here $n = 400$ and the feature covariance structure follows the bi-level model with parameters $p = 1.5, r = 0.5$. Thus $d = 8001$ and $s = 21$ for both (a) and (b). The difference between (a) and (b) is in the level of favoring of favored features: with $q = 0.15$, (a) favors them more than (b) does with $q = 0.65$. In the regime where regression works, $q + r \leq 1$, we are able to accurately estimate the top $s$ eigenvalues. In the regime where regression fails, $q + r > 1$, we are unable to estimate the top $s$ eigenvalues accurately. The blue curve plots the estimated eigenvalues and the shaded region corresponds to the 10-90 percentile of the estimated values over 20 trials. Note, that there is only very small deviation across trials.

## K  A heuristic derivation of the main result

There is a significant amount technical work required in this paper to prove the main results, and Section F provides a conjecture for what the result should be, based on a more detailed look at our proof and where we suspect looseness. However, it is also possible to derive/conjecture the main result by another heuristic method in the style of Appendix B of Muthukumar et al. [55] and Appendix A of Muthukumar et al. [56]. At the heart of this heuristic method is an asymptotic linear-algebraic perspective driven by a few principles:

- When generalized linear models are sufficiently overparameterized, they tend to behave like the Fourier features model does on one-dimensional regularly spaced training data.

- If the favored features are approximately orthogonal on the training data, we can treat them as though they were exactly orthogonal on the training data as long as the number of favored features is sublinear in $n$, the number of training points.

- In the heavily overparameterized regime, for the bi-level model considered in this paper as well as in Muthukumar et al. [56] with $s$ favored features, $n$ training points, and $d$ total features, min-norm-interpolation will behave as though there are $\frac{d}{n}$ exact aliases among the unfavored features for each of the favored features.

- Similarly, for this heavily overparameterized bi-level model, min-norm-interpolation will behave as though the unfavored features also provide another $n - s$ effectively orthogonal directions on the training data that also have $\frac{d}{n}$ exact aliases — all of these among the unfavored features.

- Any nonlinear function of a small group of training features can be viewed as having a linear part that is essentially captured by its projection onto that group of training features plus another component which is assumed to behave like "white noise" — equally spread across the $n$ dimensions (assuming the size of the group is much smaller than $n$).

- On a test point, the different underlying features behave like uncorrelated random variables whose means are 0 and whose variances are proportional to the magnitude-squared of their coefficient times their inherent weighting squared.

These heuristics, together with other standard approximations like $1 + |x| \approx |x|$ when $|x| \gg 1$ and $1 + |x| \approx 1$ when $|x| \ll 1$ permit relatively simple calculations to be used together with the survival/contamination style of analysis to predict when regression and classification type problems

will succeed. This is done in Appendix B of Muthukumar et al. [56] for binary classification. Here, it is useful to recap the key insights from that paper in the context of the same bi-level model used here:

- When the underlying score function that we would like to learn is a linear combination of the favored features, the survival captures the extent to which that particular linear combination is present in the learned score. Without loss of generality, because all favored features are equally favored, it suffices to consider a desired score that is purely one of the favored features, normalized to have unit variance. In this case, the coefficient learned for that (normalized) feature represents the survival.

- The contamination captures everything else that is learned — all the learned coefficients of features other than the true feature — and is measured in terms of the standard deviation of the predictions due to those (falsely) learned coefficients.

- For learning in a regression problem to asymptotically generalize, the survival must tend to $1$ and the contamination must tend to $0$. This is because the score function itself must be learned in a mean-square-error sense.

- For learning in a binary classification problem to asymptotically generalize, all we need to get right is the sign of the learned score. To get the sign right, it suffices to have the ratio of survival to contamination go to infinity. This can happen even if both the survival and contamination tend to $0$, as long as they do so at the right rates relative to each other.

## K.1 Understanding what we learn from multiclass training data

Using the above heuristics and reusing calculations already done in Appendix A of Muthukumar et al. [56], we can see how survival and contamination asymptotically behave for multi-class training data. For class $m$, we consider the score function we learn by interpolating the zero-mean one-hot encoding for the $n$ training points as represented by $\mathbf{y}_m$. Within $\mathbf{y}_m$, we are going to assume that there are exactly $\frac{n}{k}$ positive examples for the $m$-th class, and each of those is represented with a $1 - \frac{1}{k}$ in the appropriate position of $\mathbf{y}_m$. There are also $n - \frac{n}{k}$ negative examples for the $m$-th class, each represented with a $-\frac{1}{k}$ in the appropriate position.

The total "energy" (norm-squared) in this vector $\mathbf{y}_m$ is thus

$$\frac{n}{k}\left(1 - \frac{1}{k}\right)^2 + \left(n - \frac{n}{k}\right)\frac{1}{k^2} = \frac{n}{k} - \frac{n}{k^2}$$
$$\approx \frac{n}{k}$$

where the final approximation is due to $k$ asymptotically growing as $n^t$ to infinity.

We can understand the process of min-norm interpolation as proceeding in two conceptual steps. First, this $n$-dimensional vector $\mathbf{y}_m$ is decomposed into the $s$ orthogonal directions represented by the $s$ favored features and the $n - s$ unfavored synthetic directions according to the heuristic above. For this step, the level of favoring does not matter. After that, the level of favoring determines precisely how each of those directions is split across the representative favored feature (if any) and its $\frac{d}{n}$ unfavored aliases.

### K.1.1 Survival

How much of $\mathbf{y}_m$ ends up going into the direction representing the true favored feature $m$? Because of orthogonality, we can simply look at the correlation between $\mathbf{y}_m$ and a normalized vector in the direction of $\mathbf{z}_m$. We can approximate the standard Gaussian in $\mathbf{z}_m$ which wins a competition with $k - 1$ other iid standard Gaussians as being a constant $\sqrt{\ln k}$, which as compared to the polynomial scalings relevant here, might as well just be the constant $1$. With that, we can consider all the other entries of $\mathbf{z}_m$ as being basically their mean, which by the same logic is essentially $\frac{-1}{k}$. This gives a total correlation of

$$\frac{1}{\sqrt{n}}\left(\frac{n}{k} \cdot \left(1 - \frac{1}{k}\right) \cdot 1 + \left(n - \frac{n}{k}\right)\left(-\frac{1}{k}\right)\left(-\frac{1}{k}\right)\right) = \frac{1}{\sqrt{n}}\left(\frac{n}{k} - \frac{n}{k^3}\right) \tag{295}$$

$$\approx \frac{\sqrt{n}}{k} \tag{296}$$

with a normalized vector $\frac{\mathbf{z}_m}{\|\mathbf{z}_m\|}$.

Notice that if we had done this correlation with $\mathbf{z}_m$ itself instead of $\mathbf{y}_m$, we would have gotten simply $\sqrt{n}$. This shows how multiclass training data immediately reduces the survival by an asymptotic factor of $k$ relative to noise-free regression training data.

### K.1.2 Contamination

To understand contamination, we simply break it down into two sources: the other $s - 1 \approx s$ favored features and the unfavored features.

Here, we leverage the heuristic that the total $\frac{n}{k}$ energy of the training labels $\mathbf{y}_m$ has to be split across the true feature $m$, the other $k$ label-defining features that are not the true feature, the other $s - k \approx s$ favored features, and the unfavored features.

For positive training examples for class $m$, the other label-defining features have a mean value of $-\frac{1}{k}$ since they are not the max. For negative training examples of class $m$, the mean value for other label-defining features is essentially zero. The consequence of this (by a calculation exactly analogous to (296)) is that the projected correlation for these is like $-\frac{\sqrt{n}}{k^2}$ which might as well be zero.

How much energy is left in $\mathbf{y}_m$ after we remove the components along the $k$ label-defining directions that could have any linear relationship to it?

$$\frac{n}{k} - \left(\frac{\sqrt{n}}{k}\right)^2 - (k - 1)\left(-\frac{\sqrt{n}}{k^2}\right)^2 \approx \frac{n}{k} - \frac{n}{k^2} - \frac{n}{k^3} \tag{297}$$

$$\approx \frac{n}{k}. \tag{298}$$

Asymptotically, all the energy is still left. This can now be divided equally across the $n - k \approx n$ orthogonal directions by the heuristic.

This means that a fraction $\frac{s}{n}$ of this ends up as contamination in the favored features, which is a total energy of $\frac{s}{n} \cdot \frac{n}{k} = \frac{s}{k}$.

Meanwhile, there are $n - s$ other directions that are only represented by the unfavored features. This has total energy $\frac{n-s}{n} \cdot \frac{n}{k} = \frac{n-s}{k} \approx \frac{n}{k}$.

### K.1.3 When does survival dominate contamination?

There are qualitatively two kinds of contamination: coming from favored features and coming from unfavored features.

For favored features (like the true feature), there is a further split that happens as the min-norm-interpolation solution splits the coefficients themselves across the favored feature and its effectively $\frac{d}{n} = n^{p-1}$ unfavored aliases. Adapting the notation in Appendix A of Muthukumar et al. [56], let's say a fraction $\alpha$ goes onto the favored feature itself. Equation (23) in Muthukumar et al. [56] tells us that

$$\alpha \approx \begin{cases} 1 & \text{if } q < 1 - r \\ n^{-(q-(1-r))} & \text{if } q > 1 - r \end{cases}. \tag{299}$$

Consequently, the actually survived signal scales as $\alpha \frac{\sqrt{n}}{k}$ while in those same units, the standard deviation of favored contamination scales as $\alpha\sqrt{\frac{s}{k}}$. From this we immediately see that the $\alpha$ cancel and the condition for classification working (as far as the favored features are concerned) is that

$$\frac{\sqrt{n}}{k} \gg \sqrt{\frac{s}{k}} \tag{300}$$

which implies

$$k \ll \frac{n}{s} \tag{301}$$

and since $k = n^t$ and $s = n^r$, this gives us a condition that $t < 1 - r$ for survival to dominate contamination from favored features.

This $t < 1 - r$ condition turns out not to care about whether we are in the regime where regression works (i.e. $q < 1 - r$) or not.

To understand the impact of contamination due to unfavored features, it is important to recall that all of that gets split across the $\frac{d}{n}$ aliases in our heuristic calculation and so the standard deviation of the contamination due to unfavored features scales as $\sqrt{\frac{n}{k} \cdot \frac{n}{d}} = \frac{n}{\sqrt{kd}} = n^{1 - \frac{p}{2} - \frac{t}{2}}$. Meanwhile, survival behaves as $\alpha \frac{\sqrt{n}}{k} = \alpha n^{\frac{1}{2} - t}$.

For survival to dominate contamination from unfavored features, we need:
$$\alpha n^{\frac{1}{2} - t} \gg n^{1 - \frac{p}{2} - \frac{t}{2}} \tag{302}$$
$$\alpha \gg n^{\frac{t - (p-1)}{2}} \tag{303}$$
$$\alpha n^{\frac{p-1}{2}} \gg n^{\frac{t}{2}} \tag{304}$$

Substituting in for the case $q < 1 - r$ from (299), we immediately see the condition $t < p - 1$ for survival to dominate contamination from unfavored features.

For the case $q > 1 - r$, the substitution of $n^{-(q - (1-r))}$ for $\alpha$ in (304) gives us:
$$\frac{t}{2} < \frac{p-1}{2} - (q - (1 - r)) \tag{305}$$
$$t < (p - 1) - 2(q - (1 - r)). \tag{306}$$

Notice that at the boundary $q = (1 - r)$, the condition (306) matches the $t < p - 1$ condition for the region $q < 1 - r$.

Finally, note that our model requires $t < r$ so that the features that determine class labels are themselves favored. Putting all the terms together, we get the condition:
$$t < \min(r, 1 - r, (p - 1) - 2 \cdot \max(0, q - (1 - r))) \tag{307}$$
which agrees with the overall conjectured condition in (23).

## K.2   Why is this sufficient for multiclass classification

Notice the heuristic derivation above is actually just saying that binary classification will succeed (i.e. we will typically have the $m$-th learned score agree in sign with the actual $m$-th feature for a test point) in the given regime, assuming we trained with multiclass training data.

But successful generalization for multiclass classification requires more — it requires the true class's score to win a competition against the competing classes.

Here, we can simply leverage a quick heuristic calculation involving order statistics. We know that the max of $k$ i.i.d. random variables will concentrate to a neighborhood of where the Complementary Cumulative Distribution Function (CCDF) of the underlying random variable reaches $\frac{1}{k}$. Meanwhile, the second biggest of those variables will be around where CCDF hits $\frac{2}{k}$. For a standard Gaussian, these are basically at $\sqrt{2 \ln k}$ and then at $\sqrt{2(\ln k - \ln 2)}$. The gap between these is
$$\sqrt{2 \ln k} - \sqrt{2 \ln k - 2 \ln 2} \approx \frac{1}{2\sqrt{2 \ln k}} 2 \ln 2 \tag{308}$$
$$= \frac{\ln 2}{\sqrt{2 \ln k}} \tag{309}$$
and so what matters is that the contamination relative to the survival needs to go to zero faster than $\frac{1}{\sqrt{\ln k}} = \frac{1}{\sqrt{t \ln n}}$. However, all the scalings here are polynomial in $n$ and so if the contamination is asymptotically smaller than survival, it is smaller by a factor that is polynomial in $n$. This will beat the $\frac{1}{\sqrt{\ln n}}$ scaling of this gap, resulting in successful multiclass classification.

Notice that we crucially used the Gaussian nature of the underlying features[19] here. If the underlying features were uniformly distributed on $[-1, 1]$ instead, then the gap between the biggest and second biggest feature would instead decay polynomially like $\frac{1}{k}$. In that case, the relative contamination would have to go to zero fast enough to overcome this, making the conjectured multiclass region different.

---

[19]Any sufficiently thick tail would do. For example, an exponential or Laplacian feature would also work here.

### K.3 Comments on the power of the heuristic calculation

The heuristic calculation being done here operated at a level of abstraction for which the fine details of the model do not matter. The total energy in the $\mathbf{y}_m$ just comes from the nature of multiclass training data where every point is labeled with only one class as its label and we have a balanced training set. The dominant contamination analysis follows simply from that.

All we need to verify is the survival analysis for alternative models. Here, it is also immediately clear that the asymptotic survival calculation would work just as well if one used a Gaussian mixture model with orthogonal class means or a logistic-softmax model with orthogonal classes — like the models in Wang et al. [77].

In fact, based on this heuristic calculation, we can conjecture that even if the training data were less informative (e.g. the training label did not actually correspond to the class whose feature had the true max but instead, just one of the positive feature classes was chosen uniformly at random as the training label), multiclass classification would still asymptotically succeed in the conjectured regimes in terms of reliably predicting the class whose feature is biggest. What matters is just that positive examples of a class show a significant amount of "classiness" on average in their training features.

Note however that the heuristic calculation here is just that — a heuristic calculation in what some might call the "physics style." Proving the conjectured region is in fact correct requires us to leverage the specific details of the models in our proofs.