# OpenReview forum: "Generalization for multiclass classification with overparameterized linear models"
_NeurIPS.cc/2022/Conference — NeurIPS 2022 Accept_

### Official Review · Reviewer_xQfF · 2022-07-12

**Rating:** 7
**Confidence:** 1
**Soundness:** 3 good
**Presentation:** 2 fair
**Contribution:** 3 good

**Summary:**

The authors present a setup for multiclass classification with specific assumptions. It requires certain assumptions, e.g. the features are drawn from a gaussian distribution, the labels are a deterministic function of the dimension which has largest feature value.

They consider a setup where certain parameters depend polynomially on the sample size, that is the number of dimensions, the number of upscaled dimensions, the number of classes and the feature weights. They present a theorem which shows that the generalization error goes asymptotically to zero for a certain relationship between these parameters.

They show that for some settings multiclass classification converges, but regression does not, conditioned on the specific setup.

**Questions:**

Lemma B.1 is adapted from which result in Bartlett [4] ?

**Strengths And Weaknesses:**

strength:
a novel theoretical result.

weaknesses:
it is a very specific setting.
Readability of the paper is a bit challenging, though it comes from the complexity of the topic itself .

---

> ### Author Response · Authors · 2022-08-02
> **On the source of Lemma B.1**
>
> Thank you for the time spent on reviewing our submission and for recognizing our contributions.
>
> *Re: Lemma B.1 is adapted from which result in Bartlett [4] ?*
>
> Thanks for bringing this to our attention. Lemma B.1 is adapted from Lemma 11 in the first version of the work Bartlett[4] available at https://arxiv.org/pdf/1906.11300v1.pdf In subsequent versions the authors use a slightly weaker version of this result since it is sufficient for their purposes.
>
> We had mentioned this as a footnote in Pg 23 of the supplemental material of the original version when we provide a short proof for Lemma B.1 but as rightly pointed out we should move this footnote to where we first introduce Lemma B.1 in Pg 20 and we have done so in the rebuttal revised version.  Thank you again!

---

> ### Author Response · Authors · 2022-08-02
> **On the specifity of our model**
>
> Thank you for the time spent on reviewing our submission and for recognizing our contributions.
>
> As far as the model goes, it is indeed very specific, but this is done to permit us to distill the core issue of whether indeed multiclass classification can succeed in the regime where regression would not generalize if we did min-norm-interpolation. This regime is captured by the assumption $q+r > 1$ in our stylized bi-level model, and we have added visualizations in Appendix J to show the underlying core spiritual challenge in this regime: namely that the empirical covariance matrix of the training features does not reflect the underlying eigenstructure of the features themselves.

---

### Official Review · Reviewer_u3HB · 2022-07-12

**Rating:** 5
**Confidence:** 2
**Soundness:** 3 good
**Presentation:** 3 good
**Contribution:** 2 fair

**Summary:**

This paper studies the phenomena of benign-overfitting of interpolating minimum norm linear classifiers. The authors consider the case where the number of classes grows with the number of examples. They study a simplified case in which the ground-truth classifier is obtained by the weight matrix (I_k 0) and under the "bi-level ensemble" condition. The results show benign-overfitting can be achieved when the data contain a sufficient number of relevant features and a large number of irrelevant features, and the relevant features have a larger norm than the irrelevant ones.



**Questions:**

The 1-sparse noiseless model is very limiting. Do you think the results can easily be generalized to more general settings (e.g., the Gaussian mixture Wang let. al. [76] or general orthogonal matrix instead of (I_k 0) )?

Is it possible to provide finite-sample bounds on the probability of error?


**Limitations:**



**Strengths And Weaknesses:**

Strength:
- The paper is well-written.
- The topic is relevant to the ML community.
- The results are, to the best of my knowledge, sound.

Weakness:
- The ground-truth model is simple, and it is unclear how to generalize it to more realistic models.
- The results are stated in the asymptotic regime where the number of examples approaches infinity.

---

> ### Author Response · Authors · 2022-08-02
> **On extending our results to more general settings**
>
> Thank you for the time spent on reviewing our submission and for recognizing our contributions.
>
> *Re: The 1-sparse noiseless model is very limiting. Do you think the results can easily be generalized to more general settings (e.g., the Gaussian mixture Wang let. al. [76] or general orthogonal matrix instead of (I_k 0) )?*
>
> Thank you for raising this interesting question. The way we had initially mentioned this assumption made it appear as if our results were more restrictive than they actually are. In fact, our analysis and results hold as long as the class generation vectors $\mu_m$ (from footnote 6 of Pg 4) are orthogonal and have support only in the first s dimensions (i.e where the features are favored under our bi-level model).  This is because our minimum norm interpolator will be identical as long as the above condition is satisfied, so for simplicity we assumed that the $\mu_m$ were one sparse. For more clarity, we have renamed Assumption 4.1 (from Pg 4) to “Orthogonal classes noiseless model”  from “1-sparse noiseless model” and made the generality of our analysis more clear in the corresponding footnote in the rebuttal revised version.
>
> Further, our analysis also extends relatively easily to other settings like the multinomial logistic model (from Wang et. al [77]) with the key difference being how we compute the correlations between the covariates and the labels. (Section D.3, Pg 32 of Supplemental material).  We believe the techniques used in our proofs and the survival/contamination style of analysis can be used to analyze the Gaussian mixture model from Wang et. al [77].  Here, there are more technical challenges to overcome arising from the difference in how the covariates and labels are related to each other as compared to our model but a good starting point for the analysis is the condition for classification error in the GMM setting from Appendix D of Thrampoulidis et al. [72]) .
>
> Stepping back a bit, the reason that we study a stylized/idealized model in an asymptotic regime with our bi-level model is to be able to distill and isolate a particular question. We have added Appendix J in our Supplemental Materials to help people understand more viscerally what this question is. When $q+r > 1$, we are in the regime where the empirical covariance matrix of the training data no longer accurately reflects the underlying structure of the true covariance matrix governing these features. The work done here shows that nonetheless, it is possible to achieve successful generalization for multiclass classification as long as the number of classes is not growing too fast. This is fundamentally interesting, because it speaks to the level of inductive bias that a learning model needs to be successful.

---

> ### Author Response · Authors · 2022-08-02
> **On finite sample bounds on probability of error**
>
> Thank you for the time spent on reviewing our submission and for recognizing our contributions.
>
> *Re: The results are stated in the asymptotic regime where the number of examples approaches infinity.
> Is it possible to provide finite-sample bounds on the probability of error?*
>
> Yes, our analysis in the proof of Theorem 1 from Section B of the supplemental material (Pg 21-22) shows by summing up probabilities of misclassification events that for a large enough number of training samples, $n$, the probability of correct classification is lower bounded by $1 - \epsilon - c/n$ (and thus upper bounding the probability of error by $\epsilon + c/n$). Here, $c$ is some computable universal positive constant independent of $n$ and $\epsilon$ is a fixed small positive real number that shows up because of our analysis of the closest feature margin in Lemmas B.2 and B.3.
>
>
> It is an interesting question for future work as to whether this closest feature margin bound can be improved to generate a finite-sample probability bound as well. Since this is related to understanding the finite-sample convergence of tail order statistics for Gaussians, we believe that this should be possible to do with enough grinding.

---

### Official Review · Reviewer_P5J7 · 2022-07-13

**Rating:** 6
**Confidence:** 4
**Soundness:** 4 excellent
**Presentation:** 3 good
**Contribution:** 2 fair

**Summary:**

The paper derives scaling conditions under which the test error of min-norm interpolation for linear k-class classification goes to zero. The analysis is done for Gaussian features, a noiseless model with orthogonal 1-sparse "true" parameter vectors for each class and a bilevel covariance model. The derived scaling conditions are then used to compare k-class classification to binary and to regression.

**Questions:**

* What do we learn from comparing regimes for classification and regression? To certain extent, the fact that classification is less challenging is somewhat to be expected. At the end of the day, the metrics are different, so what is the value of the comparison? Specifically, what is the lesson learnt (perhaps algorithmic?) by such accurate characterizations (in terms of the bilevel ensemble parameter) regimes? While I appreciate the hard work in deriving the bounds that lead to say Figure 2, could the authors comment on what (if any) the take-away message is (beyond multiclass can be harder than binary which can be harder than regression).

* Can the analysis be extended to other models? Eg logistic, Gaussian mixture etc? Could the authors comment on the possible challenges?

**Ethics Review Area:**

["I don’t know"]

**Limitations:**

yes

**Strengths And Weaknesses:**

Strenghts
-----------------------
* The paper is nicely written and relatively easy to follow
* Good comparison to related work and contextualization of results
* Compared to previous work [55] on binary classification, new technical challenges arise in multiclass settings and the authors do a good job in explaining them, as well as, how they resolve
* Compared to previous work [76] on linear multiclass classification, the bounds have a tighter dependence on the number k of classes. In particular k is allowed to scale with n
* While I did not have time to check the proofs in details, the results appear correct

Weakness
-------------------------
* The paper follows along the lines of recent line of work that compares classification to regression, but is not made clear what is it that we learn from this comparison.
* The discussion on interpretation of results + proof challenges is somewhat heavily using the notions of survival/contamination, which might make the presentation less accessible to wider audience (eg those not familiar with [55]). Perhaps at least define those? (hence "3" on Presentation)
* The model is quite oversimplified. While much related work uses Gaussianity, the 1-sparse assumption together with the bilevel ensemble appears more stringent. It is also somewhat disheartening that even in this setting, the full regime (Conjecture 6.1) is not characterized.

Overall, I value this a solid set of rather non-trivial result that continue recent investigations on overparameterized linear classification. While the set of tools and ideas is perhaps not particularly new (compared to eg [4,54,55,76]) there are certain non-trivial technical challenges that are being addressed

---

> ### Author Response · Authors · 2022-08-02
> **On what we learn from comparing classification to regression in the paper's regimes**
>
> Thank you for the time spent on reviewing our submission and for recognizing our contribution.
>
> *Re: What do we learn from comparing regimes for classification and regression? To certain extent, the fact that classification is less challenging is somewhat to be expected. At the end of the day, the metrics are different, so what is the value of the comparison? Specifically, what is the lesson learnt (perhaps algorithmic?) by such accurate characterizations (in terms of the bilevel ensemble parameter) regimes? While I appreciate the hard work in deriving the bounds that lead to say Figure 2, could the authors comment on what (if any) the take-away message is (beyond multiclass can be harder than binary which can be harder than regression).*
>
> This is indeed an important question. Let’s consider the core spirit of the main result in this paper: the surprising fact that both binary and multiclass classification can asymptotically succeed (generalize well) with a level of feature favoring that would not permit regression to succeed. (e.g. Bartlett, et. al.’s seminal paper on “Benign Overfitting” gives conditions for regression to succeed. What this shows is that those conditions are conservative for classification problems.)  Why is this important? It is important because it could have something to say about how to map problems into learning models. Essentially, the architectures that we use for learning are there to provide an inductive bias towards the actual patterns that we want to learn. Intuitively, having a lower-level of such inductive bias is presumably easier to engineer than having a very strong bias, especially in the context of real-world problems where we don’t actually a-priori know (as humans) the structure of the actual patterns themselves. What our work says is that to the extent that such problems can be made to behave like classification problems (where there is a competition between learned things in making predictions), they might be able to work even when a direct regression-style approach does not work.
>
> The specificity of the bi-level model used to prove the result is just there to both permit theoretical tools to be brought to bear as well as to try and distill out a single aspect of the problem while excluding anything else that might complicate it further. The fact that such a result holds makes it worthwhile to continue exploring this regime to both catalog and understand other phenomena that exist here. What is “this regime?” In the paper, it is what we capture by the dry condition $q+r > 1$. We have added a Section J to the Appendix in the Supplementary Materials to help people understand the qualitative nature of this regime — this is the regime where the empirical covariance of the training features does not accurately reflect the underlying covariance of the features themselves. This is the contrast to the regression-works regime of Bartlett, et. al.’s “Benign Overfitting” result which is essentially in the regime where the empirical covariance would reflect the very important eigenvalues properly.
>
> This latter connection is what makes the work here particularly interesting and we believe important. Successful learning and generalization for supervised patterns is possible using straightforward interpolation (which is what SGD on cross-entropy would converge to in these regimes, etc.) even in settings where we do not have enough training data to learn the hidden essentially lower-dimensional structure hidden in the covariates/features themselves.
>
> It will of course take a lot more work to fully flesh out the story here, but it is an important story to start fleshing out and one where members of the community can readily join in — hence the need for a paper at this stage.
>
> An example of an interesting question and insight provided by this analysis is represented by what we do in the Supplementary Material where we ask what would happen if we scaled in a way that focused on the number of positive training examples per class instead of the total number of training points. With this scaling, the multiclass classification problem behaves much more like binary classification.

---

> ### Author Response · Authors · 2022-08-02
> **On extending our analysis to other models**
>
> Thank you for the time spent on reviewing our submission and for recognizing our contributions.
>
> *Re: Can the analysis be extended to other models? Eg logistic, Gaussian mixture etc? Could the authors comment on the possible challenges?*
>
> Thank you for raising this question.  Our method of analysis extends relatively easily to the multinomial logistic model (like from Wang et. al [77]) with the key difference and technical challenge being how we compute the correlations between the covariates and the labels. (Section D.3, Pg 32 of Supplemental material).
>
> We believe the techniques used in our proofs and the survival/contamination style of analysis can also be used to analyze the Gaussian mixture model. Here, there are more technical challenges to overcome arising from the difference in how the covariates and labels are related to each other as compared to our model but a good starting point for the analysis is the condition for classification error in the GMM setting from Appendix D of Thrampoulidis et al. [72]) .
>
> To make the underlying story clearer, we have added an Appendix K that contains a heuristic derivation of the main result using a stylized and idealized analysis of the type done in Appendix A of Muthukumar, et. al.’s paper on binary classification. This heuristic analysis also sheds light on why other models should behave similarly.
>
> Furthermore, we would like to thank the reviewer for their comment and suggestion regarding our use of survival/contamination notation and as suggested we have moved the description of these quantities that were originally provided in the related work section to the contributions section in the rebuttal revised version. Now, the descriptions immediately follow the introduction of these terms.
>
> As pointed out by other reviewers as well, our initial submission was written in a manner that made our model appear more restrictive than it actually is due to the name of the 1-sparse noiseless assumption. In fact, our analysis and results hold as long as the class generation vectors $\vec\mu_m$ (from footnote 6 of Pg 4) are orthogonal and have support only in the first $s$ dimensions (i.e where the features are favored under our bi-level model).  This is because our minimum norm interpolator will be identical as long as the above condition is satisfied, so for simplicity of notation and exposition we assumed that the $\vec\mu_m$ were one sparse. For more clarity, we have renamed Assumption 4.1 (from Pg 4) to “Orthogonal classes noiseless model”  from “1-sparse noiseless model” and made the generality of our analysis more clear in the corresponding footnote in the rebuttal revised version.

---

> ### Comment · Reviewer_P5J7 · 2022-08-10
> **Thanks for the response**
>
> I thank the authors for their detailed response. Overall, I believe this is a rather interesting and technically solid contribution that I am happy to see published. The work points out some interesting open questions. While answering some of these would strengthen the paper even further, it still adds value sharing with the community. I maintain my score.

---

### Official Review · Reviewer_Spga · 2022-07-15

**Rating:** 7
**Confidence:** 4
**Soundness:** 3 good
**Presentation:** 3 good
**Contribution:** 3 good

**Summary:**

As a purely theoretical work, this paper aims to study the generalization of overparameterized linear models (i.e. minimum-norm interpolator) for multi-class classification in an asymptotic setting where both the number of underlying features and the number of classes scale with the number of training points. Specifically, it considers a bi-level overparameterized linear model with a data generating model with Gaussian features (i.e., $1$-sparse noiseless model). Technically, the survival/contamination analysis framework is adapted to this setting with some novel arguments. Theoretically, the results offer several insights, especially showing that the multiclass problem is “harder” than the binary problem because there are fewer positive training examples per class.

**Questions:**

1. Can the theoretical results be connected with some experimental phenomena in practice? Please give more discussions.
2. This work is the first one (as I know) to study the multiclass classification in an asymptotic setting, and it is enough to consider the overparameterized linear model. As for the non-linear model, more discussions can be added.

**Limitations:**

Although this is a purely theoretical work, it is better to add some experimental results to support the theory.

**Strengths And Weaknesses:**

I am not an expert in the area of overparameterized regimes and benign overfitting-related topics, but I think this work is solid based on my familiar related literature although I may not be familiar with some detailed techniques.

### Originality
The results for multi-class classification in an asymptotic setting are novel although the main techniques are adapted from the binary one [55]. Besides, it considers the asymptotic setting while the related work [75] considers the non-asymptotic setting.

### Quality
This work is of high quality and sound. Besides, it clearly discusses the difference with the related work [75,55]. I have not carefully checked the proofs and it seems sound. My main concerns are the connections (or insights ) of these theoretical results with the experimental phenomena. Besides, although this is a purely theoretical work, it is better to add some experimental results to support the theory. Moreover, this paper considers the overparameterized linear model, and the non-linear model setting (including deep neural networks) can be discussed more.

### Clarity
This paper is well written and organized.

### Significance
This work considers the generalization of the overparameterized linear model regime and should be of interest to the community of modern deep learning theory, which is fundemental in the understanding of the success of modern deep learning in practice.

---

> ### Author Response · Authors · 2022-08-02
> **On experimental evidence**
>
> Thank you for the time spent on reviewing our submission and for recognizing our contributions.
>
> *Re: Can the theoretical results be connected with some experimental phenomena in practice? Please give more discussions.*
>
> As suggested, we have added a section (section I in the supplemental material  in the rebuttal revised version), that provides experimental evidence in support of our theoretical analysis. Because the theoretical analysis relies critically on the behavior of survival and contamination,  we use simulation data to plot the relevant survival, contamination and survival/contamination ratio from Equation 22 in our proof of Theorem 1. We consider the regime where $q + r = 1.05 > 1$, so regression does not work but the values of $p,q,r,t$ are such that we conjecture that multiclass classification still succeeds. We have also included a plot on how the binary classification error (i.e probability of misclassifying the true class as one particular other class) scales as we increase n and we see a clear downward trend. (This error probability times k gives an upper bound on multiclass classification error). Finally, the total misclassification error also exhibits a downward trend though the asymptotics have not kicked in sufficiently to drive this error to zero. The n we chose for the plot were dictated by our limited compute budget.
>
> We also added a second set of empirical plots in section J that serve to illustrate why the regime $q+r > 1$ is conceptually interesting. It turns out that in this regime, the overparameterization is such that the empirical covariance matrix of the features on the limited data no longer clearly reveals the true eigenstructure of the underlying covariance matrix. This is an established fact in the literature, but we hope that seeing the empirical plot helps readers understand why the fact that classification can be made to work in this regime is so interesting. Purely as a matter of speculation, this also points to the challenge in finding empirical evidence in the “wild” (i.e. using real-world data for a real-world learning problem) for this behavior. Just looking at the empirical covariance matrix of training data wouldn’t be able to tell us that we were in a regime where there were actually a few favored directions in the data — it might look like there are in fact lots of directions even though in reality, there are a few directions that are favored enough to let classification succeed. Doing the careful experiments to tease out this effect using real-world data is follow-on work that we hope will be prompted by this paper.

---

> ### Author Response · Authors · 2022-08-02
> **On potential implications for non-linear models**
>
> Thank you for the time spent on reviewing our submission and for recognizing our contributions.
>
> *Re: This work is the first one (as I know) to study the multiclass classification in an asymptotic setting, and it is enough to consider the overparameterized linear model. As for the non-linear model, more discussions can be added.*
>
> Indeed, the goal here is to build up the community’s understanding of the underlying phenomena to be able to say more about the non-linear models. Since our paper (and our contributions) are about the asymptotic case of linear models, and the page budget is limited, we did not feel it was appropriate to comment extensively on the non-linear situation. However, at this point, the general view of the community on the relationship between linear models and nonlinear ones goes something like this:
>
> In real-world models, we have nonlinear activation functions. Collectively, the nonlinearities in the model serve to perform “lifting” whereby the underlying inputs are lifted to a higher-dimensional space. Because the lifting is nonlinear, these higher-dimensional features behave in a manner that has non-colinearity with other features. When sufficiently overparameterized and initialized appropriately using an appropriate learning rate, both (some) empirical and (some limited) theoretical evidence indicates that often (but not always), the final learned solution is not that far from the initialization. In these cases, we can linearize the nonlinear model in the neighborhood of the initialization giving rise to what is called the Neural Tangent Kernel (NTK) perspective where each learnable model weight corresponds to a generalized linear feature. For such models, if we learn by using stochastic gradient descent to converged zero loss, we can view the final learned result as a kind of minimum norm interpolation where the covariance structure of these nonlinear features is what matters. The results here (along with others in the community) suggest that learning can succeed when those nonlinear features collectively sufficiently favor a limited number of underlying directions that are also relevant for representing the patterns that we want to learn.
>
> Of course, the above is far from the complete story as we know that there are also many interesting examples that seem to operate beyond where the NTK approximation is valid. But it is a first step. In our work (as well as many others) we consider an incoherent Gaussian feature family to be the underlying features and as a sort of abstraction of how a non-linear lifted feature space might look like from the perspective of the optimization/learning algorithm. Basically, the randomness and the Gaussianity are a way for us to idealize the belief that the nonlinear lifting is not adversarial against the true pattern, and in fact, the true patterns are actually favored to be learned. This paper’s focus on the q+r > 1 regime reflects our desire to understand whether things can work when the nonlinear lifting just favors the true patterns a little bit instead of in an overwhelming way.

---

### Author Response · Authors · 2022-08-02
**Overall summary of rebuttal revision**

Thank you to all the reviewers for their careful reading of the paper and thoughtful comments. Here, we’d like to point out how we have incorporated those comments in our “rebuttal version.”

Before doing that, we want to point out that we have fixed the Conjecture in the conclusion of the paper with a minor fix to reflect the behavior that occurs in the regime where regression works. This did not impact any of the plots in the paper because this small difference didn’t matter for the slices that we used to illustrate the result.

In terms of edits, we renamed the model for how the class labels are generated from “1-sparse noiseless model” to the more accurate “orthogonal classes noiseless model” since the underlying assumption is that the different classes are defined by orthogonal directions in feature space. This also makes the name closer to that used in Wang, et al, which is the closest piece of related work. We also moved a footnote in the Supplemental material to be where a particular Lemma from Bartlett et al is first introduced instead of later, in response to a comment from a reviewer who helpfully pointed out a question. Finally, in response to a reviewer comment, we moved up the definition of the concepts of survival and contamination to the first place where those ideas are mentioned.

We added Appendices I, J, and K to the supplemental material in response to reviewer comments. Two of these are centered on empirical plots. Appendix I complements the theoretical and asymptotic proofs in the paper with an empirical evaluation of relevant quantities using simulated data for finite values of $n$, the number of training points. We chose a regime where we are conjecturing results so that it is possible to see the very close match between our conjectured predictions for how these quantities should scale and how they actually do in experiments. Appendix J simply plots the difference between the singular values (and hence estimated eigenstructure) of the empirical feature matrix and the eigenstructure of the underlying features themselves. While this behavior is well known in the literature (we added a reference to Wang and Fan in the bibliography for this purpose), the plots illustrate the underlying challenge provided by the regime in which regression does not generalize — namely that the empirical eigenstructure does not reveal the true nature of the underlying features. We hope this helps readers appreciate more why the results here are important and interesting.

The third Appendix K gives a heuristic derivation of the main result using a stylized “physics-style” analysis of the type given in Appendix A of Muthukumar, et. al.’s paper on binary classification. The hope is that this elementary style of analysis, although decidedly nonrigorous, helps shed light on where conjectures are coming from and what is actually going on. This style of analysis also makes it more immediately clear what kind of models the results here should directly extend to in spirit, even though the more intricate and involved analysis of the paper leans into the specific details of the model being studied.

Finally, because we added two references for the aid of readers who might need more background (one to Gallager’s textbook and the other to Wang and Fan), the reference numbers have changed slightly.

---

### Meta-Review · Area_Chair_qntZ · 2022-08-24

**Recommendation:** Accept
**Confidence:** Certain

**Metareview:**

The paper makes a solid progress in our understanding of generalization in overpametrized models

**Award:**

No

---

### Decision · Program_Chairs · 2022-09-14

Accept